# Orphan G protein-coupled receptor GPRC5B controls macrophage function by facilitating prostaglandin E receptor 2 signaling

Jeonghyeon Kwon [1], Haruya Kawase [1], Kenny Mattonet [2], Stefan Guenther [3], Lisa Hahnefeld [4,5,6], Jamal Shamsara[7], Jan Heering [4,5], Michael Kurz [8], Sina Kirchhofer [8], Cornelius Krasel [8], Michaela Ulrich [8], Margherita Persechino[7], Sripriya Murthy[9], Cesare Orlandi [10], Christian D. Sadik[9], Gerd Geisslinger[4,5,6], Moritz Bünemann [8], Peter Kolb [7], Stefan Offermanns [1,11] & Nina Wettschureck [1,11] ✉

Macrophages express numerous G protein-coupled receptors (GPCRs) that regulate adhesion, migration, and activation, but the function of orphan receptor GPRC5B in macrophages is unknown. Both resident peritoneal and bone marrow-derived macrophages from myeloid-specific GPRC5B-deficient mice show increased migration and phagocytosis, resulting in improved bacterial clearance in a peritonitis model. In other models such as myocardial infarction, increased myeloid cell recruitment has adverse effects. Mechanistically, we found that GPRC5B physically interacts with GPCRs of the prostanoid receptor family, resulting in enhanced signaling through the prostaglandin E receptor 2 (EP2). In GPRC5B-deficient macrophages, EP2-mediated anti-inflammatory effects are diminished, resulting in hyperactivity. Using in silico modelling and docking, we identify residues potentially mediating GPRC5B/EP2 dimerization and show that their mutation results in loss of GPRC5B-mediated facilitation of EP2 signaling. Finally, we demonstrate that decoy peptides mimicking the interacting sequence are able to reduce GPRC5B-mediated facilitation of EP2-induced cAMP signaling in macrophages.

Macrophages are specialized cells of the innate immune system that contribute to host defense by phagocytosis of pathogens, secretion of cytokines and antimicrobial agents, and antigen presentation to lymphocytes[1,2]. Macrophages can be divided into two developmentally and functionally distinct populations, bone marrow-derived macrophages (BMDM) and tissue-resident macrophages such as microglia of the brain, dermal Langerhans cells, or resident peritoneal macrophages (RPM)[1,2]. BMDM originate from circulating monocytes that enter damaged tissues during infection or inflammation; they are extremely plastic and can - depending on the local environment -

[1]Department of Pharmacology, Max Planck Institute for Heart and Lung Research, Bad Nauheim, Germany. [2]Imaging Platform, Max Planck Institute for Heart and Lung Research, Bad Nauheim, Germany. [3]Deep sequencing platform, Max Planck Institute for Heart and Lung Research, Bad Nauheim, Germany. [4]Fraunhofer Institute for Translational Medicine and Pharmacology ITMP, Frankfurt am Main, Germany. [5]Fraunhofer Cluster of Excellence for Immune Mediated Diseases CIMD, Frankfurt am Main, Germany. [6]Goethe University Frankfurt, University Hospital, Institute of Clinical Pharmacology, Frankfurt am Main, Germany. [7]Department of Pharmaceutical Chemistry, University of Marburg, Marburg, Germany. [8]Department of Pharmacology and Clinical Pharmacy, University of Marburg, Marburg, Germany. [9]Department of Dermatology, Allergy, and Venereology, University of Lübeck, Lübeck, Germany. [10]Department of Pharmacology and Physiology, University of Rochester Medical Center, Rochester, NY, USA. [11]Centre for Molecular Medicine, Medical Faculty, Goethe-University Frankfurt, Frankfurt am Main, Germany. ✉e-mail: Nina.Wettschureck@mpi-bn.mpg.de

transition from a pro-inflammatory state to a more reparative, anti-inflammatory state[2,3]. Tissue-resident macrophages such as RPM, in contrast, are derived from embryonic progenitor cells that persist and self-renew in their respective niches; they are involved both in homeostatic functions such as clearance of cell debris as well as orchestration of immune responses after damage[4,5].

The mechanisms controlling macrophage activation and differentiation are complex. Pattern recognition receptors such as toll-like receptors are activated by molecular structures on the surface of pathogens or damaged cells and initiate nuclear factor-κB-dependent gene expression, mitogen-activated protein kinase activation, and interferon production[3,6]. In addition, numerous cytokine receptors modulate macrophage activity, prominent among them those for tumor necrosis factor α (TNFα), interferons, and interleukins[3,6]. Also G protein–coupled receptors (GPCRs) are essential regulators of myeloid cell function, for example in chemokine-induced migration[6,7]. In addition, GPCRs mediate the effects of numerous immunomodulatory lipid mediators, for example (lyso)phospholipids, prostanoids and other arachidonic acids derivatives[7–10], Particularly well studied are in this context the prostanoid receptors, which play a central role in inflammation, but also numerous other functions[9,11,12]. The prostanoid family consists of prostaglandins $PGE_2$, $PGD_2$, and $PGF_{2a}$, as well as prostacyclin and the thromboxanes. Prostanoids are already under basal conditions synthesized in numerous cell types, but their production is further induced under inflammatory conditions[12]. Best studied in the context of macrophage activation is $PGE_2$, which is produced at high levels during inflammation, both by macrophages themselves and other cell types[13,14]. In macrophages, $PGE_2$ was shown to decrease pro-inflammatory cytokines (e.g., TNFα) and simultaneously increase anti-inflammatory cytokines (e.g., interleukin-10)[13–15]. $PGE_2$ mediates its intracellular effects through four subtypes of GPCRs, EP1-EP4. In macrophages, $PGE_2$-mediated activation of the $G_s$-coupled subtypes EP2 and EP4 has mainly anti-inflammatory effects, whereas activation of the $G_{q/11}$- and $G_i$-coupled subtypes EP1 and EP3, potentiates inflammation[13–15].

In addition to these well-studied GPCRs with known ligands, immune cells express a number of receptors with yet unknown ligand or function, so-called orphan receptors[16,17]. We previously characterized the expression of GPCRs in different cell populations of the vascular and immune system using single-cell real-time quantitative reverse transcription PCR (qRT-PCR) and bulk RNA analyses[18,19]. We observed that certain orphan GPCRs show particularly high expression in specialized macrophage populations such as RPM, for example G Protein-Coupled Receptor Class C Group 5 Member B, shortly GPRC5B. GPRC5B is a class C orphan receptor whose expression is induced by retinoic acid (RA)[20]. Gprc5b, the mouse gene encoding GPRC5B, is strongly expressed in the central nervous system[21], and global GPRC5B-deficient mice display various mild neurological phenotypes, such as behavioral abnormalities, altered cortical neurogenesis, and disturbed cerebellar morphogenesis and long-term motor learning[22,23]. In addition, reduced obesity and obesity-associated inflammation have been described in Gprc5b knockout mice[23]. Based on in vitro studies it was suggested that GPRC5B negatively regulates insulin secretion[24] and influences inflammatory cytokine production and fibrotic activity in cardiac fibroblasts[25]. How GPRC5B contributes to these processes is largely unknown. In vascular smooth muscle cells (SMC), we recently showed that GPRC5B regulates prostacyclin receptor signaling and thereby modulates SMC differentiation and contractility[26].

In this work, we show that in the immune system, GPRC5B expression is particularly high in RPM, and that this orphan receptor plays a crucial role in the regulation of macrophage functions by controlling intracellular localization and signaling of the EP2 receptor.

## Results

### Generation and characterization of myeloid-specific GPRC5B-deficient mice

RNA sequencing showed that BMDM and RPM express numerous orphan GPCRs, most prominently in both populations Gprc5b, Gpr35, Gpr146, Gpr65, Gpr183, and Gpr132 (Fig. 1A). While the role of GPR35, GPR65, GPR183 (aka EBI2), and GPR132 (aka G2A) has been studied previously in immune cells[27–29], nothing is known about the function of GPRC5B in leukocytes. NanoString-based expression profiling confirmed the abundant expression of Gprc5b in RPM, but did not detect transcripts in lymphocytes (Fig. 1B). To study the role of GPRC5B in RPM and other myeloid cells, we bred recently generated Gprc5b-flox mice[26] to LysM-Cre mice expressing recombinase Cre under control of the myeloid-specific LysM promoter (B6.129P2-Lyz2$^{tm1(cre)Ifo}$/J)[30]. QRT-PCR and western blotting of RPM showed a clear reduction of GPRC5B expression in LysM-Cre-positive; Gprc5b$^{fl/fl}$ mice (henceforth myeloid-Gprc5b knockout mice, M-G5b-KO) (Fig. 1C, D). In a first set of experiments we investigated whether loss of GPRC5B altered numbers of leukocytes in different immune compartments, but did not find significant differences for myeloid cells or lymphocytes (Supplementary Fig. 1A-F). Since GPRC5B expression is highest in RPM, we next investigated consequences of GPRC5B deficiency in RPM under basal conditions and after stimulation with lipopolysaccharide (LPS). Gene expression analysis by qRT-PCR showed enhanced expression of inducible nitric oxide synthase (Nos2) (Fig. 1E), while other inflammatory markers such as Tnf or Il6 did not show significant changes (Fig. 1F, Supplementary Fig. 1G). Also RNA sequencing did not reveal clear differences in the expression of macrophage marker genes between the genotypes (Supplementary Fig. 1H–J). We then tested whether key macrophage functions such as phagocytosis or migration were altered and found that LPS-treated RPM from M-G5b-KOs showed enhanced chemokine-induced migration both in transwell assays and live cell imaging (Fig. 1G, H). Furthermore, phagocytosis of E. coli particles and apoptotic cells was significantly increased in RPM from KO mice (Fig. 1I-L). To determine the relevance of increased migration and phagocytosis in RPM in vivo, we induced bacterial peritonitis in mice using a fecal bacteria transfer model. M-G5b-KO mice showed reduced body weight loss and the peritoneal lavage fluid harvested after 24 h contained significantly lower numbers of colony forming bacteria (Fig. 1M, N). When analyzing peritoneal leukocyte populations before and after bacteria inoculation, we found that resident macrophage populations showed in both genotypes the expected macrophage disappearance reaction (Fig. 1O), and that recruitment of bone marrow derived macrophages was more efficient in M-G5b-KO (Fig. 1P). Recruitment of other leukocyte populations such as T cells or granulocytes was not altered (Supplementary Fig. 1K, L). Taken together, GPRC5B-deficient RPM are characterized by increased migration and phagocytosis, resulting in improved bacterial clearance in a peritonitis model.

### Enhanced activity in GPRC5B-deficient BMDM

The increased recruitment of monocyte-derived macrophages during bacterial peritonitis suggests that also BMDM function is altered in M-G5b-KOs, which led us to study consequences of GPRC5B deficiency in these cells. As observed in RNA sequencing (Fig. 1A), Gprc5b transcript levels were lower in BMDM than in RPM, but KO mice still showed significant further reduction of Gprc5b expression (Fig. 2A). This resulted in an enhancement of LPS-induced expression of proinflammatory genes such as Nos2 and Il1b (Fig. 2B, C), and LPS-induced production of nitrogen oxides (NOx) and proinflammatory cytokines was increased (Fig. 2D, E). Furthermore, we found that chemokine-induced migration and phagocytosis of E. coli particles was significantly increased in KO cells (Fig. 2F–H). To rule out that the observed phenotypes were due to a yet unknown effect of LysM-Cre expression in macrophages, we repeated phagocytosis studies in WT

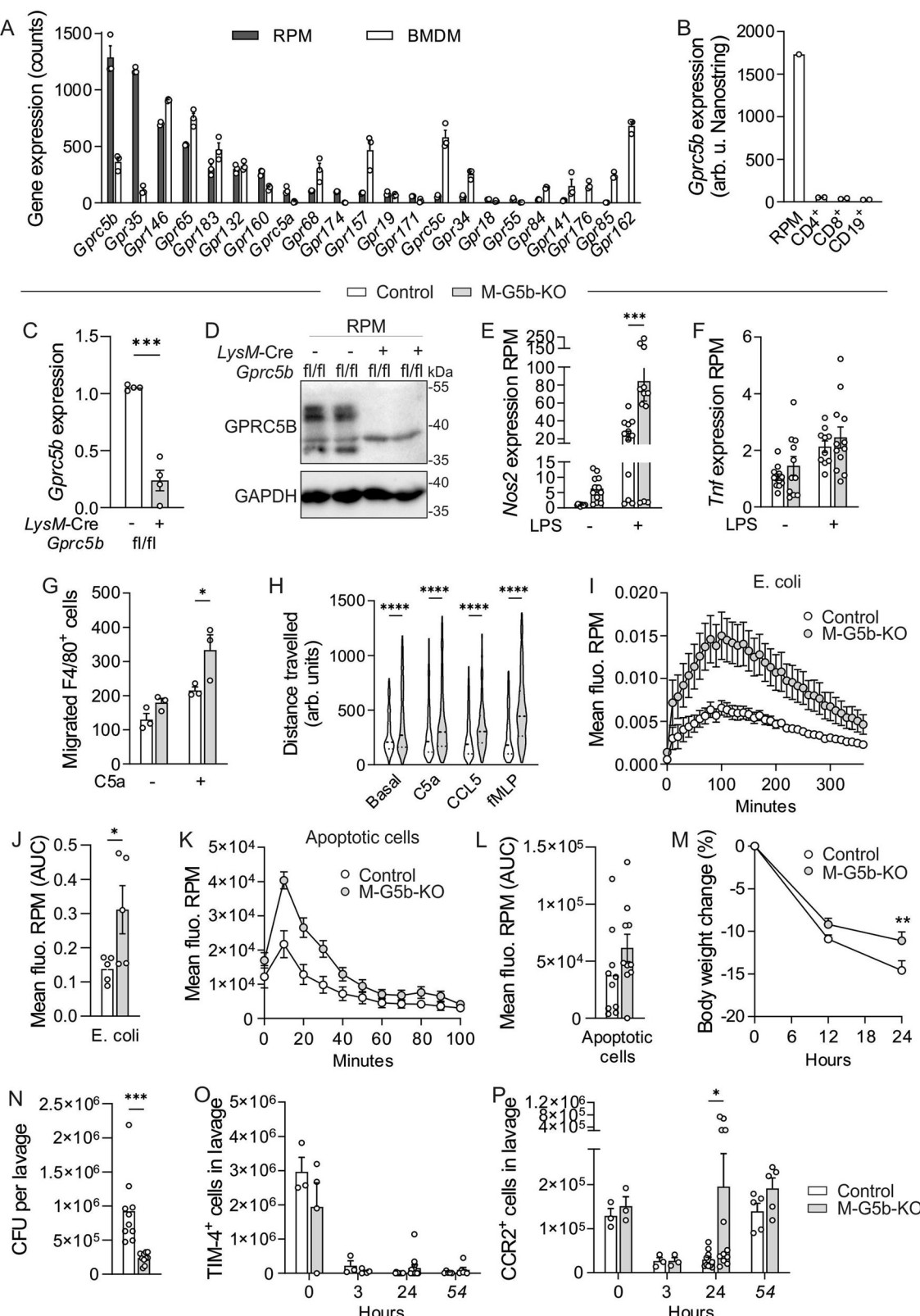

mice with or without *LysM*-Cre, but did not find significant differences (Supplementary Fig. 2A, B). In contrast, comparison of *LysM*-Cre + ;*Gprc5b*(fl/fl) mice with *LysM*-Cre + ;*Gprc5b*(wt/wt) littermates reproduced the hyperactivation phenotype (Supplementary Fig. 2C, D).

Altered responsiveness of BMDM might affect immune responses beyond acute defense against peritoneal pathogen invasion, which led

us to study M-G5b-KOs in two disease models that involve damage-induced monocyte/macrophage recruitment and activation, myocardial infarction and dextran sulfate sodium (DSS)-induced colitis[31,32]. Four days after induction of myocardial infarction by ligation of the left anterior descending artery, we found significantly increased numbers of CD11b-positive myeloid cells in ischemic zone and border zone of KO mice (Fig. 2I). In contrast, myeloid cell numbers did not differ

**Fig. 1 | Generation of myeloid-specific GPRC5B-KOs and role of GPRC5B in RPM.**
**A** Library size-normalized counts detected by RNA sequencing in mouse RPM and in M0-differentiated BMDM (3 mice each; only GPCRs with average count >15 displayed). **B** *Gprc5b* expression in RPM and lymph node-derived lymphocytes was determined by NanoString RNA analysis (cells pooled from two mice per data point). **C** Knockout efficiency in RPM from control mice (white) and M-G5b-KOs (gray) was analyzed by qRT-PCR (C, n = 4, data normalized to *Gapdh* and controls set to 1). **D** Knockout efficiency in RPM was analyzed by immunoblotting (unspecific band around 38 kDa; the higher of the two specific bands probably represents glycosylated GPRC5B[21]; GAPDH as loading control. **E, F** Expression of *Nos2* (E) and *Tnf* (F) was determined in RPM by qRT-PCR under basal conditions and after 6 h of stimulation with 1 μg/ml LPS (n = 11/12/12/12 in E, 12/10/11/12 in F), data normalized to *Gapdh* and control set to 1. **G** Basal and C5a (20 ng/ml)-induced transwell migration in RPM (all cells pretreated with LPS 1 μg/ml for 3 h to facilitate migration) (n = 3). **H** The distance travelled by individual RPM in response to different chemotactic factors (C5a, 20 nM; CCL5, 10 ng/ml; fMLP, 10 nM) was determined by live cell imaging (n = 512 cells from 2 mice per group; cells pretreated with LPS 1 μg/ml for 3 h to facilitate migration, arb. units: arbitrary units). Phagocytic activity of LPS (1 μg/ml, 6 h)-stimulated RPM was determined by uptake of pHrodo E. coli bioparticles (**I, J**, n = 5) or pHrodo-labeled apoptotic thymocytes (**K, L** n = 10); I + K show original traces, J + L statistical evaluation of areas under the curve (AUC). Body weight change (**M**) and bacterial colony-forming units in peritoneal lavage fluid harvested 24 h after injection of fecal bacteria (**N**) (n = 10). **O, P** Numbers of CD11b[+], F4/80[+], MHCI[r], Tim4[+] RPM and CD11b[+], F4/80[lo], MHCII[+], CCR2[+] BMDM before and 3, 24, and 54 h after i.p. injection of fecal bacteria (n = 3/3/3/3/11/12/5/5). Data are means ± SEM; comparisons between genotypes were performed using unpaired two-sided Student's t-test (C, J, L, N), two-way ANOVA (E-H) or two-way RM-ANOVA (M) with Sidak's multiple comparison test, unpaired two-sided t-test corrected for multiple testing by two-stage step-up method Benjamini, Krieger and Yekutieli (O, P). *P < 0.05; ***P < 0.001; ****P < 0.0001; n, number of individual mice. Source data are provided as a Source Data file.

significantly in the remote zones of the same hearts, nor did they differ in healthy heart tissue or in the blood (Supplementary Fig. 2E–G). Since enhanced myeloid cell infiltration might, depending on time course and polarization state of the cells, have beneficial or detrimental effects on the cardiac damage response[31], we determined scar size and cardiac function 3 weeks after infarction. M-G5b-KOs showed impaired cardiac function compared to control mice (Fig. 2J, Supplementary Fig. 2H–J), and scar size was increased (Fig. 2K, Supplementary Fig. 2K). In the DSS colitis model, in contrast, M-G5b-KO mice showed only a mildly and transiently increased disease activity index (Fig. 2L), and no clear difference in colonic shortening (Fig. 2M). Analysis of immune cell infiltration revealed an increase in CD11b-positive cells in the colonic lamina propria (Supplementary Fig. 2L), but there were no significant differences in the expression of epithelial junction proteins such as occludin or ZO-1, mucin, or inflammatory mediators (Supplementary Fig. 2M, N). Taken together, hyperactivity of GPRC5B-deficient macrophages may be beneficial, neutral, or detrimental depending on the disease model.

Since *LysM*-Cre mediates recombination also in neutrophilic granulocytes, a cell population that contributes to the pathogenesis of colitis and cardiac remodeling, we analyzed consequences of GPRC5B deficiency in these cells. Bone marrow neutrophils from M-G5b-KOs showed reduced *Gprc5b* expression and enhanced chemoattractant-induced migration (Supplementary Fig. 3A, B), whereas phagocytosis was not clearly altered (Supplementary Fig. 3C). To test the relevance of GPRC5B for neutrophil-mediated diseases in vivo, we employed the passive immunization model of bullous pemphigoid-like epidermolysis bullosa acquisita (EBA), a blistering skin disease in which deposition of autoantibodies to type VII collagen at dermal/epidermal junctions drives neutrophilic infiltration and consecutive skin inflammation[33]. However, we did not observe differences between the genotypes with respect to lesion area development or strength and composition of immune cell infiltration (Supplementary Fig. 3D, E). Also in the model of thioglycolate-induced peritonitis, neutrophil recruitment was not altered (Supplementary Fig. 3F). In other populations of the myeloid lineage, such as eosinophils or platelets, *Gprc5b* expression is negligible.

### Mechanisms underlying GPRC5B-dependent regulation of macrophage activity

GPRC5B has previously been suggested to mediate phosphorylation of sphingomyelin synthase 2 and thereby influence ceramide and sphingoid base levels in mouse embryonic fibroblasts[23], but we did not observe clear differences in these lipid species in resting or LPS-treated BMDM (Supplementary Fig. 4).

We recently showed in SMC that GPRC5B regulates the membrane availability of the prostacyclin receptor IP[26]. This receptor is known to mediate anti-inflammatory effects in macrophages, though the effects

of IP receptor agonists may vary in different macrophage populations[34]. We studied the effects of IP receptor agonist iloprost in RPM from control mice and M-G5b-KOs, but did not observe iloprost-induced modulation of inflammatory gene expression (Supplementary Fig. 5A–C) or phagocytic behavior (Supplementary Fig. 5D–F) in either genotype. Also, we did not observe iloprost-induced elevation of cAMP levels, the canonical 2nd messenger downstream of the $G_s$-coupled IP receptor (Supplementary Fig. 5G). This was not due to a general unresponsiveness to $G_s$-coupled receptor activation since butaprost, a stable agonist at the $G_s$-coupled prostaglandin E receptor EP2, was able to induce cAMP production (Supplementary Fig. 5G). It was also not due to inactivity of the ligand itself, since the same iloprost induced cAMP elevation in IP-transfected HEK293T (HEK) cells (Supplementary Fig. 5H). These data indicate that the phenotypic differences observed in GPRC5B-deficient mouse macrophages are not secondary to altered IP signaling.

Since the IP receptor is only one of several structurally related prostanoid receptors expressed in macrophages, we systematically investigated whether also other members of the prostanoid receptor family interact with GPRC5B. To test this, we performed co-immunoprecipitation experiments in HEK cells transfected with C-terminally FLAG/Myc-tagged GPRC5B (GPRC5B-FLAG/Myc) and the nine members of the prostanoid receptor family (all N-terminally HA-tagged): the prostacyclin receptor IP, the prostaglandin (PG) E2 receptors EP1/2/3/4, the $PGD_2$ receptors DP1 and DP2, the $PGF_{2a}$ receptor FP, and the thromboxane $A_2$ receptor TP. Western blotting of proteins precipitated by anti-FLAG-beads ("pulldown FLAG" in Fig. 3A) showed that not only IP co-precipitated with GPRC5B, but also EP1, EP2, and DP1. In contrast, receptors EP3, EP4, DP2, DP, TP did not co-precipitate with GPRC5B (Fig. 3A), and also chemokine receptors such as CCR5 did not interact (Supplementary Fg. 6A). Also when anti-HA-beads were used for precipitation, GPRC5B selectively co-precipitated with IP, EP1, EP2, and DP1 (Supplementary Fig. 6B). Physical interaction between GPRC5B and EP2 was further evidenced by elevated Förster resonance energy transfer in HEK cells (Supplementary Fig. 6C). To study the functional relevance of these findings, we focused on those prostanoid receptors that are robustly expressed in both RPM and BMDM, EP1 (encoded by *Ptger1*), EP2 (*Ptger2*), and EP4 (*Ptger4*) (Fig. 3B). Since GPRC5B is expressed in HEK cells, we first tested whether siRNA-mediated knockdown of GPRC5B affected signaling of the $G_{q/11}$-coupled EP1 receptor or the $G_s$-coupled receptors EP2 and EP4. GPRC5B transcript levels were significantly reduced after knockdown (Supplementary Fig. 6D), but this did not affect calcium mobilization induced by EP1 agonist 17-pt-PGE2 (Fig. 3C). cAMP formation in response to EP2 agonist butaprost, however, was significantly reduced after GPRC5B knockdown (Fig. 3D). Activation of the non-interacting EP4 receptor by agonist L-902688, in contrast, was not affected by GPRC5B knockdown (Fig. 3E).

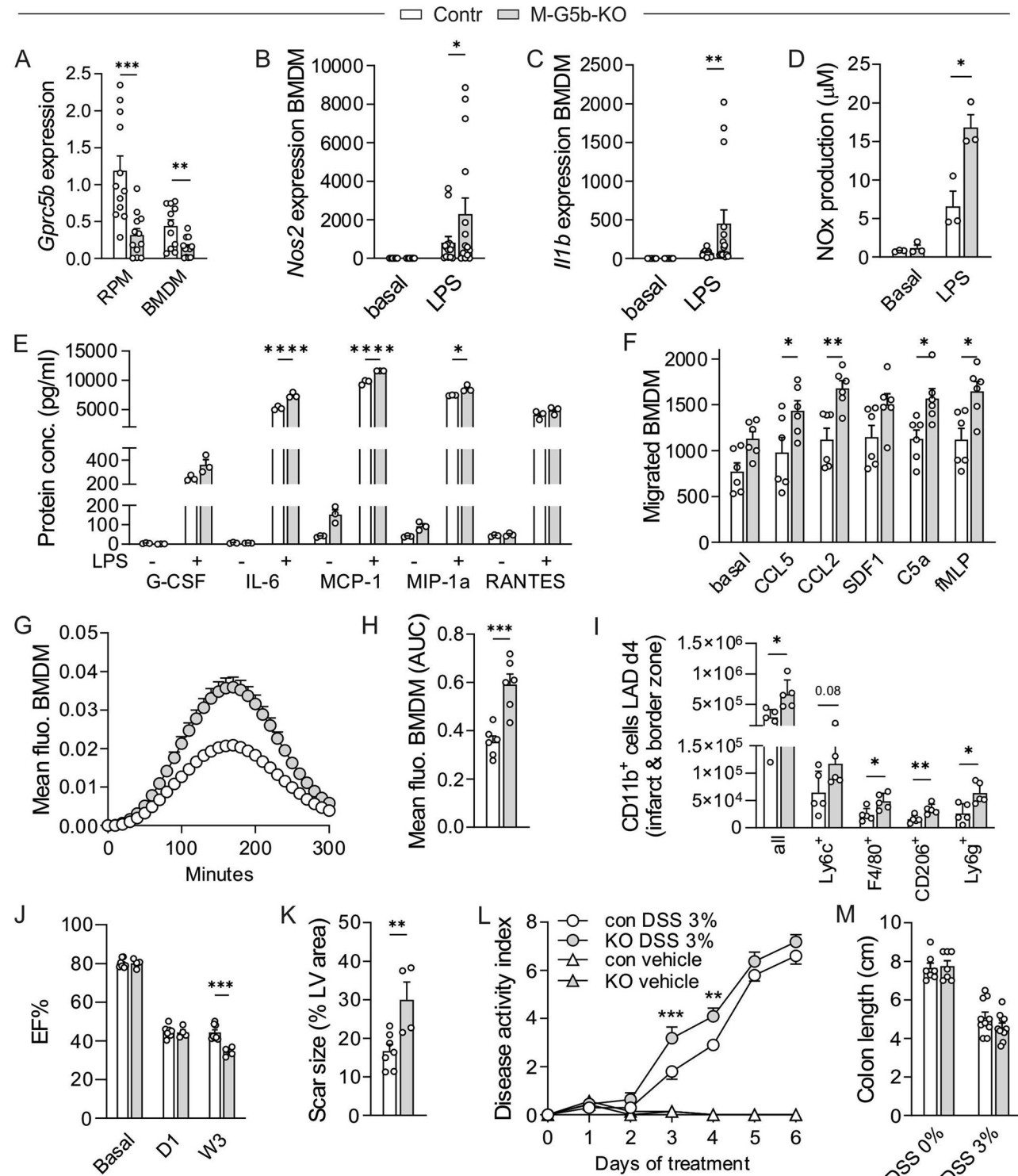

**Fig. 2 | Enhanced activity of GPRC5B-deficient BMDM. A** Knockout efficiency was determined by qRT-PCR in RPM and M0 BMDM (data normalized to *Gapdh* and RPM controls set to 1) (*n* = 12). Analyses in resting and LPS (1 µg/ml, 6 h)-stimulated M0 BMDM: Expression of inflammatory genes (**B**, **C**; *n* = 15/14/15/15 in B, 15/14/15/15 in C), production of NOx (**D**; n = 3) or release of cytokines (**E**, *n* = 3). **F** Transwell migration of M1 BMDM in response to different chemotactic factors (*n* = 6) (CCL5: 75 ng/ml, CCL2: 10 ng/ml, SDF-1β: 100 ng/ml, C5a: 20 ng/ml, fMLP: 10 nM). Uptake of pHrodo E.coli fragments by M0 BMDM: **G**, exemplary curves; **H**, statistical analysis of AUC (*n* = 6). **I** Flow cytometric analysis of CD11b-positive cells in the combined infarct and border zones of hearts harvested 4 days after infarction (*n* = 5). **J** Echocardiographic analysis of ejection fraction (EF%) before and after infarction (8

controls, 4 KOs). **K** Histological analysis of scar size in hearts harvested 21 days after infarction (*n* = 7 controls, 4 KOs), left ventricle (LV). **L, M** DSS colitis: Disease activity index integrating body weight change, stool consistency, intestinal bleeding (L) and colon length on day 6 (M) (*n* = 7(L), 7/7/10/11 in M). Data are means ± SEM; comparisons between genotypes were performed using unpaired two-sided Student's *t*-test (A, H, K), two-way ANOVA with Sidak's multiple comparisons test (B-F, J, M), unpaired two-sided *t*-test corrected for multiple testing by two-stage step-up method Benjamini, Krieger and Yekutieli (I), two-way repeated measures ANOVA with Sidak's multiple comparisons test (L). n, number of mice per group; *P* < 0.05; **P* < 0.01; ***P* < 0.001; ****P* < 0.0001. Source data are provided as a Source Data file.

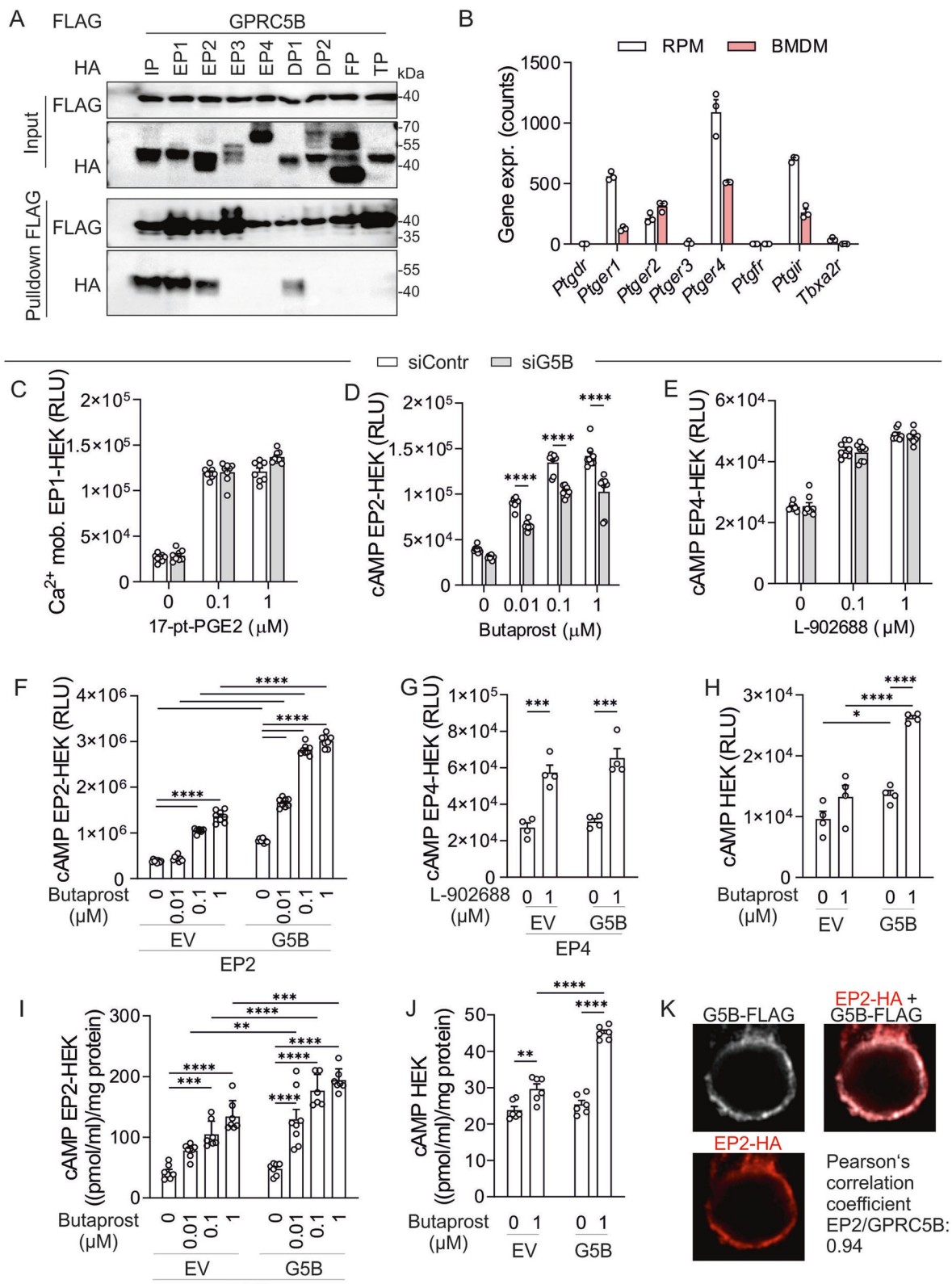

To corroborate the hypothesis that GPRC5B facilitates EP2 signaling, we next tested whether GPRC5B overexpression resulted in increased EP2 signaling. We found that overexpression of GPRC5B facilitated butaprost/EP2-mediated cAMP production compared to empty vector (Fig. 3F), whereas L-902688/EP4-mediated responses were not altered (Fig. 3G). The facilitation of butaprost/EP2 signaling was also observed for endogenously expressed EP2 (Fig. 3H) and

when using ELISA-based cAMP determination instead of luminescence (Fig. 3I, J). In line with a physical interaction between GPRC5B and EP2, we found that in HEK cells co-transfected with GPRC5B-FLAG/Myc and HA-EP2, HA and FLAG signals showed overlapping intracellular distribution (Fig. 3K). Taken together, GPRC5B interacts with EP2 and facilitates EP2-dependent cAMP production in HEK cells.

**Fig. 3 | Interaction of GPRC5B with other prostanoid receptors. A** Western blot detection of HA and FLAG signals in lysates of HEK cells expressing FLAG-tagged GPRC5B in combination with different HA-tagged prostanoid receptors before ("input") and after immunoprecipitation of GPRC5B-FLAG/Myc ("Pulldown FLAG"). **B** Expression of prostanoid receptors was determined by RNA sequencing ($n = 12$) (same samples as in Fig. 1A). **C** Calcium mobilization induced by EP1 agonist 17-pt-$PGE_2$ in HEK cells transfected with calcium sensor aequorin, EP1 receptor, and control siRNA (siContr) or siRNA directed against GPRC5B (siG5B), respectively ($n = 8$). RLU: relative light unit. cAMP production induced by EP2 agonist butaprost (**D**) or EP4 agonist L-902688 (**E**) was determined in HEK cells transfected with a cAMP GloSensor plasmid, EP2 (D) or EP4 (E) receptors, as well as control siRNA or siRNA directed against GPRC5B, respectively ($n = 8$). **F, G** cAMP production induced by EP2 agonist butaprost (F) or EP4 agonist L-902688 (G) in HEK cells transfected with a cAMP GloSensor plasmid, EP2 (F) or EP4 (G) receptors, as well as empty vector (EV) or GPRC5B (G5B) as indicated ($n = 8$ and 4, respectively). **H** Butaprost-

induced cAMP production in HEK cells transfected with cAMP GloSensor plasmid and empty vector (EV) or GPRC5B as indicated (without EP2 transfection; $n = 4$). **I, J** Butaprost-induced cAMP production in HEK cells transfected with empty vector (EV) or GPRC5B in the presence (I) or absence (J) of overexpressed EP2 receptor was determined by ELISA ($n = 6/8/7/7/6/8/6/7$ in I, 6(J)). **K** Spatial correlation between HA and FLAG signals in HEK cells co-transfected with HA-EP2 and GPRC5B-FLAG/Myc: exemplary photomicrographs and statistical analysis (Pearson's coefficient, $n = 29$ cells). Data are means ± SEM; comparisons between genotypes (C-E) or both genotypes and treatments (G-J) were done using two-way ANOVA and Sidak's post hoc test (C-E, G-J) or one-way ANOVA with Tukey's multiple comparison test (F). EV, empty vector; G5B, GPRC5B-encoding plasmid; n, number of independent experiments; siContr, scrambled control siRNA; siG5b, siRNA directed against GPRC5B; *$P < 0.05$; **$P < 0.01$; ***$P < 0.001$; ****$P < 0.0001$. Source data are provided as a Source Data file.

## Altered EP2 signaling in RPM/BMDM

We next investigated whether GPRC5B-mediated modulation of EP2 signaling also existed in primary macrophages. To address the putative dimerization between EP2 and GPRC5B, we immunoprecipitated EP2 receptors from freshly isolated RPM using EP2-specific monoclonal antibodies and found that GPRC5B co-precipitated (Fig. 4A). We then determined EP2 responses in macrophages from control and M-G5b-KO mice and found that butaprost-stimulated cAMP production was present in control cells, but strongly diminished in GPRC5B-deficient RPM (Fig. 4B) and BMDM (Supplementary Fig. 7A). LPS-induced upregulation of *Nos2* or *Tnf* was reduced by butaprost in control RPM, but not in KO cells (Fig. 4C, D). Furthermore, butaprost inhibited phagocytosis of E. coli in control RPM, but not in GPRC5B-deficient RPM (Fig. 4E, F), and similar findings were obtained in butaprost-treated BMDM (Fig. 4G, Supplementary Fig. 7B). We also investigated the effect of endogenous EP2 agonist $PGE_2$ on phagocytic activity and found that the anti-phagocytic effect of $PGE_2$ was blocked both in RPM and BMDM from M-G5b-KOs (Fig. 4F, G, Supplementary Fig. 7C, D). To investigate whether GPRC5B modulates EP2 signaling also in human macrophages, we performed siRNA-mediated knockdown and transfection-based overexpression of GPRC5B in THP1 cells, a human monocytic cell line (Fig. 4H, Supplementary Fg. 7E). We found that knockdown of GPRC5B resulted in a significant reduction of butaprost-induced cAMP production (Fig. 4I), whereas overexpression of GPRC5B significantly enhanced responses to butaprost (Fig. 4I).

In the next step we investigated whether up- or downregulation of GPRC5B expression was also in a physiological context able to modulate EP2 signaling. GPRC5B was initially identified as a retinoic acid (RA)-induced gene[20], and we confirmed strong upregulation of *Gprc5b* expression in BMDM on RNA and protein level (Fig. 4J, Supplementary Fig. 8A, B). *Ptger2*/EP2 expression, in contrast, was downregulated by RA on RNA and protein level (Fig. 4K, Supplementary Fig. 8C, D). These findings prompted us to investigate whether enhanced GPRC5B-mediated facilitation of EP2 signaling would be able to functionally compensate reduced EP2 expression. We found that 6 h after RA treatment, a time point at which both *Ptger2* and *Gprc5b* levels were low, butaprost-induced cAMP production was almost completely suppressed, whereas at 24 h, a time point at which *Ptger2*/EP2 levels were still low but GPRC5B was upregulated, EP2 responses were partly restored (Fig. 4L).

We also investigated the effect of macrophage activation on GPRC5B expression and found that exposure to LPS for 24 h resulted in downregulation of *Gprc5b*/GPRC5B mRNAs in mouse BMDM and RPM as well as in THP1 cells and primary human blood CD11b-positive cells (Fig. 4M, N), and also on the protein level GPRC5B expression was reduced (Fig. 4O). In mouse BMDM and human CD11b-positive blood cells, this was associated with a downregulation of *Ptger2*/PTGER2 levels, whereas EP2 expression remained stable in RPM and THP1 (Fig. 4O, P; Supplementary Fig. 8E–G). We tested whether LPS-induced

downregulation of GPRC5B affected EP2 signaling in cells with stable EP2 expression and found that butaprost-induced cAMP production was clearly reduced in LPS-treated RPM (Fig. 5Q), and also THP1 showed reduced responses (Supplementary Fig. 8H). Taken together, these data suggest that transcriptional regulation of GPRC5B expression by metabolic or inflammatory cues is able to modulate EP2 signaling strength.

## Effect of GPRC5B on EP2 expression and membrane availability

We next investigated the mechanisms underlying GPRC5B-dependent facilitation of EP2 signaling. RNA and protein levels of *Ptger2*/EP2 did not differ between genotypes in RPM or M0 BMDM (Fig. 5A–C, Supplementary Fig. 9A), suggesting that GPRC5B does not affect receptor expression or stability. Since GPRC5B was previously shown to regulate the membrane availability of the prostacyclin receptor[26], we determined whether GPRC5B also influences intracellular trafficking of EP2. To do so, we analyzed membrane localization of HA-tagged EP2 receptor (HA-EP2) in HEK cells in the absence and presence of GPRC5B-FLAG/Myc. We found that in HA-EP2/GPRC5B co-expressing cells, HA signal intensity in the wheat germ agglutinin (WGA)-stained plasma membrane was significantly higher than in cells expressing only HA-EP2 (Fig. 5D, E). To test whether GPRC5B controlled EP2 localization also in primary macrophages, we determined surface EP2 expression in non-permeabilized BMDM and RPM using an antibody directed against extracellular loop 3 of the EP2 receptor (EP2(ec)). Using an ELISA type assay, we found that non-permeabilized macrophages from M-G5b-KOs showed reduced EP2 membrane staining compared to control cells (Fig. 5F, G). Butaprost application did not induce clear EP2 internalization, and the difference in membrane availability was maintained in the presence of ligand (Supplementary Fig. 9B). Also immunofluorescence microscopy showed significantly reduced EP2(ec) signals in the CD11b-stained plasma membrane of GPRC5B-deficient RPM (Fig. 5H, I). We then determined to which intracellular compartments EP2 translocated in the absence of GPRC5B. We found that reduced colocalization with CD11b (i.e., plasma membrane) coincided with increased colocalization both with Golgi marker GM130 and endoplasmic reticulum marker calreticulin (Fig. 5J, K; Supplementary Fig. 9C). These data suggest that loss of GPRC5B expression results in partial retention of EP2 in endoplasmic reticulum and Golgi apparatus.

## Identification of interacting residues

To determine the regions mediating the interaction between GPRC5B and EP2, we created a homology model of GPRC5B using the CryoEM structure of metabotropic glutamate receptor 2 as a template (for details, see methods). Similarly, homology models of the GPRC5B-interacting receptors EP1, DP1 and IP were created based on the structure of EP2. Since the related receptor GPRC5C shows similar interaction patterns as GPRC5B, we focused our protein docking studies on transmembrane regions conserved between GPRC5B and

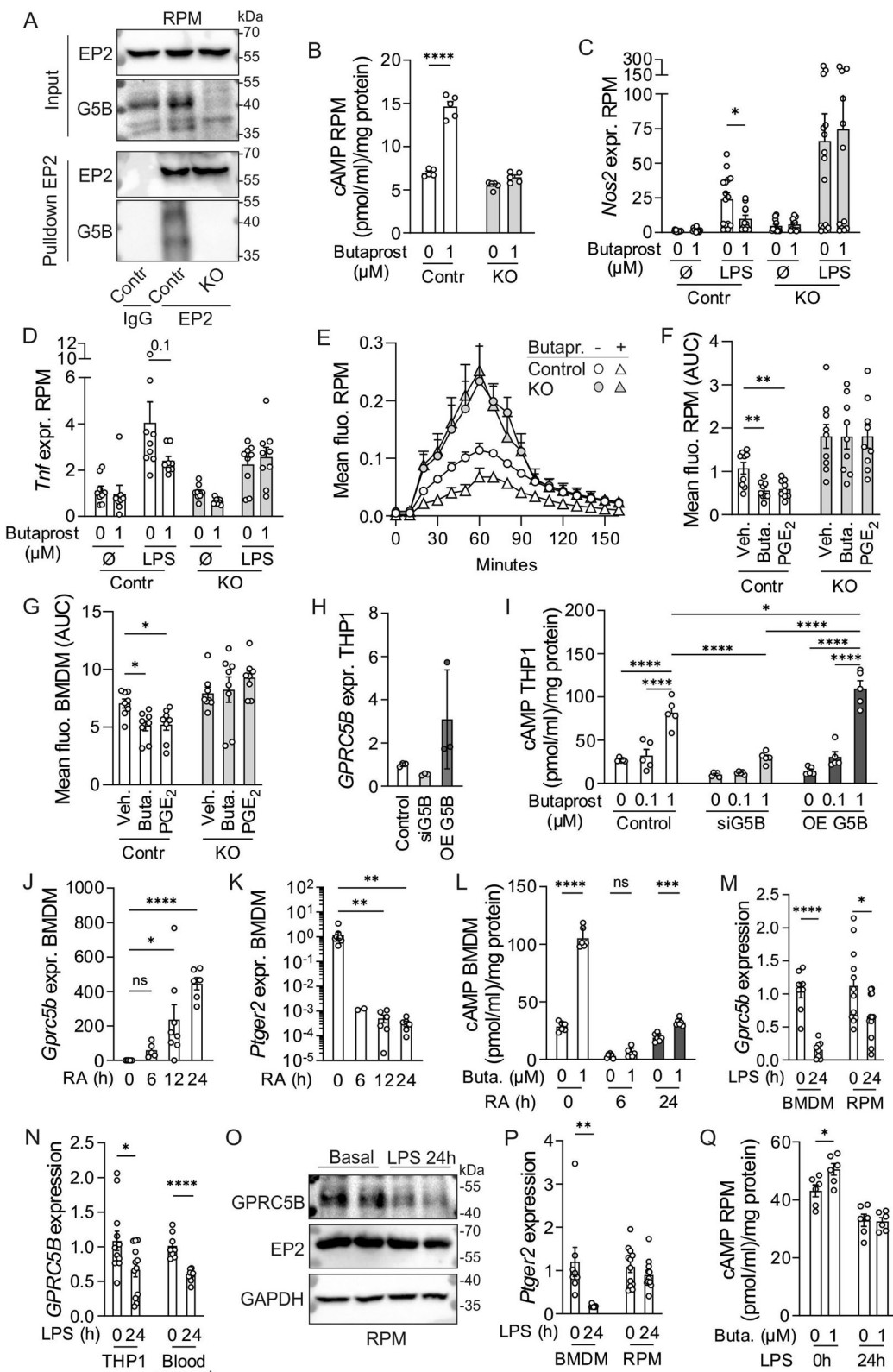

GPRC5C and identified the following residues in transmembrane region 2 as potential contributors to the interaction: F97, L101 and L104 ("set 1"), F111 ("set2"), and Q116, D118 and L128 ("set 3") (Fig. 6A, B). Based on these data we generated a GPRC5B version in which these 7 residues were mutated to alanine (G5B-mut7) and tested whether this resulted in loss of interaction. The expression pattern of G5B-mut7 in HEK cells was comparable to wild type GPRC5B (Fig. 6C),

but showed a significantly reduced ability to co-precipitate with HA-EP2 in HA-pulldown experiments (Fig. 6D, quantification in Supplementary Fig. 10A). Furthermore, the facilitating effect of GPRC5B on basal and butaprost-induced cAMP production was lost in G5B-mut7 (Fig. 6E). Mutation of these 7 residues did not only disrupt the GPRC5B/EP2 interaction for human sequences, but also when we used mouse sequences (Supplementary Fig. 10B–D). To narrow down the number

**Fig. 4 | Altered EP2 signaling in macrophages. A** Western blot detection of EP2 and GPRC5B (G5B) signals in lysates of RPM before ("input") and after immunoprecipitation of EP2 using anti-EP2 antibodies ("Pulldown EP2") (IgG and GPRC5B-KO as negative control). **B** Butaprost-induced cAMP production was determined in RPM by ELISA ($n = 5$). **C, D** The effect of butaprost on expression of *Nos2* (C) and *Tnf* (D) was determined by qRT-PCR in basal and LPS ($1\,\mu g/ml$, 6 h)-stimulated RPM ($n = 15/11/15/11/15/12/15/12$ in C, $9/8/9/8/9/9/9/9$ in D); data normalized to *Gapdh* and basal controls set to 1. **E–G** Effect of EP2 agonists on the uptake of pHrodo E. coli bioparticles in RPM (E, F) or M0 BMDM (G): E, exemplary curves ($n = 3$); F + G, statistical evaluation ($n = 9$(F), 8(G)). **H, I** Effect of GPRC5B knockdown (siG5B) or overexpression (OE G5B) on butaprost-induced cAMP production in human THP1 cells: efficiency of knockdown/overexpression (H, $n = 3$) and analysis of cAMP levels by ELISA (I, $n = 5$). All three groups were transfected both with plasmid and siRNAs (Control: EV+siControl; siG5B: EV + GPRC5B siRNA; OE G5B: GPRC5B plasmid + control siRNA). **J, K** Expression of *Gprc5b* (J) and *Ptger2* (K) was determined by qRT-PCR in M0 BMDM exposed for the indicated times to $1\,\mu M$ retinoic acid (RA) ($n = 8/5/8/6$ in J, $8/2/6/6$ in K). **L** Butaprost-induced cAMP production in BMDM that had been exposed to RA for the indicated times ($n = 6$). **M, N** Expression of *Gprc5b/*

*GPRC5B* in mouse macrophages (M) or human THP1 and CD11b-positive blood cells (N) was determined by qRT-PCR after 24 h exposure to vehicle or $1\,\mu g/ml$ LPS ($n = 8$(BMDM)/12(RPM) in M, 12(THP1)/8(CD11b-positive blood cell) in N; data normalized to *Gapdh/GAPDH* and basal set to 1). **O** Western blot detection of EP2 and GPRC5B in lysates of vehicle- or LPS-treated RPM (GAPDH as loading control). **P** Expression of *Ptger2* in vehicle- and LPS-treated mouse macrophages ($n = 8$(BMDM)/12(RPM)). **Q** Butaprost-induced cAMP production in RPM exposed to vehicle or LPS for 24 h ($n = 6$). Data are means ± SEM; comparisons between agonist- and respective vehicle-treated groups (B-D, F, G, J, K, L-Q) or all groups (I) was done using one-way ANOVA with Sidak's multiple comparisons test (B), two-sided Mann-Whitney test with Holm-Sidak's multiple comparisons test (C, D), one-way ANOVA with Dunnett's multiple comparison test (F, G), Kruskal-Wallis test with Dunn's multiple comparison test (H, J, K), two-way ANOVA with Tukey's (I) or Sidak's (L, Q) post hoc test, unpaired two-sided $t$ tests (M, N, P), two-sided Mann-Whitney test (O). EV, empty vector; n, number of independent experiments or mice per group; veh., vehicle (DMSO for butaprost and $PGE_2$; $H_2O$ for LPS); *$P < 0.05$; **$P < 0.01$; ***$P < 0.001$; ****$P < 0.0001$. Source data are provided as a Source Data file.

of residues involved in the interaction, we generated three partial mutants that selectively address the three sets of residues initially identified (Fig. 6B). We found that mutation of the lower pocket (set 1) recapitulated the phenotype of G5B-mut7 with respect to co-immunoprecipitation and cAMP production, whereas sets 2 and 3 did not or only partially do so (Fig. 6F, G; quantification for Fig. 6F in Supplementary Fig. 10E). We also investigated how mutations affected the ability of GPRC5B to enhance EP2 membrane localization, and found that this effect was significantly reduced for G5B-mut7 and G5B-mut_set1, but not for the other variants (Fig. 6H, I). These data indicate that the amino acids F97 A, L101A, L104A are crucial for the interaction of GPRC5B with EP2.

### Blocking GPRC5B-mediated facilitation of EP2 signaling using decoy peptides

We finally sought to establish pharmacological approaches to interfere with the GPRC5B/EP2 interaction. Since previous studies showed that GPCR dimerization can be interrupted by cell-permeable peptides mimicking the dimerization interface[35], we tested whether a peptide mimicking the region around amino acids F97/L101/L104 was able to interfere with GPRC5B-mediated facilitation of EP2 signaling. We designed a C-terminally amidated peptide consisting of the HIV-TAT cell-penetrating amino acid sequence (YGRKKRRQRRR)[35] fused to amino acids 95–106 of GPRC5B (LHFLFLLGTLGL) ("target peptide"). As control peptide, we fused a scrambled version of the target sequence (FLTLLLLLFGHG) to HIV-TAT. We found that in HEK cells expressing only EP2 receptor, neither control nor target peptide affected butaprost-induced cAMP production (Fig. 7A, left). In HEK cells coexpressing EP2 and GPRC5B, the expected facilitation of butaprost-induced cAMP production was observed, and this response was significantly reduced by the target peptide, but not by control peptide (Fig. 7A, right). Furthermore, we found that target peptide, but not control peptide, was able to reduce the amount of GPRC5B-FLAG/Myc that was co-immunoprecipitated with HA-EP2 in transfected HEK cells (Fig. 7B, C). Also in immunostainings, the colocalization of HA-EP2 with GPRC5B-FLAG/Myc was reduced by target peptide (Supplementary Fig. 11A, B), and target peptide was also able to reduce bioluminescence resonance energy transfer between GPRC5B-NLuc and EP2-Venus (Supplementary Fig. 11C). We next tested whether this approach was suitable for interfering with GPRC5B-mediated facilitation of EP2 signaling in primary macrophages. Both in RPM and BMDM, target peptide reduced butaprost-induced cAMP production in a dose dependent manner, while control peptide was without significant effect (Fig. 7D, E). With respect to the subcellular localization of EP2, we observed reduced availability at the membrane, and increased Golgi retention in the presence of target peptide (Supplementary

Fig. 11D–F). Finally, we investigated whether inhibition of the GPRC5B-EP2 interaction was able to reproduce the GPRC5B-KO phenotype of enhanced phagocytosis in BMDM. We found that target peptide, but not control peptide enhanced E.coli uptake in control BMDM, but not in KO BMDM (Fig. 7F–H). Taken together, these data show that a cell-permeable peptide mimicking the interaction interface between GPRC5B and EP2 is able to partially reproduce M-G5b-KO phenotypes such as reduction of EP2 signaling and consecutive macrophage hyperactivity.

### Discussion

To maintain the balance between efficient pathogen defense and maladaptive chronic inflammation, macrophage activation has to be tightly controlled. Understanding the signals that modulate macrophage activation is therefore crucial in harnessing their defensive abilities while preventing harmful immune responses. We show here that orphan receptor GPRC5B plays a crucial role in the regulation of macrophage activity and that its inactivation results in enhanced migration and phagocytosis. This hyperactivity does not result in basal changes in myeloid cell number or distribution, but allows improved clearance of bacteria in peritonitis. In myocardial infarction, however, increased myeloid cell recruitment is associated with increased scar size and deterioration of cardiac function. Such immune-related adverse effects are a frequent problem in therapies aiming at the enhancement of immune responses. Facilitating T cell responses by immune checkpoint inhibitors, for example, is associated with inflammatory responses in numerous organ systems including the heart[36]. Therapies specifically targeting macrophage activation are still in development[37], but it seems likely that the same caveats apply to them. It is in this context interesting to note that M-G5b-KOs did not show clear changes in their responses to DSS colitis or EBA, indicating that the relevance of GPRC5B-mediated regulation depends on many factors such as relative importance of macrophages in the respective model or the availability and strength of other pathways controlling their activity. Furthermore, the complex transcriptional regulation of GPRC5B expression by local factors such as LPS or RA will impact on the relevance of GPRC5B-mediated effects in the different models. Our expression studies indicate that RPM might be more sensitive to GPRC5B-mediated activity regulation than other cell types, both because of their high basal GPRC5B expression and the lack of down-regulation in responses to activation.

As to the mechanism underlying altered macrophage activity in M-G5B-KOs, we show that GPRC5B regulates the signaling strength of the $G_s$-coupled $PGE_2$ receptor EP2. The EP2 receptor is widely expressed and regulates numerous physiological functions including inflammation, pain sensation, vascular tone regulation,

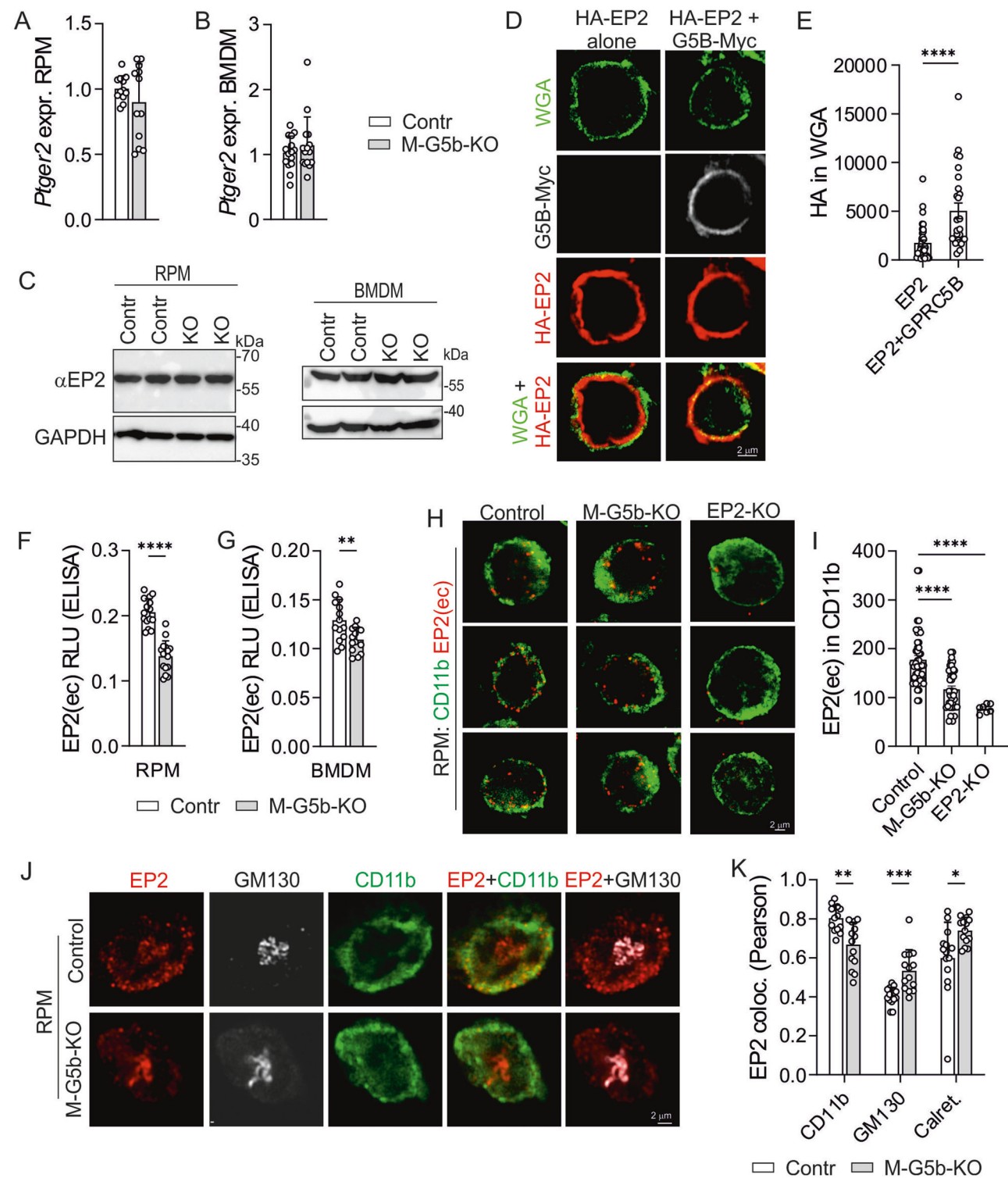

neuroprotection and neuroplasticity[38]. The role of EP2 in inflammation is complex and the receptor has been implicated both in pro- and anti-inflammatory processes[38]. In macrophages, however, EP2-mediated effects are largely anti-inflammatory and include the inhibition of cytokine production and phagocytosis[13,15]. Reduction of these anti-inflammatory effects is well suited to explain the observed hyper-activity, though we can at this point not elude that other factors contribute. The previously suggested regulation of sphingomyelin synthase 2 by GPRC5B[23], however, seems not to play a central role in macrophages, since we did not detect clear difference in the levels of sphingoid bases or ceramides.

Regarding the mechanistic details of GPRC5B-mediated EP2 regulation, our co-immunoprecipitation studies suggest a direct interaction between these two receptors. GPCR dimerization, i.e., the association of two GPCR molecules to form a complex, is a significant aspect of GPCR biology that impacts receptor signaling and trafficking[39,40]. A number of studies suggested that the PGE₂ receptors can form homodimers or heterodimers. In airway SMC, for example, bioluminescence resonant energy transfer showed that EP1 and β2 adrenergic receptors form heterodimers, and ligand binding studies suggested that the EP1 component of the dimer uncoupled β2-adrenergic receptor from Gα_s, thereby reducing the bronchodilatory

**Fig. 5 | Effect of GPRC5B on EP2 expression and membrane availability. A, B** *Ptger2* expression in RPM (A, $n = 12$)) and M0 BMDM (B, $n = 15$) was determined by qRT-PCR (normalization to *Gapdh*, control set to 1). **C** EP2 expression was determined by Western blotting in RPM and M0 BMDM lysates from control mice (Contr) and M-G5b-KOs (KO). GAPDH as loading control. **D, E** HEK cells co-transfected with HA-EP2 alone or HA-EP2 + GPRC5B-FLAG/Myc (G5B-Myc) were first subjected to plasma membrane staining with WGA, then fixed, permeabilized and stained with anti-Myc and anti-HA antibodies: exemplary photomicrographs (D) and statistical evaluation of the HA-EP2 signal within WGA (E) ($n = 26$ and 36 cells). **F, G** ELISA-based detection of EP2(ec) in the plasma membrane of RPM (F) and M0 BMDM (G) ($n = 14$(F), 16(G) samples from 3-4 mice per group). **H, I** Antibody-mediated detection of EP2(ec) in non-permeabilized RPM from control mice and M-G5b-KOs (CD11b staining indicates plasma membrane, global EP2-KOs shown as specificity control): exemplary photomicrographs (H) and statistical evaluation of EP2(ec) signal in CD11b area (I) ($n = 44/37/7$ cells from 3 mice per group). **J, K** RPM from control and KO mice were fixed, permeabilized and stained for EP2(ec) and Golgi marker GM130, and then stained with anti-CD11b antibodies (for plasma membrane): exemplary photomicrographs (J) and statistical evaluation (K) of the colocalization of EP2(ec) with CD11b, GM130 or calreticulin (examples for the latter in Supplementary Fig. 9C) ($n = 14$ cells from 2 mice each). Data are means ± SEM; differences between groups were analyzed using two-sided Mann-Whitney test (A, B, E), unpaired two-sided *t*-test (F, G), Kruskal-Wallis test with Dunn's correction for multiple testing (I), unpaired two-sided *t* test with Holm-Sidak correction for multiple testing (K). *n*, number of analyzed mice (A, B, F, G) or cells (E, I, K); *$P < 0.05$; **$P < 0.01$; ***$P < 0.001$; ****$P < 0.0001$. Source data are provided as a Source Data file.

effects of β2 receptor agonists[41]. The EP2 receptor has been suggested to heterodimerize with the calcitonin receptor CTR in an ovarian granulosa cell line, thereby reducing CTR-mediated calcium mobilization[42]. The molecular determinants mediating these interactions, as well as their physiological relevance, are unknown. Also GPRC5B has previously been reported to form dimers: In vascular SMC, GPRC5B dimerizes with the prostanoid IP receptor, thereby regulating SMC contractility and differentiation[26]. GPRC5B is a class C orphan GPCR, and other class C members are known to function as stable dimers: metabotropic glutamate receptors and the calcium-sensing receptor form homodimers, whereas $GABA_B$ and sweet and umami taste receptors are obligatory heterodimers[39,40]. Typical features mediating class C dimerization are the N-terminal Venus flytrap motif (VFTM) or the cysteine-rich domain between VFTM and heptahelical domain[39], but GPRC5B lacks both VFTM and full cysteine-rich domain. Our data show that GPRC5B is nevertheless able to undergo dimerization with certain prostanoid receptors, and we identify three residues in transmembrane region 2 of GPRC5B that are essential for the interaction with EP2: F97, L101, and L104. Mutation of these residues results in strongly reduced interaction with EP2 in co-immunoprecipitation, and a near complete loss of GPRC5B-mediated facilitation of EP2 signals in HEK cells and macrophages. Interestingly, mutation of amino acids Q116, D118, L128 (set 3) reduced GPRC5B-mediated facilitation of EP2 signaling without breaking the physical interaction and without altering EP2 membrane localization, suggesting that GPRC5B might influence EP2 signaling through additional, yet unknown mechanisms.

To explore the therapeutic potential of GPRC5B-mediated regulation of EP2 signaling, we investigated pharmacological strategies to interfere with the dimerization. Protein-protein interactions are notoriously difficult to target, but peptide inhibitors mimicking the interface of two interacting GPCRs have been successfully used to interfere with homo- and heterodimerization[43]. For example, both the β2-adrenergic receptor and chemokine receptor CXR4 have been suggested to signal more efficiently as homodimers, and peptide inhibitors of interacting transmembrane regions were shown to impair both interaction and signaling[44,45]. Furthermore, cell-penetrating peptides were used to block the formation of GPCR heterodimers, for example between cannabinoid receptor CB1 and serotonin receptor 5-HT2A[35], or between AT1 angiotensin II receptor and the secretin receptor[46]. We show here that a cell-permeable peptide mimicking the amino acids 95-106 of GPRC5B is able to reduce the facilitating effect of GPRC5B on EP2 signaling, while a scrambled version of this peptide does not have this effect. Importantly, this is not only true in HEK cells, but also in primary macrophages, where decoy peptides were able to mimic the hyperphagocytosis phenotype of GPRC5B-deficient cells. These findings indicate that pharmacological interference with GPRC5B/EP2 dimerization is possible and may be exploited to modulate macrophage activation in vivo.

Regarding the physiological relevance of the GPRC5B/EP2 interaction, it is interesting to consider the mechanisms by which EP2 signaling is regulated: ligand availability, coupling efficiency, membrane availability, and desensitization[38,47]. Under inflammatory conditions, $PGE_2$ synthesis is strongly increased[13], but in the acute phase of pathogen defense, EP2/EP4-mediated anti-inflammatory effects in macrophages might be counterproductive. The EP4 receptor is quickly desensitized upon activation via GRK/β-arrestin–mediated uncoupling, internalization, and eventual down-regulation, and this effect relies on regulatory sites on the C-terminus of the EP4 receptor[47,48]. EP2 receptors, in contrast, have a short C-terminal tail, lack the residues for phosphorylation by G protein-coupled receptor kinases, and thus do not internalize[47,48]. Therefore, while EP4 receptors are quickly desensitizing, additional levels of regulation are needed to control EP2 signaling, and we hypothesize that dimerization-dependent regulation of membrane availability is one such mechanism. It is in this context interesting to note that GPRC5B expression is downregulated both in mouse and human BMDM by LPS. LPS is a central trigger of macrophage activation, and it is tempting to speculate that reduction of GPRC5B-mediated facilitation of anti-inflammatory EP2 signaling contributes to efficient macrophage activation. However, it should be noted that in other cell types LPS treatment resulted in enhanced GPRC5B expression, for example in fibroblasts[25]. The reasons for this discrepancy are currently unclear and may be related to cell type-specific transcriptional or epigenetic regulation processes. Equally interesting is the upregulation of GPRC5B by vitamin A1 metabolite RA, which has been implicated in numerous processes related to growth and differentiation[20]. Using RA-stimulated BMDM, we were able to show that GPRC5B-mediated facilitation compensates downregulation of EP2, further supporting the notion that regulation of GPRC5B by local mediators fine-tunes EP2 signaling. Interestingly, RA plays a pivotal role in development of RPM, because it drives the expression of GATA6, a lineage-determining transcription factors in these cells[49]. RA is produced by various cell populations in the peritoneal cavity[49], and this local production of RA might be the reason that GPRC5B expression is significantly higher in RPM than in other myeloid populations.

Taken together, we show here that orphan GPCR GPRC5B dimerizes in RPM and BMDM with the $G_s$-coupled EP2 receptor, thereby enhancing EP2 membrane availability and signaling efficiency. In the absence of GPRC5B, anti-inflammatory EP2 signaling is reduced, resulting in enhanced macrophage migration and phagocytosis. It is possible that also other mechanisms contribute to the observed phenotypes, but the fact that decoy peptides targeting the GPRC5B/EP2 interface are able to reproduce phenotypes of GPRC5B-deficient macrophages strongly supports the functional relevance of this dimerization.

## Methods
### Ethical approval
All animal experiments were approved by the Institutional Animal Care and Use Committee of the *Regierungspräsidium* Darmstadt and in

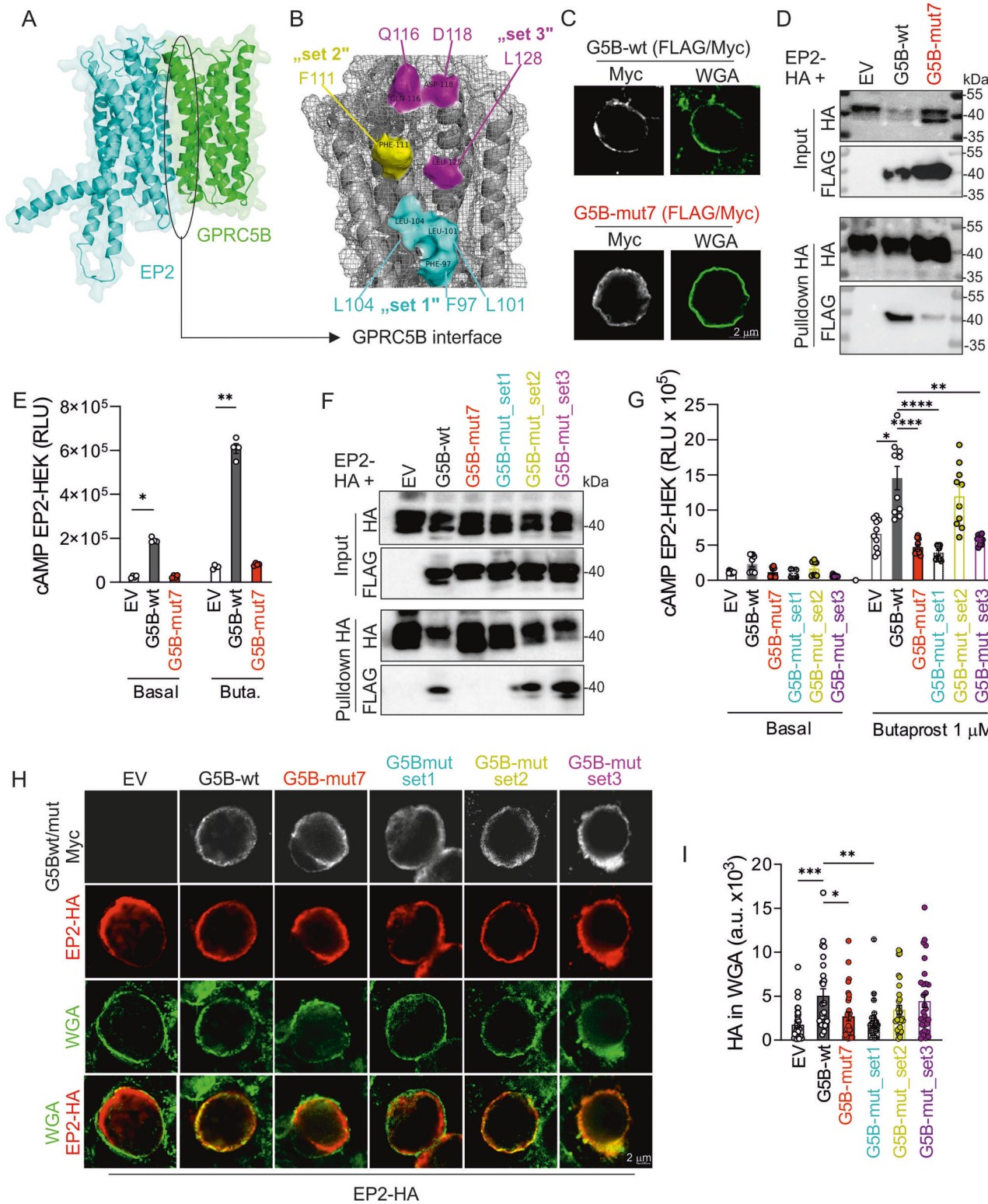

accord with Directive 2010/63/EU of the European Parliament on the protection of animals used for scientific purposes. Analyses were performed by investigators blinded to genotype and treatment.

**Reagents**
The following reagents were used: Lipopolysaccharide (LPS, L4391) and N-Formyl-Met-Leu-Phe (fMLP, F3506) from Sigma-Aldrich, mouse macrophage colony-stimulation factor (M-CSF, 315-02), mouse IFN-γ (315-05), mouse CCL5 (250-07), mouse CCL2 (250-10), mouse SDF-1β

(250-20B) from PeproTech. PGE$_2$ (2296), 3-Isobutyl-1-methylxanthine (IBMX, 2845), and iloprost (2038) were from Bio-Techne GmbH. Luciferin D (122799) was purchased from PerkinElmer. Coelenterazine h (S2001) was from Promega GmbH. Butaprost (13741), L-902668 (10007712), 17-phenly trinor PGE$_2$ (17-pt-PGE$_2$, 14810), leukotriene B4 (LTB4, 20110), PGE$_2$ for FRET experiments (14010) were from Cayman chemical company. Complement component C5a (2150-C5-025) was from R&D systems. CXCL1 (573704) and CXCL3 (590802) were from Biolegend.

**Fig. 6 | Interfering with the GPRC5B/EP2 interaction: in silico docking and mutation of GPRC5B. A** The refined predicted orientation of the GPRC5B-EP2 complex (GPRC5B in green, EP2 in cyan). **B** Three sets of residues within GPRC5B considered for site-directed mutagenesis. **C** Exemplary photomicrographs showing cellular distribution of FLAG/Myc-tagged wild type GRC5B (G5B-wt, top) and GPRC5B carrying 7 alanine mutations (G5B-mut7, bottom) in permeabilized HEK cells (WGA as membrane staining). **D** Western blot detection of HA and FLAG signals in lysates of HEK cells expressing HA-tagged EP2 in combination with empty vector (EV), FLAG-tagged human G5B-wt or G5B-mut7: "input" shows lysates before, "pulldown HA" after immunoprecipitation with anti-HA beads. **E**, Butaprost-induced cAMP production in HEK cells transfected with cAMP GloSensor plasmid, HA-EP2, and wild type or mutant GPRC5B as indicated ($n = 4$). **F** Western blot detection of HA and FLAG signals in lysates of HEK cells expressing HA-tagged EP2 in combination with empty vector (EV) or FLAG-tagged human wild type or mutant GPRC5B as indicated. **G**, Butaprost-induced cAMP production in HEK cells transfected with cAMP GloSensor plasmid, HA-EP2, and wild type or mutant GPRC5B as indicated ($n = 10$). **H, I** HEK cells were transfected with HA-EP2 and wild type or mutant GPRC5B as indicated and subjected to plasma membrane staining with WGA, followed by permeabilization and staining with anti-HA antibodies: exemplary photomicrographs (H) and statistical evaluation of the HA-EP2 signal within WGA (I) ($n = 36/26/31/37/35/30$ cells). Data are means ± SEM; comparisons with empty vector-transfected samples (E) or G5B-wt-transfected samples (G, I) were performed using Kruskal-Wallis test with Dunn's correction for multiple testing. EV, empty vector; $n$, number of independent experiments; *$P < 0.05$; **$P < 0.01$; ***$P < 0.001$; ****$P < 0.0001$. Source data are provided as a Source Data file.

Decoy peptides were synthesized at GenScript and consisted of the N-terminal HIV-TAT sequence YGRKKRRQRRR (for cell permeability), followed by target peptide LHFLFLLGTLGL or its scrambled version FLTLLLLLFGHG. Both peptides were c-terminally amidated to increase stability.

## Experimental Animals

*Gprc5b*[fl/fl] mice[26] were intercrossed with *LysM*-Cre mice (B6.129P2-*Lyz2*[tm1(cre)lfo]/J)[30] to generate myeloid cell-specific GPRC5B-deficient mice (M-G5b-KO). Mice were maintained on a C57BL/6 J background and if not otherwise indicated, genetically matched Cre-negative *Gprc5b*[fl/fl] mice were used as controls. Mice were kept in individually ventilated microisolator cages with 12-hour light-dark cycles in a specific pathogen-free facility with a temperatures range between 20–24 °C and humidity between 45–65%. Standard chow (Altromin GmbH Lage, Altromin 1320), tap water, and nesting material were provided ad libitum. Genotyping for *Gprc5b* was done with the primers 5'gctggaaggtttctccctct-3' and 5'aagagacaaccaccagacagg-3', resulting in band sizes of 361 bp for the wild-type allele and 478 bp for the floxed allele. Genotyping for *LysM*-Cre was done with primers 5'cttgggctgccagaatttctc3' and 5'cccagaaatgaattacg3', resulting in a band size of 834 bp. Mice were analysed at an age of 6–16 weeks and both female and male mice were used, with the exception of the MI model, in which only males were used. Mice were euthanized in $CO_2$ anaesthesia.

## Animal models

Bacterial peritonitis was induced by intraperitoneal injection of fecal suspensions as described previously in ref. [50]. Briefly, fresh stool was collected and dissolved in Dulbecco's Phosphate-Buffered Saline (PBS, Gibco, 14040141) at a concentration of 30 mg/ml. The stool suspension was filtered through a 70 μm cell strainer and 100 μl of the resulting suspension were injected intraperitoneally into mice. Thereafter, mice were observed and scored every 1.5 h up to 24 h. Body weight was measured every 6 h up to 24 h. 24 h after of peritonitis induction, mice were sacrificed under $CO_2$ and peritoneal lavage was performed using 7 ml of PBS containing 0.5% BSA (Serva, 11930) and 2 mM EDTA (Roth, 8043). Lavage fluid was used for FACS analysis of leukocyte populations or determination of bacterial clearance. For the latter, lavage fluid was serially diluted and plated on LB agar plates. After 16 h of incubation, numbers of colony-forming units were determined.

For induction of myocardial infarction, coronary artery ligation was performed in male mice 10 to 14 weeks old. Briefly, following anesthetization (isoflurane inhalation) and tracheal intubation, the chest cavity was opened from the 4th intercostal space. The left ascending coronary artery was ligated tightly with a 8-0 suture under microscopy. Myocardial ischemia was confirmed by color changes in the segment of the left ventricle subjected to coronary flow occlusion. The wound was closed with a 5-0 suture. After surgery, mice were monitored every day for 5 days, and treated with an intraperitoneal injection of buprenorphine (0.1 mg/kg) twice a day and metamizole in drinking water (200 mg/kg). Surgery was performed by an individual who was blinded to the mouse genotype.

Cardiac function was assessed by echocardiography before and after surgery at indicated time points using the Vevo 2100 ultrasound system (VisualSonics, Fujifilm) equipped with a MS550d transducer. Left ventricular ejection fraction (LVEF) and LV systolic and diastolic volumes were measured by Simpson's method.

For determination of immune cell infiltration on d4 after surgery, hearts were harvested after PBS perfusion and remote zone as well as infarct + border zone were dissected. The respective regions were cut into 1 mm pieces, digested with 0.02% collagenase II (Worthington, LS0004174) and 0.05% Dispase II (Sigma-Aldrich, 4942078001) at 37 °C for 1 h, and filtered using a 40 μm cell strainer. The suspension was subjected to FACS analysis after erythrocytes depletion using red blood cell lysis buffer (Biolegend, 420301) for 3 min at room temperature. To determine cell infiltration into healthy hearts, tissue pieces of approximately the same size as in the infarcted hearts were subjected to the same procedure.

For the histological analysis of scar size, hearts were harvested on day 21. After perfusion with PBS, tissues were fixed in 4% PFA overnight at 4 °C, embedded in paraffin, cut at 5-μm thickness, and stained with Sirius red-saturated picric acid buffer for 90 min, followed by image analysis. Images were obtained under a light microscopy. Finally, the scar size was measured using Fiji software (NIH).

DSS-induced colitis was induced by oral administration of 3% (w/v) dextran sodium sulfate (DSS, MW 36-50 kDa, MP biomedicals, 02160110) for 5 days as described previously in ref. [51]. After 1 day of normal drinking water, mice were sacrificed using $CO_2$ and tissues were harvested for further analysis. During the DSS treatment, a disease activity index (DAI) score was assessed based on body weight, stool consistency, and rectal bleeding[51]. Body weight loss was determined as follows: score 0, no weight loss compared to initial weight; 1, weight loss within 1–5%; 2, weight loss within 5–10%; 3, weight loss within 10–20%; 4, greater than 20% weight loss. Stool consistency was scored as follows: score 0, normal (solid pellet); 1, soft but in pellet shape; 2, loose stool but with some solidity; 3, loose stool with signs of liquid consistency; 4, watery diarrhea. Rectal bleeding was determined as follows: score 0, no sign of blood; score 1, no bleeding; 2, slight bleeding; 3, bloody diarrhea; 4, gross bleeding. Mice were sacrificed under $CO_2$ and colons were collected. The obtained colon was cut longitudinally and washed with ice-cold PBS and incubated with 5 ml of Hank's Balanced Salt Solution (HBSS, Thermo Fischer Scientific, 14175095) containing 5 mM EDTA at 37 °C for 20 min. Then, colon was washed three times with ice-cold HBSS and incubated with 5 ml of HBSS containing 2.5 mg of collagenase D (Roche, 11088858001), 2.5 mg of DNaseI (Sigma-Aldrich, 11284932001), and 15 mg of Dispase II (Sigma-Aldrich, 4942078001) 37 °C for 1 h. The digested colon was filtered using 100 μm cell strainer. The cells were pelleted by centrifugation ($450 \times g$ for 10 min at 4 °C) and resuspended in 1 ml of PBS containing 0.5% BSA and 2 mM EDTA. The cells were used for FACS analysis.

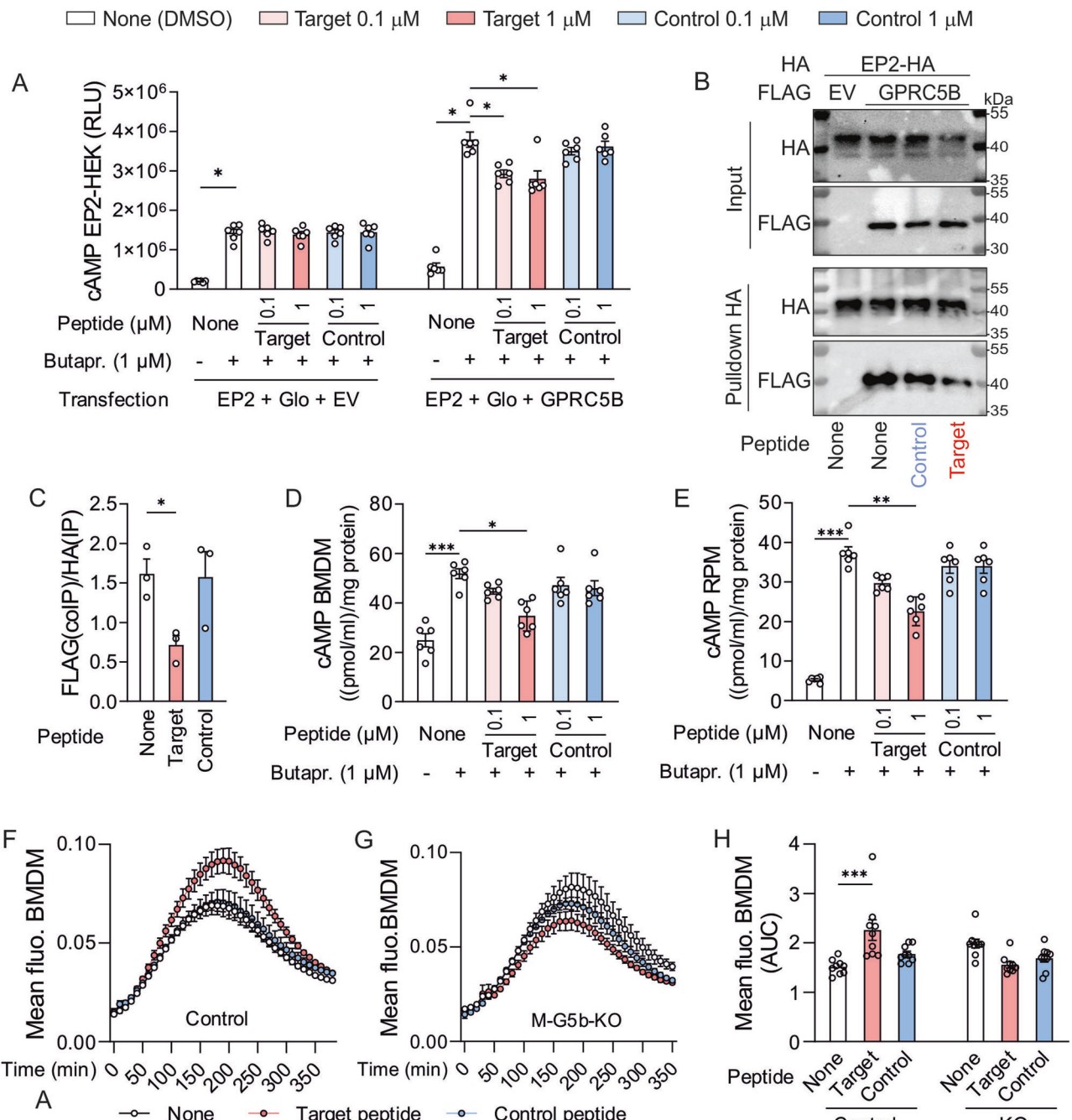

**Fig. 7 | Blocking GPRC5B-mediated facilitation of EP2 signaling using a decoy peptide. A** Effect of target and control peptides (0.1 or 1 μM each) on butaprost-induced cAMP production in HEK cells transfected with cAMP GloSensor plasmid, HA-EP2, empty vector (EV) or GPRC5B-FLAG/Myc as indicated ($n = 6$). **B, C** Effect of target and control peptides (10 μM) on co-immunoprecipitation of GPRC5B-FLAG/Myc with HA-EP2 in HEK cells: exemplary blots (B) and densitometric analysis of signal strength for co-precipitated GPRC5B-FLAG (FLAG(coIP)) relative to immunoprecipitated HA-EP2 (HA(IP)) ($n = 3$). **D, E** Peptide effect (0.1 or 1 μM each) on butaprost-induced cAMP production was determined in M0 BMDM (D) and RPM (E)

($n = 6$). **F–H**, Peptide effect (1 μM each) on phagocytosis of pHrodo E. coli bioparticles in M0 BMDM: Exemplary traces from control (F) and KO (G) mice; H: statistical evaluation of AUC ($n = 9$). Note that basal difference between control and KO is less pronounced in the presence of vehicle. Preincubation with peptides was 1 h in all cases. Data are means ± SEM; differences between peptide-treated and untreated groups were analyzed using Kruskal-Wallis test with Dunn's multiple comparisons test (A, D, E, H) or one-way ANOVA with Dunnett's correction for multiple testing (C). n, number of independent experiments; *$P < 0.05$; **$P < 0.01$; ***$P < 0.001$. Source data are provided as a Source Data file.

Bullous pemphigoid-like epidermolysis bullosa acquisita (EBA) was induced as described previously in ref. 52. Briefly, 50 μg of anti-mouse COL7 (kindly provided by S. Murthy, Lübeck, Germany) in 100 μl of PBS was injected subcutaneously on days 0, 2, and 4. The percentage of the total body surface presenting with erythema, blisters, erosions, crusts, or alopecia was determined every 3 days for up

to 16 days. Mice were sacrificed under $CO_2$, and perilesional skin was collected using a 6 mm biopsy punch (Kai medical, 48601). The obtained skin was washed with ice-cold HBSS and incubated with 1 ml of Dispase II (50 mg/ml) in HBSS at 37 °C for 1 h and then digested with 1 ml of Dulbecco's modified Eagle's medium (DMEM, Gibco, 10938) containing 100 μg of DNase I and 1 mg of Collagenase D 37 °C for 1 h.

The digested skin was filtered using 40 μm cell strainer. The cells were pelleted by centrifuge (400 × g for 5 min at 4 °C) and resuspended in 100 μl of PBS containing 0.5% BSA and 2 mM EDTA. The cells were used for FACS analysis.

Thioglycolate-induced peritonitis was induced by intraperitoneal injection of 1 ml of a sterile 4% (w/v) thioglycolate (Millipore, B2551). 12 h after of peritonitis induction, mice were sacrificed under $CO_2$ and peritoneal cavity cells were collected by washing the peritoneal cavity with 7 ml of PBS containing 0.5% BSA and 2 mM EDTA. Cells were used for FACS analysis of leukocyte populations.

## Isolation of primary leukocytes

Resident peritoneal macrophages (RPM) were isolated from the peritoneal cavities of mice without any stimulation. Peritoneal lavage was performed using 7 ml of ice-cold PBS with 0.5% BSA and 2 mM EDTA. Resident peritoneal macrophages were isolated from peritoneal lavage fluid using the mouse macrophage isolation Kit (peritoneum) (Miltenyi, 130-110-434) according to the manufacturer's protocol.

Bone marrow derived macrophages (BMDM) were isolated and differentiated into M1 macrophages as described previously (Zhang 2008)[53]. Briefly, bone marrow was isolated from the hindlimb of mice by flushing out bones with DMEM/F-12 nutrient mixture (Gibco, 11320) using a 26 G needle. $4 \times 10^6$ bone marrow cells were cultured in DMEM/F-12 containing 10% (v/v) FCS, 2 mL L-glutamine, 5 units/ml penicillin and streptomycin, and 100 units M-CSF in 10 cm dish. On day 4, another 5 ml of M-SCF containing medium was added. On day 7, M0 macrophages were used for qRT-PCR, cAMP assays or phagocytosis. For transwell migration, M0 macrophages were differentiated into M1 macrophages. To do so, cells were treated with 150 units/ml of INF-γ for 6 h and then 20 ng/ml of LPS for 16 h.

Bone marrow derived neutrophils were isolated from mouse bone marrow as previously described[54], with some modifications. Briefly, bone marrow was flushed from hindlimbs using Roswell Park Memorial Institute (RPMI) 1640 medium (Gibco, 21875). After centrifugation (400 × g for 3 min at room temperature), bone marrow cells were resuspended with 500 μl of PBS containing 20 mM EDTA and 0.5% BSA and then incubated with 7 μl of Ly6G-APC antibody (Biolegend, 127613) per $10^7$ total cells at 4 °C for 10 min. Ly6G-APC positive cells were isolated with anti-APC microbeads (Miltenyi, 130-090-855) according to the manufacture's protocol.

Experiments with human samples were performed according to the regulations of the local ethics committee of the Hessian Regional Medical Board (Ethikkommission des Fachbereiches Medizin der Goethe-Universität Frankfurt; AZ 110/11), and informed consent was obtained from all participants. CD11b-positive cells from human blood were isolated from human peripheral blood as previously described[55], with some modifications. Human blood samples were collected from 3 healthy donors (2 males, 1 female; age 30–40 years). The peripheral blood mononuclear cells were obtained using density gradient centrifugation on a Ficoll-Paque premium gradient (Cytiva, 17-5446-02). Centrifugation was performed at 800 × g for 15 min at 20 °C. After isolation, cells were washed with PBS containing 2 mM EDTA. Cells were suspended with 500 μl of PBS containing 20 mM EDTA and 0.5% BSA and were incubated with 7 μl of CD11b-APC antibody (BD Biosciences, 553312) per $10^7$ total cells at 4 °C for 10 min. CD11b-APC positive cells were isolated with anti-APC microbeads according to the manufacture's protocol.

## Isolation of mouse spleen cell

Spleen was washed with cold PBS and mechanically mashed through a 70 μM cell strainer with cold PBS containing 2 mM EDTA and 0.5% BSA. After centrifugation (500 × g for 5 min at 4 °C), the cell pellet was resuspended in 2 ml of red blood cell lysis buffer and incubated for 10 min at 4 °C. Following dilution in cold PBS containing 2 mM EDTA and 0.5% BSA, centrifugation at 500 × g for 5 min at 4 °C, and resuspension in PBS containing 2 mM EDTA and 0.5% BSA, cells were subjected to FACS analysis.

## Preparation of mouse blood cell suspension

Mouse blood obtained via cardiac puncture was collected in an EDTA coated tube (SARSTEDT, 16.444). 100 μl of blood was incubated with 5 ml of red blood cell lysis buffer for 10 min at room temperature. Cells were washed two times with cold PBS containing 2 mM EDTA and 0.5% BSA and were pelleted at 500 × g for 5 min 4 °C. Cells were resuspended in PBS containing 2 mM EDTA and 0.5% BSA and were used for FACS analysis.

## Cell culture

Human embryonic kidney cells (HEK-293T) cells were obtained from American Type Culture Collection (CRL-3216). Cells were maintained in DMEM supplemented with 10% (v/v) FCS (Gibco, 10270106), 2 mM L-glutamine (Thermo Fisher Scientific, 25030123), 5 units/ml penicillin/streptomycin (Gibco, 15140122). Cells were incubated at 37 °C in a humidified atmosphere with 5% $CO_2$.

Human leukemia monocytic THP-1 cells were obtained from American Type Culture Collection (TIB-202). Cells were maintained in RPMI 1640 medium with 10% (v/v) FCS, 2 mM L-glutamine, 5 units/ml penicillin/streptomycin, and 0.05 mM β-mercaptoethanol (Gibco, 31350010). Cells were incubated at 37 °C in a humidified atmosphere with 5% $CO_2$.

## Transfection

For overexpression experiments, HEK-293T cells were transfected with expression vectors listed in Table 1 using Opti-MEM (Gibco, 31985062) and Lipofectamine 2000 transfection reagent (Invitrogen, 11668019).

For siRNA-mediated knockdown experiments, HEK-293T were transfected with 28.5 nM of siRNA directed against human GPRC5B (Sigma-Aldrich, SASI_Hs01_00171699, NM_016235, target sequence: 5'-CGUUUAGAAGCAACGUGUA-3') was done using Opti-MEM and Lipofectamine RNAiMAX (Invitrogen, 13778075) according to the manufacturer's instructions. siRNA Universal Negative Control #1 (Sigma-Aldrich, SIC001) was used as a control.

For combined overexpression and knockdown in HEK cells, HEK-293T were transfected with 28.5 nM of siRNA directed against target gene using Opti-MEM and Lipofectamine RNAiMAX according to the manufacturer's instructions. After 20–24 h, cells were transfected with expressing vectors using Opti-MEM and Lipofectamine 2000 transfection reagent. After 48 h, cells were used for experiments.

Overexpression and knockdown experiments in THP-1 cells were conducted using the SG Cell Line 4D Nucleofector kit (Lonza, V4XC-3024) according to the manufacturer's instructions. The plasmid encoding human GPRC5B and siRNA against human GPRC5B were the same as in HEK cell experiments. Cells were first transfected with 30 nM of siRNA (siControl or siGPRC5B), then after 24 h re-transfected with empty vector or GPRC5B expression vector as indicated. Cells were used 48 h after transfection.

## qRT-PCR

RNA was extracted using Quick RNA micro kit (Zymo, R1050) and complementary DNA was synthesized by reverse transcription of extracted RNA using and ProtoScriptII (New England Biolabs, M0368) according to the manufacturer's instruction. Quantitative real-time PCR was performed using SYBR green PCR mix (Applied Biosystems, 4368708) using a Light Cycler 480 II (Roche) and QuantStudio 1 (Applied Biosystems). The relative expression levels of each gene were calculated using ΔΔ Ct method and normalized to reference gene *Gapdh/GAPDH*. If not otherwise indicated, the average of the respective control/basal condition is set to 1. For the primer sequences, see Table 2.

**Table 1 | Expression vectors ("N" for N-terminal tags; "C" for C-terminal tags)**

| Gene | Tag | Catalog # | Source | GeneID |
|---|---|---|---|---|
| **Human** | | | | |
| **Gene** | **Tag** | **Catalog #** | **Source** | **GeneID** |
| PTGIR | N-3xHA | PTGIR0TN00 | cDNA.org | AY242134 |
| PTGER1 | N-3xHA | | Self-made | AY275470 |
| PGTER2 | N-3xHA | PER020TN00 | cDNA.org | AY275471 |
| PTGER3 | N-3xHA | | VectorBuilder | NM_000957.2 |
| PTGER4 | N-3xHA | | Self-made | AY29109 |
| PTGDR1 | N-3xHA | | Self-made | NM_000953 |
| PTGDR2 | N-3xHA | | Self-made | AY507142 |
| TBXA2R | N-3xHA | | Self-made | AY429110 |
| PTGFR | N-3xHA | PTGFR0TN00 | cDNA.org | AY337000 |
| FPR1 | N-HA | 62603 | Addgene | M60627 |
| CCR5 | N-HA | 98950 | Addgene | NM_000579 |
| GPRC5B | C-NLuc | | VectorBuilder | NM_016235 |
| PTGER2 | C-Venus | | VectorBuilder | NM_000956.4 |
| PTGER2 | C-mTurq2 | | Self-made | NM_000956.4 |
| GPRC5B | C-mCitrine | | Self-made | NM_016235 |
| ADRB2 | C-mCitrine | | Self-made | NM_000024.6 |
| GLP1R | C-mCitrine | | Self-made | NM_002062.5 |
| P2RY12 | C-mCitrine | | Self-made | NM_022788.5 |
| GPRC5B | C-Myc-FLAG | RC205201 | Origene | NM_016235 |
| GPRC5B-mut7 | C-Myc-FLAG | | VectorBuilder | NM_016235 |
| GPRC5B-mutset1 | C-Myc-FLAG | | VectorBuilder | NM_016235 |
| GPRC5B-mutset2 | C-Myc-FLAG | | VectorBuilder | NM_016235 |
| GPRC5B-mutset3 | C-Myc-FLAG | | VectorBuilder | NM_016235 |
| **Mouse** | | | | |
| **Gene** | **Tag** | **Catalog #** | **Source** | **GeneID** |
| PTGER2 | N-3xHA | | VectorBuilder | NM_008964.4 |
| GPRC5B | C-Myc-FLAG | | VectorBuilder | NM_001195774.1 |
| GPRC5B-mut7 | C-Myc-FLAG | | VectorBuilder | NM_001195774.1 |

**Table 2 | Primer sequences qRT-PCR**

| Gene | | Sequences | Species |
|---|---|---|---|
| IL-1b | Forward | AGACAGGTCGCTCAGGGTCA | Mouse |
| | Reverse | AAGTGGTTGCCCATCAGAGG | Mouse |
| NOS2 | Forward | AAGGGGACGAACTCAGTGG | Mouse |
| | Reverse | CCCGGAAGGTTTGTACAGC | Mouse |
| IL-6 | Forward | TGATGGATGCTACCAAACTGG | Mouse |
| | Reverse | TTCATGTACTCCAGGTAGCTATG | Mouse |
| IL-10 | Forward | CAGAGCCACATGCTCCTAGA | Mouse |
| | Reverse | TGTCCAGCTGGTCCTTTGTT | Mouse |
| TNF | Forward | GGTCTGGGCCATAGAACTGA | Mouse |
| | Reverse | TCTTCTCATTCCTGCTTGTGG | Mouse |
| Occludin | Forward | ATGTCCGGCCGATGCTCTC | Mouse |
| | Reverse | TTTGGCTGCTCTTGGGTCTGTAT | Mouse |
| ZO-1 | Forward | TTTTTGACAGGGGGAGTGG | Mouse |
| | Reverse | TGCTGCAGAGGTCAAAGTTCAAG | Mouse |
| MUC2 | Forward | GGGAGGGTGGAAGTGGCATTGT | Mouse |
| | Reverse | TGCTGGGGTTTTTGTGAATCTC | Mouse |
| IL-17 | Forward | ATCAGGACGCGCAAACATGA | Mouse |
| | Reverse | TTGGACACGCTGAGCTTTGA | Mouse |
| PTGER2 | Forward | GATGAAGCAACCAGAGCAGAC | Mouse |
| | Reverse | CAGAGAGGACTCCCACATGAA | Mouse |
| GPRC5B | Forward | CGGGCCTACATGGAGAACAA | Mouse |
| | Reverse | GGACGCATTTCAGTCCCT | Mouse |
| GAPDH | Forward | TCCTCAGTGTAGCCCAAGA | Mouse |
| | Reverse | GGAGAAACCTGCCAAGTATGA | Mouse |
| GPRC5B | Forward | CCGCAGAGATGTGACTCG | Human |
| | Reverse | TCTCTGATGCCACGAACATT | Human |
| PTGER2 | Forward | CTCTCCTTGTTCCACGTGCT | Human |
| | Reverse | GCCAGGCTGAAGAAGGTCAT | Human |
| GAPDH | Forward | GCATCCTGGGCTACACTGA | Human |
| | Reverse | CCAGCGTCAAAGGTGGAG | Human |

## Western blotting

Samples were lysed in RIPA buffer (100 mM Tris-HCl pH 7.5, 5 mM EDTA, 50 mM NaCl, 50 mM β-glycerophosphate, 50 mM NaF, 0.1 mM $Na_3VO_4$, 0.5% NP-40, 1% TritonX-100, 0.5% sodium deoxycholate) supplemented with protease/phosphatase inhibitors (Thermo Fisher Scientific, 78445). Proteins were separated by SDS-PAGE and transferred onto nitrocellulose membranes (Cytiva, 10600003). After blocking in 5% (w/v) or 3% (w/v) BSA for 1 h at room temperature, membranes were incubated overnight or for 2 days at 4 °C with primary antibodies (Table 3). The membranes were then washed three times with TBST for 15 min and incubated with horseradish peroxidase-conjugated antibodies directed against rabbit IgG (1:1000, Cell Signalling Technology, 7074) for 1 h at room temperature. Target proteins were visualized by enhanced chemiluminescence reagent (Millipore, WBKLS0500) and a ChemiDoc MP Imaging System using Image Lab Software (Bio-Rad). Quantification of band intensities was done using Fiji software.

## FACS

Cells were pelleted by centrifugation ($500 \times g$ for 10 min at 4 °C) and resuspended in PBS. For each 200 µl of cells, 1 µl of antibody per $10^7$ cells was added. The cells were incubated for 10 min at room temperature in the dark, then 400 µl of PBS was added. Cells were pelleted by centrifugation ($500 \times g$ for 10 min at 4 °C) to remove non-binding

**Table 3 | Antibodies for Western blotting**

| Primary antibody | Catalog # | Clone | Source | Dilution |
|---|---|---|---|---|
| GAPDH | 2118 | 14C10 | Cell Signaling Technology | 1:1000 |
| GPRC5B | | | Cesare Orlandi, Rochester[21] | 1:1000 |
| FLAG-tag | A8592 | M2 | Sigma-Aldrich | 1:1000 |
| HA-tag | H6533 | HA-7 | Sigma-Aldrich | 1:1000 |
| EP2 | ab167171 | EPR8030(B) | Abcam | 1:1000 |

antibodies, and then resuspended in 200–500 µl of PBS and analysed by FACS. 2.5 µl of AccuCount Fluorescent Particles (Spherotech, ACFP-100-3) was added into each sample to count the absolute cell number by flow cytometry.

For the analysis of immune cell populations from blood, spleen, and peritoneum, the following antibodies were used: CD45-FITC, CD19-PE, Ly6G-APC, CD11b-eFluor450, TCRβ-Percp-cy5.5 (leukocyte); CD45-FITC, CD11b-eFluo450, Ly6C-PE-Cy7, F4/80-PE-Cy5, Ly6G-APC (myeloid cell).

For the analysis of peritoneal leukocytes during bacteria-induced peritonitis, we used CD45-FITC, CD11b-BV510, Ly6G-BU421, Ly6G-APC, CD19-eFluor450, CD19-PE, TCRβ-BU421, Ly6C-PE-Cy7, F4/80-PE, MHCII-APC-Cy7, Tim4-PerCP-cy5.5, CCR2-APC (populations defined according to[56,57]).

For the analysis of heart-infiltrating cells, we used CD11b-BU510, Ly6G-BV421, F4/80-APC- eFluor780, CD206-PE, Ly6C-FITC.

For the analysis of colonic-infiltrating cells, we used CD45-FITC, CD19-PE, TCRβ-Percp-cy5.5, CD11b-BU510, Ly6G-BU421, Ly6C-PE-Cy7, F4/80-APC- eFluor780.

For the analysis of skin-infiltrating cells, we used CD45-FITC, CD19-PE, TCRβ-BU421, Ly6G- Percp-cy5.5, CD11b-BV510, Ly6C-PE-Cy7, F4/80-APC-eFluor780.

For the analysis of peritoneal leukocytes during thioglycolate-induced peritonitis, CD45-FITC, CD19-PE, TCRβ-BU421, Ly6G- APC, CD11b-BV510.

All flow cytometric analyses were performed using a FACS Canto II flow cytometer (BD Biosciences). Gating strategies are shown in supplemental file. FlowJo software (version 10.8.1) was used for data analysis. For antibody information, see Supplementary Table 1.

### Transwell migration

Mouse peritoneal cells were resuspended at a density of $3 \times 10^5$ cells/ml in RPMI 1640 containing 10% FCS and LPS 1 μg/ml, and 100 μl of the cell suspension were added to 8 μm pore size 96 well inserts (Corning, 3374). The lower wells contained either 250 μl of medium alone (RPMI 1640 with 5% FCS) or 250 μl of medium containing 20 ng/ml C5a. Cells were allowed to transmigrate for 3 h at 37 °C and 5% $CO_2$, then inserts were discarded and transmigrated cells collected from the bottom well. After washing cells in cold PBS, the number of transmigrated cells was determined by flow cytometry. In order to count the absolute cell number by flow cytometry, AccuCount Fluorescent Particles were used. For the analysis, CD45-FITC and F4/80-APC-eFluor780 were used.

Mouse M1 BMDM were suspended at a density of $1 \times 10^6$ cells/ml in DMEM/F-12 and 100 μl of the cells suspension were added to 8 μm pore size 96 well insert. The lower wells contained either 250 μl of medium alone (DMEM/F-12 with 5% FCS) or medium containing CCL5 (75 ng/ml), CCL2 (10 ng/ml), SDF-1β (100 ng/ml), C5a (20 ng/ml), fMLP (10 nM). After 3 h of incubation at 37 °C and 5% $CO_2$, transmigrated cells were measured by flow cytometry. In order to count the absolute cell number by flow cytometry, AccuCount Fluorescent Particles were used. For the analysis, CD45-FITC and F4/80-APC-eFluor780 were used.

Mouse neutrophils were suspended at a density of $1 \times 10^6$ cells/ml in RPMI 1640 and 100 μl of the cell suspension were added to 5 μm pore size 96 well insert (Corning, 3388). The lower wells contained either 250 μl of medium alone (RPMI 1640 with 5% FCS) or medium containing C5a (20 ng/ml), fMLP (10 nM), LTB4 (1 uM), CXCL1 (1 uM), or CXCL3 (1uM). After 3 h of incubation, transmigrated cells were measured by flow cytometry. In order to count the absolute cell number by flow cytometry, AccuCount Fluorescent Particles were used. For the analysis, CD45-FITC and Ly6G-APC were used.

### Ibidi chemotaxis

Ibidi chemotaxis assay using peritoneal macrophages were performed as previously described[58], with some modifications. Briefly, MACS-isolated RPM were suspended at a density of $3 \times 10^6$ cells/ml in RPMI 1640 containing 10% FCS and 1 μg/ml LPS. 6 μl of cell suspension were seeded into the μ-Slide Chemotaxis chamber (Ibidi GmbH, 80326). After 3 h, the chemotaxis chamber was filled with RPMI 1640 containing 20 mM 4-(2-hydroxyethyl)-1-piperazineethanesulfonic acid (HEPES, Gibco, 15630), 10 % (w/v) FCS and 1 μg/ml LPS. Then, medium alone or medium containing C5a 20 nM or CCL5 10 ng/ml or fMLP 10 nM was added to the opposite reservoir. 5 μl of crystal violet (1 μg/μl in PBS, Sigma-Aldrich, 61135) was used as a visual indicator of concentration gradient formation. Phase-contract images were captured using an Olympus IX81 live cell imaging system (4× objective lens) at 37 °C in an environmental chamber with humidified atmosphere and 5% $CO_2$. Images were collected at 5 min interval for 12 h. Images were imported into Imarisx64.9.7.2 and randomly chosen cells were tracked for each experiment and analyzed with the Chemotaxis and Migration Tool (Ibidi GmbH).

### Phagocytosis assay of E. coli

To determine the phagocytotic activity of RPM, MACS-isolated RPM were suspended at a density $10^6$ cells/ml in RPMI 1640 containing 10% (w/v) FCS and 100 ng/ml LPS, and 100 μl of this suspension were seeded into the wells of a 96 well plate. After 6 h, the medium was replaced by 100 μl of serum-free medium containing Hoechst 33342 (Thermo Fisher Scientific, H3570) to quantify the number of cells. After 10 min, the medium was removed and 100 μl of RPMP1640 medium containing pHrodo deep red E. coli bioparticles (1 mg/ml, Invitrogen, P35360) were added. Live cell imaging of pHrodo fluorescence was performed using the Zeiss Axio live cell imaging system (10x objective lens) in an environmental chamber at 37 °C. Images were collected for 6 h every 10 min. For the quantification, integrated density was determined using Fiji and resulting values were normalized to the numbers of cells.

To determine the phagocytotic activity of BMDM, M0 BMDM were suspended at a density of $10^6$ cells/ml in DMEM/F-12 containing 10% (w/v) FCS and 100 μl of cell suspension were seeded into the wells of a 96 well plate. After 24 h, the medium was replaced by 90 μl of serum-free DMEM/F-12 medium. 10 μl of medium alone or medium containing ligands (butaprost 5 μM or PGE$_2$ 10uM) were added for 15 min. Thereafter, 100 μl of serum-free medium containing pHrodo deep red E. coli bioparticles (1 mg/ml) was treated. The cells were observed using Zeiss Axio live cell imaging system (10x objective lens) in an environmental chamber at 37 °C every 10 min for 6 h. For the quantification, integrated density was determined using Fiji and resulting values were normalized to the numbers of cells.

To determine the phagocytotic activity of neutrophils, BM-derived neutrophils were suspended in RPMI 1640 containing 10% (w/v) FCS at a density of $3 \times 10^6$ cells/ml and 100 μl of this cell suspension was seeded into the wells of a 96 well plate. 100 μl of pHrodo FITC E.coli bioparticles (1 mg/ml, Invitrogen, P35366) were added and incubated for 30, 60, and 90 min at 37 °C. After incubation, the cells were placed on ice to halt phagocytosis and then washed two times with cold PBS. To evaluate phagocytic activity, the number of $FITC^+;Ly6G^+$ cells was determined by flow cytometry. In order to count the absolute cell number by flow cytometry, AccuCount Fluorescent Particles were used as reference particles.

### Phagocytosis assay of apoptotic cells

Phagocytosis assays with apoptotic cells were conducted as described (Miksa, 2008)[59], with some modifications.

For induction of apoptosis in thymocytes, mouse thymus was isolated from C57BL/6 J mice. Thymocytes were suspended at density $10^7$ cells/ml in DMEM containing 10% (w/v) FCS, 2 mM L-glutamine, 5 units/ml penicillin/streptomycin and 0.1 μM dexamethasone (Sigma-Aldrich, D4902) was added for 16 h. Before performing the phagocytosis assay, apoptosis was assessed using 7-AAD staining (BD Biosciences, 559925) by flow cytometry. Apoptosis was generally over the 85 %. After the induction of apoptosis, thymocytes were washed 2 times with PBS and resuspended in PBS at $10^6$ cells/ml. 50 ml of cell suspension was mixed with 1 μl of 1 mg/ml pHrodo-SE (Invitrogen, P36600). After incubation for 30 min at room temperature, cells were washed 2 times with PBS and resuspend in Opti-MEM at $4 \times 10^6$ cells/ml.

For the phagocytosis assay, MACS-isolated RPM ($10^5$ cells per well) were seeded onto a 96 well plate. After 24 h, 100 μl of $4 \times 10^5$ pHrodo-SE-labeled apoptotic thymocytes were added and phagocytosis was assessed by using Olympus IX81 live cell imaging system (10x objective lens) in an environmental chamber at 37 °C (images taken every 10 min for 2 h). For the quantification, integrated density was determined using Fiji and resulting values were normalized to the numbers of cells.

## NOx production

Nitrate/nitrite concentration was measured using a Colorimetirc Assay Kit (Cayman, 780001), according to the manufacturer's instruction. Briefly, $2 \times 10^5$ cells of M0 BMDM were treated with 1 µg/ml LPS in DMEM/F-12 containing 10% (w/v) FCS, 2 mM L-glutamine, 5 units/ml penicillin/streptomycin for 24 h. Total nitrate/nitrite level in the supernatant was measured by Flexstation 3 (Molecular Devices) at 540 nm.

## Multiplex bead array assay

$10^6$ cells of M0 BMDM were treated with 1 µg/ml LPS in DMEM/F-12 containing 10% (w/v) FCS, 2 mM L-glutamine, 5 units/ml penicillin/streptomycin. After 24 h, cytokines and chemokines in the supernatant were assessed by using a mouse cytokine/chemokine multiplex bead array (Millipore) and a Luminex MAGPIX analyzer, according to the manufacturer's protocol.

## Immunoprecipitation

For immunoprecipitation of overexpressed proteins, HEK cells were lysed with RIPA buffer supplemented with protease/phosphatase inhibitors for 15 min on ice. Cell lysates were centrifuged at 13,000 rpm for 15 min at 4 °C and protein concentration was determined by BSA assay. 600–700 µg of lysate was incubated with 20 µl of HA-tagged magnetic beads (MBL International, M180-11) or FLAG-tagged magnetic beads (Thermo Fisher Scientific, M8823) at 4 °C with gentle rotation for 2 h. After 3 washes with cold RIPA buffer, proteins bound to the beads were eluted by boiling the beads in 60 µl of 4x Laemmli sample buffer at 50 °C for 5 min and then separated by SDS-PAGE, transferred to a nitrocellulose membrane and immunoblotted with indicated antibodies. Target proteins were visualized by enhanced chemiluminescence reagent and a ChemiDoc MP Imaging System using Image Lab Software (Bio-Rad). Per sample, 600–700 µg of protein was used for immunoprecipitation and 20 µg of total lysates were used for input controls.

For immunoprecipitation of endogenous EP2, RPM were lysed in RIPA buffer supplemented with protease/phosphatase inhibitor cocktail for 15 min on ice. Samples were centrifuged at 13,000 rpm for 15 min at 4 °C. Supernatant was incubated with 10 µl of the anti-EP2 (Abcam, ab167171) at 4 °C with gentle rotation overnight. Thereafter, 20 µl of A/G-Sepharose beads (Santa Cruz, sc-2003) were added and incubated for 1 h at 4 °C with gentle rotation. After 3 times washed with cold RIPA buffer, proteins bound to the beads were eluted with 40 µl of 4x Laemmli sample buffer at 50 °C for 5 min. Samples were subjected to SDS-PAGE, transferred to a nitrocellulose membrane and immunoblotted with Veri-Blot IP detection reagent (Abcam, ab31366). Per sample, 500 µg of protein was used for immunoprecipitation and 20 µg of total lysates were used for input controls.

## Calcium mobilization assays

HEK-293T cells were transfected with siRNA against GPRC5B by using Opti-MEM and Lipofectamine RNAiMAX according to the manufacturer's instructions. After 20–24 h, Lipofectamine 2000 was used to transfect HEK cells with plasmids containing cDNAs encoding EP1 and a calcium sensitive bioluminescent fusion protein. After 24 h, 100 µl of cells ($4 \times 10^5$ cell/ml) were seeded onto 96 well plate. The next day, cells were incubated for 2 h at 37 °C in the presence of 5 µM coelenterazine h in HBSS containing 10 mM HEPES. Calcium transients were assessed for 2 min after 17-pt-PGE$_2$ (1 uM) addition by using a luminometric plate reader (Flexstation 3). The area under each calcium transient was calculated by using SoftMaxPro software (Molecular Devices) and data were expressed as area under the curve.

## cAMP determination

For the examination of cAMP levels in HEK cells, cells were transfected with plasmids containing cDNAs encoding GPRC5B (0.1 µg) and the indicated prostanoid receptors (0.1 µg). After 24 h, 100 µl of cells ($4 \times 10^5$ cell/ml) were transferred to a 96 well plate and after another 16–20 h medium was replaced with serum-free DMEM containing 50 µM IBMX for 30 min. Thereafter, ligands were added for 30 min. The cAMP levels in cell lysates were measured by using direct cAMP kit (Enzo lifesciences, ADI-900-066) according to the manufacturer's protocol. Protein concentration of the same samples was determined by BSA protein quantification assay. Data were normalized to the protein amounts (((cAMP pmol/ml)/mg of protein).

In order to determine cAMP levels in macrophages, RPM and M0 BMDM were seeded at $10^5$ cells per well onto 96 well plate. The next day, cells were treated with 1 µg/ml LPS for 6 h. Thereafter, the medium was replaced with serum-free medium containing 50 µM IBMX for 30 min and then ligands were added for 30 min. cAMP levels in cell lysates were measured using the same kit as for HEK-293T cells.

To determine cAMP levels in THP-1 cells, $10^6$ THP-1 cells were transfected with plasmids containing cDNA encoding GPRC5B (0.5 µg) or siRNA against human GPRC5B. After 48 h, the cells were transferred to a 96 well plate and 50 µM IBMX was added for 30 min, followed by butaprost for 30 min. cAMP levels in cell lysates were measured using the same kit for HEK-293T cells.

For GloSensor-based determination of cAMP levels, HEK cells were transfected with plasmids containing cDNAs encoding GPRC5B (0.1 µg), the indicated prostanoid receptors (0.1 µg), and the GloSensor 22 F (0.1 µg, Promega, E2301). After 24 h, 100 µl of cells ($4 \times 10^5$ cell/ml) were transferred to a 96 well plate. The next day, the medium was replaced with serum-free DMEM containing 50 µM IBMX and 3 mM luciferin D. After 2 h, ligands were added and luminescence was measured for 30 min. The area under curve of the luminescence traces was calculated by using SoftMaxPro software.

## Immunofluorescence staining

For immunofluorescence staining in HEK cells, sterilized circular cover glasses were placed in a 48 well plate and then coated with poly-D-lysine (2 µg /cm², Sigma-Aldrich, P9155) for 45 min at 37 °C followed by laminin coating (1 µg/cm², Sigma-Aldrich, L2020) at 37 °C overnight. 200 µl of HEK cells ($4 \times 10^5$ cell/ml) were seeded on cover glasses. The next day, medium was removed and cells were washed twice with HBSS. Cells were incubated with WGA (5 µg/ml, Biotium, W11261) for 10 min at room temperature. After 3 times washes in HBSS, cells were fixed with 4% PFA for 10 min at room temperature followed by washing three times with HBSS. Cells were permeabilized in PBS containing 0.1% Triton X-100 during 10 min at room temperature. Blocking was done with PBS containing 2% BSA for 30 min at room temperature and incubation with primary antibodies was performed for 90 min at room temperature. Primary antibodies used are listed in Supplementary Table 2. After incubation with primary antibodies, cells were washed three times with PBS and incubated with secondary antibodies (1:500 dilution) for 1 h at room temperature. Nuclei were counterstained with 4',6-Diamidino-2-Phenylindole (DAPI, D3571, 1:1000 dilution). After 3 times washing with PBS, the slides were mounted with Fluromount W (Serva, 21634.01). To visualize the plasma membrane with high contrast, single optical sections through the axial center of the cells were obtained using a LD LCI Plan-Apochromat 63x/1,2 Imm Korr DIC M27 objective on a Zeiss LSM 880 confocal microscope with Airyscan 2 detector in SR mode. To calculate the overlap between HA-EP2 and the WGA-positive plasma membrane, the WGA signal was filtered using a mean filter (FIJI, radius=4) and segmented using a pixel classifier (Ilastik) trained on the experimental data. The segmentation result was cleaned by size filtering (FIJI), and used for intensity measurements (MorpholibJ) of the HA-EP2 signal. The Pearson's correlation coefficient was determined using FiJi.

For immunofluorescence staining in RPM, 200 µl of cells ($1 \times 10^6$ cell/ml) were seeded on sterilized circular cover glasses coated with poly-D-lysine (2 µg/cm²) for 30 min at 37 °C. After 3 h, cells were

washed two times with HBSS and fixed with 0.4% PFA for 10 min at 4 °C followed by washing three times with HBSS. For cell organelle staining, cells were permeabilized with 0.1% Triton X-100 during 1 min at room temperature followed by washing three times with HBSS. Cells were then blocked with 2% BSA in PBS for 30 min at room temperature and incubated with primary antibodies for 90 min at room temperature. After three times washes with PBS, cells were incubated with secondary antibodies (1:500 dilution) and CD11b (cell surface marker) for 1 h at room temperature and washed three times with PBS and mounted with fluromount. Images were obtained by using Zeiss LSM 880 with Airyscan 2. The Pearson's correlation coefficient was determined using Fiji. EP2 in CD11b-positive plasma membrane was calculated using Fiji and Ilastik. For antibody information, see Supplementary Table 2.

## Measurement of surface EP2 expression

100 µl of cells ($1 \times 10^6$ cell/ml) were seeded on 96-well plate coated with poly-D-lysine (0.1 mg/ml) and incubated overnight. Cells were washed 2 times with PBS and fixed with PFA 0.4% in PBS for 10 min at 4 °C. Cells were washed 3 times with PBS and then blocked for 30 min with 2% BSA in PBS at room temperature. Thereafter, cells were incubated with anti-EP2 (extracellular) antibody (1:2000 dilution, 2% BSA in PBS) for 2 h at room temperature. After 3 times washes with PBS, cells were incubated with horseradish peroxidase-conjugated antibodies directed against rabbit IgG (1:1000). Cells were washed with 3 times with PBS and then bound antibodies were detected by incubation with TMB Substrate (Cell Signaling Technology, 7004) for 10 minutes and thereafter the reaction was stopped using STOP Solution (Cell Signaling Technology, 7002). The absorption intensity was determined using a plate reader (Flexstation 3) at 450 nm.

## Generation of a GPRC5B homology model and in silico protein-protein docking with prostanoid receptors

General description: A homology model of GPRC5B was created using metabotropic glutamate receptor 2 (mGlu2) with PDB ID 7mts as the template. Although the template covered only 64% of GPRC5B, the transmembrane domain of the receptor was included, thus providing sufficient information for meaningful modelling. The homology model included residues 48–296 of GPRC5B and the sequence similarity was 32%. The homology models of EP1, DP1 and IP were created based on the structure of EP2 (PDB ID 7cx2).

To identify conserved residues within GPRC5B, a sequence alignment was performed. Aligning the sequences of GPRC5B and GPRC5C revealed four conserved regions within the transmembrane region of GPRC5B. These conserved regions are designated as follows: region_A spans residues 97 to 137, region_B includes residues 55 to 66 and 267 to 291, and region_C encompasses residues 198 to 211.

The ab initio protein-protein docking showed that two regions were favorable for binding to the prostanoid receptors. Interestingly, these two regions coincided with region_A and region_B. Region_C rarely showed any participation in forming complexes with the four prostanoid receptors.

In the next step, a series of targeted protein-protein docking calculations was conducted. This time either residues of region_A or of region_B were defined as interacting residues from GPCR5B's side. In the case of region_A, docking demonstrated more reasonable docking poses. In these poses, the prostanoid receptors interact with their transmembrane helices II, III, IV and V with the transmembrane helices II and III of the GPRC5B receptors. On the other hand, docking prostanoid receptors to region_B of GPCR5B produced poses in which the prostanoid receptors bind with their extracellular domain facing towards the transmembrane domain of the GPRC5B receptor. Although the simulation results suggested such a perpendicular arrangement, it is highly unlikely given that both GPRC5B and the prostanoid receptor are located within the cell membrane.

Based on the docking results, region_A was considered as the most likely region of GPCR5B to interact with the prostanoid receptors. According to the sequence alignment and the protein-protein docking results, the following major residues within region_A of the GPRC5B receptor were considered for site-directed mutagenesis: set1: F97, L101 and L104; set2: F111; and set3: Q116, D118 and L128 (Fig. 6B).

After experimental confirmation of the involvement of set1 and the partial role of set3 residues in the interaction between GPRC5B and EP2, another round of protein-protein docking calculations was conducted to refine the complex. In the refined complex, it was predicted that EP2 interacts with GPRC5B via transmembrane helices II, III, and IV instead of helices IV and V, as was the case in the previously predicted complex. Furthermore, the alignment of the two receptors was improved such that their extracellular, transmembrane, and intracellular domains were in close proximity, situated at the same level along their z-axis. Notably, the receptors were nearly perfectly parallel to each other. The final predicted conformation of the complex is shown in Fig. 6A.

Methodological details: SWISS-MODEL server[60] was used to create a homology model for GPRC5B (UniProt ID Q9NZH0) and EP1, IP and DP1. The 3D structure of EP2 (UniProt ID Q9NZH0) is available with PDB ID 7CX2. However, because there were significant gaps in the structure, homology modeling was employed to model the missing loops of EP2. The global QMEANDisCo (Qualitative Model Energy ANalysis Distance Constraints) scores for the EP2 and GPRC5B models are 0.67 and 0.62, respectively, while the values for the transmembrane domains are 0.81 and 0.73, respectively.

Sequence alignment between GPRC5B and 5 C was done using the UniProt sequence alignment tool. Both proteins (GPRC5B and GPRC5C) have the same interaction pattern with other receptors. In other words, the two interaction partners and their relative affinity are the same. Therefore, it seems likely that the residues involved in the interactions should be conserved.

By considering four or more consecutive conserved residues as a criterion to define the conserved region, six regions were identified. The sequence alignment shows these six regions, four of which are located within the modeled residues (regions A, B and C) whereas two other regions are located within the cytoplasmic part of GPRC5B.

The HADDOCK 2.4 server[61,62] was used for protein docking. It has two methods of (ab initio) blind docking. One uses random patches and the other uses center of mass restraints. They are suitable when no information is available for both proteins. Both approaches were used and for both, the sampling number was increased from 1000 to 50000 and 200 to 400 for it0 (rigid body minimization stage) and it1 (semi-flexible simulated annealing stage), respectively.

Based on the results of blind docking, region_A and region_B were identified as possible targets for prostanoid receptors. Therefore, these two regions of GPRC5B (region_A and region_B) were selected for targeted protein-protein docking. For this purpose, all accessible residues of prostanoid receptors were defined as passive residues in HADDOCK. For GPRC5B either region_A or region_B residues were defined as active residues and the surrounding accessible residues were defined as passive residues. The sampling number was increased from 1000 to 10000 and 200 to 400 for stages it0 and it1, respectively. In HADDOCK, active residues are the residues that are penalized in the evaluation of docking if they do not participate in the interaction, whereas passive residues are the residues that could participate in the interaction, but the evaluation is not penalized if they do not interact with the binding partner.

After confirmation of the interacting residues of GPRC5B with EP2 by mutagenesis, we tried to refine the model of the complex to be consistent with the experimental results and orientation in the membrane. The LZerD protein docking server[63] has an interface for docking of membrane proteins. It accepts transmembrane proteins as input, provided in a PDB file format where the protein is positioned relative to

the membrane. The produced conformations were evaluated to find the best one in terms of interacting residues of GPRC5B and orientation of the complex in the membrane. The final model was refined by HADDOCK refinement server.

## RNA sequencing

Library-Preparation: For RNA-seq analysis, total RNA was isolated from $1 \times 10^5$ RPM or $1.6 \times 10^6$ M0 BMDM per mouse using the RNeasy micro Kit (Qiagen, 74004) combined with on-column DNase digestion (RNase-Free DNase Set, Qiagen, 79254) to avoid contamination by genomic DNA. RNA and library preparation integrity were verified with LabChip Gx Touch 24 (Perkin Elmer). 1 µg of total RNA was used as input for VAHTS Stranded mRNA-seq V6 Library preparation following manufacture's protocol (Vazyme). Sequencing was performed on NextSeq2000 instrument (Illumina) with 1x72bp single end setup.

RNA-Seq analysis: Trimmomatic version 0.39 was employed to trim reads after a quality drop below a mean of Q15 in a window of 5 nucleotides and keeping only filtered reads longer than 15 nucleotides[64]. Reads were aligned versus Ensembl mouse genome version mm10 (Ensembl release 101) with STAR 2.7.10a[65]. Aligned reads were filtered to remove: duplicates with Picard 2.27.4, multi-mapping, ribosomal, or mitochondrial reads. Gene counts were established with featureCounts 2.0.3 by aggregating reads overlapping exons on the correct strand excluding those overlapping multiple genes[66]. The raw count matrix was normalized with DESeq2 version 1.36.0[67]. Contrasts were created with DESeq2 based on the raw count matrix. Genes were classified as significantly differentially expressed at average count > 5, multiple testing adjusted $p$-value < 0.05, and −0.585 <log2FC > 0.585. The Ensemble annotation was enriched with UniProt data (Activities at the Universal Protein Resource (UniProt)).

## Lipidomics analysis of ceramides and sphingoid bases using LC-MS/MS

The analysis and evaluation was performed as described previously in ref. 68. Briefly, to the cell pellets ($2.5 \times 10^5$ in 50 µL PBS), 200 µl citric acid buffer 30 mM with disodium hydrogen phosphate 40 mM, 20 µl internal standard solution and 600 µl chloroform/methanol/HCl (83:15:2, v/v/v) were added and vortexed vigorously. After centrifugation for 5 min at 15000 rpm, the organic phase was split into two and evaporated at 45 °C under a nitrogen stream. Prior analysis of ceramides the samples were dissolved in 50 µl tetrahydrofuran/02% formic and 10 mM ammonium formate (9:1, v/v). The samples for analysis of sphingoidbases were dissolved in 50 µL methanol/formic acid (95:5, v/v). The analysis was performed on a QTrap 6500+ coupled to an Agilent 1290 Infinity II UHPLC system. Chromatographic separation was achieved using a Zorbax Eclipse Plus C18 2.1 × 50 mm 1.8 µm column (Agilent) for analysis of ceramides and a Zorbax Eclipse Plus C8 2.1 × 30 mm 1.8 µm (Agilent) for the analysis of sphingoid bases with a pre-column of the same type, respectively.

## NanoString analysis of GPCR expression in bulk RNA

NanoString analyses were performed as described previously in refs. 69,70. In brief, 250-500 ng RNA from sorted RPM (CD11b[+], F4/80[+]) or sorted lymph node cells (CD4[+] or CD8[+] or CD19[+]) cells was applied in a total volume of 30 µl in the assay. Barcodes were counted for ~1,150 fields of view per sample. Counts were first normalized to the geometric mean of the positive control spike counts, then a background correction was done by subtracting the mean + two standard deviations of the eight negative control counts for each lane. Data were not normalized to reference genes because none of the reference genes showed sufficiently stable expression in all cell types according to the geNorm algorithm. Values that were <20 were fixed to background level.

## NanoBRET

The NanoBRET assay was performed following the technical manual for the NanoBRET Protein:Protein Interaction System (Promega). In brief, cells were transfected with different amounts of plasmids C-terminally fused with NLuc or venus via a linker encoded by DNA sequence 5′- GGTGGCACCGGTGGATCC-3′; the control vector was expressing only NLuc. After 24 h, 100 µl of cells ($4 \times 10^5$ cells/ml) were transferred to a poly-l-lysine-coated clear bottom white wall 96 well plate. After another 16–20 h, medium was replaced with 75 µl of serum-free Opti-MEM with or without control or target peptide (10 µM) and cells were incubated for 1 h. Thereafter, 25 µl of Nano-Glo Substrate (25 µl of 5X solution in 2.5 ml of Opti-MEM) (Promega, N1571) was added and mixed for 30 s. Readings were performed within 10 min using Flexstation 3 at 535 nm (acceptor, Venus) and 460 nm (donor, NLuc) with an integration time of 1 s. The raw BRET ratio was calculated by dividing the acceptor emission value (535 nm) by the donor emission value (460 nm), and further converted into milliBRET units (mBU) by multiplying by 1000.

## Initial FRET measurements

For the initial FRET measurements, 1 µg of EP2 receptor-Turq2 and 1 µg of P2Y12-mCitrine, GLP1R-mCitrine, or ß2AR-mCitrine were transiently transfected into HEK293T cells. Effectene transfection reagent from Qiagen (301425) was used. FRET signals were recorded from selected single cells using an inverted microscope (Axiovert 100) and a previously described measurement setup[71], which received a new pco.-panda 4.2 sCMOS camera due to a necessary technical update, equipped with an oil-immersion objective (UPlanSApo 60x/1.35 Oil Olympus). During FRET measurements, excitation was applied at approximately 1 Hz, with data recorded for 20 s. Data was collected by the VisiView software (Visitron Systems). The collected data was corrected for background fluorescence, bleed-through and false excitation using Microsoft Excel. These data were referred to as emission ratio mCit/mTq2. These data were referred to as YFP/CFP initial FRET. The average of 5 initial values was then calculated and plotted in a bar graph for comparison.

Plasmids used in FRAP experiments were generated as follows: mCitrine-tagged β₂-adrenergic and GLP1 receptors were constructed by replacing the M3 muscarinic receptor in an M3-mCitrine construct[71] with the ORF for the respective receptor by excising the cDNA for the M3 receptor with HindIII and XbaI and ligating in corresponding fragments that had been generated by restriction digest from Flag-tagged β₂AR-YFP[72] and GLP1R-YFP[73], respectively. All constructs were verified by Sanger sequencing.

## Data presentation and statistical analyses

All data are presented as means ± standard errors of the means (SEM). The following statistical tests were used for normally distributed samples: unpaired or paired student's t-test for comparisons between two groups, one-way ANOVA with Dunnett's multiple comparisons test for multiple groups, two-way ANOVA with Sidak's multiple comparisons test for comparisons between two groups with different treatments, two-way repeated measures ANOVA with Bonferroni's or Sidak's post hoc test for two groups over time. For data sets that did not follow normal distribution, Mann-Whitney test was used for the comparison of two groups, in some cases followed by the two-stage step-up method (Benjamini, Krieger, and Yekutieli) to adjust for multiple testing. "n" refers to the number of independent experiments or mice per group. The number of mice per group was determined based on power calculations for the primary parameter with mean differences and standard deviations taken from pilot data at a power of 80% with a standard level of significance of 0.05. Exemplary images and blots were selected to represent the average of the respective cohort. All statistical analyses were performed using GrahPad Prism 10 (version 10.1.2).

**Reporting summary**

Further information on research design is available in the Nature Portfolio Reporting Summary linked to this article.

## Data availability

The mRNA sequencing data used in this study are available in the GEO database under accession code GSE261586 and GSE261584. All other data are available in the article and its Supplementary files or from the corresponding author upon request. Source data are provided with this paper.

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

## Acknowledgements

We thank Martina Winkels, Ulrike Krüger, and Claudia Kopp for expert technical assistance. This work was funded by the *Deutsche Forschungsgemeinschaft* (DFG, German Research Foundation) through project-ID 456687919–SFB 1531 (A04 to N.W., A06 to SO), project-ID 454193335–SFB 1526 (B02 to C.S. & N.W.; S03 to S.M.), project-ID 204083920–SFB 1039 (A10 to NW, ZO1 to G.G.), as well as the LOEWE initiative GLUE (to M.B., P.K., N.W.).

## Author contributions

J.K. performed most experiments and wrote parts of the manuscript. She was supported by H.K. (LAD ligation model), K.M. (image analysis), S.G. (mRNA sequencing), J.H. (assay establishment), L.H. and G.G. (Lipidomics), SM and CS (analyses in neutrophils), M.U. and M.B. (FRET assays), J.S., M.P. and P.K. (virtual docking). C.O., M.K., S.K. and CK provided reagents. M.B., P.K., S.O. interpreted data and reviewed the manuscript, N.W. designed and supervised the study, analyzed data and wrote the manuscript.

## Funding

## Competing interests

The authors declare no competing interests.

 
