## [Transparent Peer Review file · Nature Communications]

Orphan G protein-coupled receptor GPRC5B controls macrophage function by facilitating prostaglandin E receptor 2 signaling

Corresponding Author: Professor Nina Wettschureck

Version 0:

Reviewer comments:

Reviewer #1

(Remarks to the Author)

In this manuscript, Kwon and colleagues study the function of the orphan receptor GPRC5B in macrophages and elucidate the underlying mechanisms. The authors first showed that the knockout of GPRC5B enhanced the migration, phagocytosis and activity in resident peritoneal macrophages (RPM, which have highest expression of GPRC5B) and bone marrow-derived macrophage (BMDM) isolated from myeloid-specific knockout mice, as well as bacterial clearance in a peritonitis model. The authors then showed that GPRC5B interacted with several GPCRs including EP2 in co-IP and that manipulation of GPRC5B expression levels affected EP2 signaling and plasma membrane localization in different cell types. Finally, they generated several GPRC5B mutants to analyze EP2 interaction with GPRC5B, signaling and subcellular localization, and synthesized a peptide derived from the second transmembrane domain of GPRC5B to disrupt GPRC5B effects on EP2 function.

Overall, the phenotypic effects of GPRC5B knockout on macrophages are clear. However, the mechanistic studies are not well designed and the experimental data presented in the manuscript do not strongly support the major conclusion that the function of GPRC5B in macrophages is mediated through interaction/dimerization with EP2 to facilitate EP2 signaling.

Major points:

1. GPRC5B-EP2 interaction:

- a. Based on the co-IP results shown in Fig.3A and Suppl. 6A, it is premature to conclude that GPRC5B and EP2 form dimers or interact, although different GPCR members were compared. GPCRs with seven transmembrane domains are very sticky and they easily form nonspecific aggregates, particularly when overexpressed.
- b. Fig. 3K showed the strong colocalization of GPRC5B and EP2. As both are plasma membrane proteins or partially localized at the plasma membrane, this data does not support their dimerization or interaction.
- c. GPRC5B-mut set3 interacted with EP2, but did not affect EP2 signaling (Fig. 6F and 6G). One could argue that the interaction between the two receptors is not absolutely required for GPRC5B to affect EP2.
- d. The disruption of the peptide on the GPRC5B-EP2 interaction in co-IP was not impressive (Fig. 7B).

2. Effect of GPRC5B on EP2 localization:

- a. Throughout the manuscript, the effects of GPRC5B on the surface expression or subcellular distribution of EP2 as quantified using WGA or CD11 as markers were relatively weak (albeit statistically significant).
- b. From images presented in manuscript, it is difficult to see clear effects of GPRC5B on EP2 subcellular localization. For example, in Fig. 5H, it seems more EP2 staining (red) in GPRC5B knockout cells as compared with control cells.
- c. Where is EP2 normally expressed in cell? Fig. 5D, 5J and 6H showed apparent different intracellular distribution of EP2.
- d. The image shown in the top right of Fig. 6C is exactly the same as one shown in the top panel 2nd left of Fig. 6H.
- e. The effects of the peptide on the surface expression and subcellular location of EP2 were not studied.

3. Effect of GPRC5B on EP2 signaling:

- a. It is interesting to note that GPRC5B knockdown inhibited the signaling via EP2, but not EP1 and EP4. However, this

inhibitory effect was no more than 20% (Fig. 3D).

b. The effect of GPRC5B overexpression on basal cAMP production was quite dramatic (Fig. 3F). How to explain?

c. The data shown in Fig. 4J-L are confusing. Although RA stimulation produced opposite effects on the expression of GPRC5B and EP2, it would also effect the expression of many other proteins.

Minor points:

1. What genotypes revealed in Suppl. Fig. 1H-J?

2. Why are different inflammation markers differentially regulated (Fig. 1E, 1F, Suppl. 1G)?

3. In Fig. 5K, should WGA be CD11b?

4. In Fig. 4H, Western blots should be shown.

5. The data presented in Fig. 4H-I are not correct. siRNA and overexpression experiments should have different control groups.

6. In Fig. 5D, based on the figure legend, cells were stained with WGA, fixed, permeabilized and then stained with antibodies. It is confusing how WGA stained Golgi-like intracellular compartments?

7. In Fig. 5J, the images shown in control group have different orientation.

Reviewer #2

(Remarks to the Author)

The manuscript by Kwon et al titled "Orphan G protein-coupled receptor GPRC5B controls macrophage function by facilitating prostaglandin E receptor 2 signaling" examines the function of the orphan GPRC5B in controlling immune cell function via dimerization with the EP2 receptor. The study utilizes a comprehensive approach to study two different populations of primary macrophages derived from wildtype and GPRC5B KO mice and human cells using both in vitro and in vivo approaches. The authors conclude that GPRC5B dimerizes with EP2 receptor in macrophages resulting in an increase in EP2 expression at the cell surface, enhanced anti-inflammatory signaling and responses. The data are robust including 7 Figures with 10 -16+ panels and 9 Supplementary Figures with multiple panels, at times the data seem excessive, which contributes to a lack of focus and strong rationale for the studies. For example, are the in vivo MI and colitis studies essential to the main conclusions of the paper? Other concerns relate the Introduction, which lacks critical relevant background information and Results section that lack strong rationale for studies, detailed interpretation, and conclusions as well as some issues with data analysis and lack of control experiments that provide a mechanistic understanding of GPRC5B regulation of EP receptor function.

Comments.

1. Figure 1 shows GPRC5b expression is greater in RPM then BMDM, generation of GPRC5b KO mouse using LysM CRE specific for macrophages, neutrophils, granulocytes and impact on inflammatory signaling, migration etc. Panel D, is an immunoblot of GPRC5b protein expression and shows multiple bands ranging from 35 to 55 kDa. Are all the bands presumed to be the receptor? This is important since data shown in other Figures show only a small portion of the blot with only one band GPRC5b that migrates at the lower molecular weight 35to 40 kDa and sometimes higher? The authors need to determine if all of the GPRC5b low and high molecular weight species are expressed on the cell surface and determine which GPRC5b form is dimerizing with EP2. Panels J and L are missing statistical analysis. Overall, the description of the data this figure needs to be improved, it lacks clear interpretation of the findings and conclusions.

2. Figure 2. The rationale for studying BMDM in a similar manner RPM is not clear. Panel F, multiple GPCR agonists enhance GPRC5b KO macrophage migration and raises the question of whether GPRC5b regulates expression of the GPCRs via dimerization. Panels I – J, show MI and colitis model study results. It is unclear why these studies are critical to the main conclusions of study.

3. Figure 3. Panel A, HEK293 cells and co-IP are used to demonstrate GPRC5b interaction with EP1, EP2 and DP1 receptor. The immunoblots of GPR5Cb show a band around 40 kDa. Is this appropriate size for GPRC5b? Is the GPRC5b 40 kDa form expressed at the cell surface and colocalized with EP2 at the plasma membrane – in panel D, an apparent non-specific bands from RPM GPRC5b KO lysates migrates at 40 kDa. In panel H and other panels throughout the paper, the statistical analysis comparisons should be between control (si or WT) with GPRC5B si or KO with and without agonist. The conclusions for this set of results are missing.

4. Figure 4. Panel A the GPRC5b band appears as a doublet migrating at 40 kDa? The IgG pulldown lane for G5B looks concerning. The authors should include co-IP from WT and GPRC5b KO mice RPM. Similar studies should be performed in THP1 human macrophages with siRNA GPRC5 knockdown as a control. In panel B, as a control the authors should show that GRPC5b KO does not affect EP2 expression.

5. Figure 5. Panel C data of the EP2 receptor immunoblot need to be quantified from at least 3 independent experiments because the control EP receptor blots looks higher than EP2 expression in KO. Panel D, why is WGA staining appear to be inside the cell. Panel K, images of calreticulum staining are missing.

6. Figure 6. Panel A, how valid is the homology model of GPRC5b-EP2 receptor, can the authors provide the PLDDT (predicted local distance values). Panel D, immunoblot for G5B WT and Mut shows a different size compared to G5B bands shown in Figure 1. Panel F, Immunoblot of EP2-G5B co-IP's need to be quantitated from at least three independent experiments. Panel F and G, show loss of EP2-G5B mut 7 interaction and the function, however G5B mut set3 retains

interaction with EP2 but appears to lose function, which is counter intuitive, and need to address. Panel H, are the G5B wildtype and mut receptors expressed at the cell surface.

7. Figure 7 shows HIV-TAT -peptide expression to block interaction with a modest effect on cAMP formation. Panel B, impact of target peptide on EP2-G5B interaction immune-complex need to be quantified from at least three independent experiments. In addition the authors should determine if the target peptide enhanced EP2 retention in the golgi.

8. The authors should attempt to determine if dimer G5B-EP2 complex functions differently compared to the EP2 protomer for example by examining G protein coupling, desensitization or internalization.

Reviewer #3

(Remarks to the Author)

In this manuscript, Kwon et al. investigate the contribution of the orphan receptor GPCRC5B towards macrophage function, and its mechanism of action. Utilizing a mouse model of myeloid-lineage-specific KO of GCRC5b, they demonstrate that both peritoneal macrophages and BMDM display augmented chemotaxis, phagocytosis, and cytokine production. In a model of peritonitis, myeloid GPCRC5b deficiency appeared beneficial to improve clearance of pathogens. In models of chronic inflammation (myocardial infarction, DSS colitis), myeloid GPCRC5b deficiency was detrimental which they attribute to exacerbated macrophage-mediated inflammation and tissue damage. Detailed experiments in HEK293 cells (with some replicated in primary human macrophages), delineate a negative regulatory impact of GPCRC5b on EP2 signaling through physical interaction and control of surface expression.

Overall, this is a very well conducted study, with detailed and comprehensive experimental design, and thorough investigation of the function and mechanism of the orphan receptor GPCRC5B on macrophages. The manuscript is well written, and the figures are clear and data well presented. I have only a few questions to ensure that their overall conclusions are soundly supported by the data presented:

Major:

1. In macrophages of their M-G5b-KO mice, is EPA surface expression increased, and EPA-mediated signalling augmented as their overall hypothesis would suggest? And does augmented EPA-mediated signaling contribute to the altered in vivo macrophage functions observed in these mice, as their hypothesis would suggest?
2. The structural work, determination of interacting residues, and development of a peptide blocking strategy is very nice. However, I was surprised that there were no experiments to confirm the functional implications of uncoupling GPCRC5b and EP2 interactions. That is, is macrophage function similarly affected by blocking physical interaction of GPCRC5b with EP2 as it is when GPCRC5a is deleted/knocked down? This is particularly important given the authors proposal that peptide-based blockade of GPCRC5B-AP2 interactions could be used therapeutically.

Minor:

- Fig 1A-B – GPRC5b is expressed in both RPM and BMDM (albeit at much lower levels), but appears negligibly expressed in T cells and B cells. Is GPRC5b expressed at similarly high levels in other tissue resident macrophages from other organs in mice, or is this unique to peritoneal macrophages? In addition to BMDM, is GPRC5b expressed in other myeloid cells? Data on neutrophils is presented in Supp Fig 3, but I wonder about circulating monocytes? Platelets? I appreciate that this manuscript focuses on macrophages, but given the importance of prostaglandins towards the functions of these other cells types it would be of interest to know whether this receptor is expressed in other hematopoietic cell populations.

Reviewer #4

(Remarks to the Author)

In this study, the authors investigated the role of the orphan G protein-coupled receptor GPRC5B and the prostaglandin receptor EB2 on the regulation of macrophage function. For this purpose, the authors employed various in vivo models and an impressive array of in vitro and in silico exploratory approaches. The topic of finetuning the pro- and anti-inflammatory regulatory pathways during inflammatory processes is of great scientific interest and certainly holds great translational potential for the development of future therapeutic strategies. Generally, the manuscript is well written, and the results are presented in a structured manner. However, the following points need to be addressed:

- The authors show the expression of GPRC5B in RPMs, BMDMs and lymphocytes, but not in other leukocytes. Is there anything known about granulocytes, e.g. neutrophils? LysM-Cre is also very prominently expressed in neutrophils, so in how far does the conditional knockout refer to the macrophage function and renders possible neutrophil-mediated effects irrelevant?
- Figure 1G-H: While the text refers to LPS treatment of RPMs, the figure refers to C5a, CCL5 and fMLP stimulation. Please clarify.
- Figure 1M-N: Was the activation and recruitment of other immune cells, e.g. neutrophils, significantly altered in M-G5b-KO mice?
- Figure 1N-P: From the figure and the figure legend it remains unclear which bar refers to which genotype. Please clarify. Furthermore, in how far do the observed effects after 24-54 days relate to the findings during the first 24 hours?
- Figure 2A: should GPRC5b expression actually not be fully abrogated in conditional KO mice? Why is there still partial

expression both in RPMs and BMDMs?

- Figure 2B-C: this finding appears contradictory compared to the data shown in Figure 1F and Supplemental Figure 1G. Please clarify.
- Figure 2I-K: Representative images of cardiac histology and exemplary cardiac ultrasound measurements should be included in the figure file.
- Given the effects of GPRC5B knockout in neutrophils shown in Supplemental Figure 3, the negative in vivo results come rather as a surprise. Furthermore, the EBA model is not among the most commonly used models to study the pathophysiological relevance of gene knockouts in neutrophils and the authors should consider additional models. This is very important, as LysM-Cre driven conditional knockout models also affect gene expression in neutrophils.
- Supplemental Figure 4: what is the relevance of this data for the study?
- Supplemental Figure 6B: the knockdown efficiency reaches about 75% on average, with low efficiency samples in the range of roughly 50%. This knockdown efficiency appears rather low and raises the question if it is enough to detect possible effects.
- Figure 4H: a knockdown efficiency of 50% does not appear suitable for functional assays.
- Figure 4J and general comment: gene expression should not only be assessed on the transcriptional level, but also by protein expression.

Version 1:

Reviewer comments:

Reviewer #1

(Remarks to the Author)

Although the authors have tried to address my comments, the following two major points remain.

1. Related to previous review 1a: specific interaction or heterodimerization between GPRC5B and EP2 is one of important conclusions presented in the manuscript and this conclusion is solely based on the data generated from co-IP. As pointed out in previous review, these data are not strong enough to support the conclusion and would like to suggest the authors to prove the two GPCRs can physically interact or are in close proximity in other protein-protein interaction assays, such as live cell BRET or FRET, using other GPCRs as controls, and also determine the effects of decoy peptides.

2. Related to previous review 1b: as the authors think that colocalization of the two receptors supports their interaction or dimerization, can their colocalization be disrupted by decoy peptides?

Reviewer #2

(Remarks to the Author)

The authors made a significant effort to address each issue that I raised in the initial review and there were a lot of comments. The authors have added new data, controls and re-evaluated and improved the data analysis. These changes included in the revised manuscript have considerably strengthen the paper and overall conclusions and warrants publication.

Reviewer #3

(Remarks to the Author)

The authors have addressed my questions sufficiently through the addition of supportive new data as well as additional discussion.

I have no further questions. The manuscript is comprehensive and scientifically sound, and presents a compelling and novel story.

Reviewer #4

(Remarks to the Author)

The authors have to be acknowledged for the timely and comprehensive revision of the manuscript. All relevant points raised by the reviewer have been sufficiently addressed and the manuscript is suitable for publication. Given the possibility of extensive inclusion of supplemental materials, the authors should consider including all data presented in the response to reviewer comments also in the manuscript.

Version 2:

Reviewer comments:

Reviewer #1

(Remarks to the Author)

The authors have adequately addressed my points.

Responses to Reviewers for Kwon et al., 2024

Reviewer #1 (Remarks to the Author)

In this manuscript, Kwon and colleagues study the function of the orphan receptor GPRC5B in macrophages and elucidate the underlying mechanisms. The authors first showed that the knockout of GPRC5B enhanced the migration, phagocytosis and activity in resident peritoneal macrophages (RPM, which have highest expression of GPRC5B) and bone marrow-derived macrophage (BMDM) isolated from myeloid-specific knockout mice, as well as bacterial clearance in a peritonitis model. The authors then showed that GPRC5B interacted with several GPCRs including EP2 in co-IP and that manipulation of GPRC5B expression levels affected EP2 signaling and plasma membrane localization in different cell types. Finally, they generated several GPRC5B mutants to analyze EP2 interaction with GPRC5B, signaling and subcellular localization, and synthesized a peptide derived from the second transmembrane domain of GPRC5B to disrupt GPRC5B effects on EP2 function.

Overall, the phenotypic effects of GPRC5B knockout on macrophages are clear. However, the mechanistic studies are not well designed and the experimental data presented in the manuscript do not strongly support the major conclusion that the function of GPRC5B in macrophages is mediated through interaction/dimerization with EP2 to facilitate EP2 signaling.

We thank the reviewer for the helpful feedback and will address all major and minor points in the following section.

Major points:

1. GPRC5B-EP2 interaction:

1a. Based on the co-IP results shown in Fig.3A and Suppl. 6A, it is premature to conclude that GPRC5B and EP2 form dimers or interact, although different GPCR members were compared. GPCRs with seven transmembrane domains are very sticky and they easily form nonspecific aggregates, particularly when overexpressed.

We agree that unspecific interactions pose a major problem when studying GPCR dimerization, but there are four aspects that support the notion that the co-immunoprecipitation between GPRC5B and EP2 is specific:

- A. GPRC5B interacts only with a specific subset of class A receptors: As the reviewer already alluded to, only a specific subset of prostanoid receptors co-precipitates with GPRC5B (IP, EP1, EP2, DP1), while others (EP3, EP4, DP2, FP, TP) do not (Fig. 1A below; same as manuscript Fig. 3A). Our input and pulldown controls show that this interaction pattern is neither due to poor expression in the input material nor inefficient pulldown in individual samples, indicating that GPRC5B indeed co-precipitated only with specific prostanoid receptors. We also employed the reciprocal approach and precipitated HA-tagged prostanoid receptors, and this resulted in the same pattern of interaction (Fig. 1B below, from Suppl. Fig. 6B). Furthermore, an analysis of potential interactions between GPRC5B and chemokine receptors (requested by Reviewer 2) showed that chemokine receptor CCR5 does not co-IP with GPRC5B (Fig. 1C below).
- B. EP2 interacts only with specific class C orphans: We also addressed the issue of specificity from the viewpoint of EP2 and investigated which of the GPCRs most closely related to GPRC5B co-precipitate with EP2. We found that pulldown of EP2 co-precipitated the GPRC5 family members B and C, but not with the closely related class C orphan GPR156 (Fig. 1D).

Figure 1: Western blot detection of HA and FLAG signals in lysates of HEK cells expressing FLAG-tagged receptors (GPRC5B in A-C; various class C orphan GPCRs in D) in combination with different HA-tagged prostanoid or chemokine receptors before (“input”) and after immunoprecipitation of FLAG-tagged receptors (“Pulldown FLAG” in A and C) or immunoprecipitation of HA-tagged receptors (“Pulldown HA” in B and D). Receptor FPR1 failed to be expressed in C and interaction therefore could not be determined.

- C. The interaction between GPRC5B and EP2 can be modulated by mutation and decoy peptides. Based on our *in silico* modelling data, we identified residues crucial for the interaction between GPRC5B and EP2 and show that mutation to alanine reduces co-immunoprecipitation for some of them (Fig. 2A, from manuscript Fig. 6F). We have now quantified the effect of the different mutations in three independent experiments and the results confirm that both “G5B-mut7” and “G5B-mut_set1” significantly reduce co-IP with EP2 (Fig. 2B). Furthermore, we show that decoy peptides designed to mimic the interacting region reduce co-IP, whereas control peptide does not have a clear effect (Fig. 2C, from manuscript Fig. 7B). Also for these data we have added a quantification of mutation-induced changes (Fig. 2D). Of note, both mutation and decoy peptide do not only prevent the interaction as determined by co-IP, they are also accompanied by corresponding functional changes on the level of cAMP production and macrophage function (see manuscript Figs. 6G-I and 7C-G).

D. The interaction between GPRC5B and EP2 is also observed for endogenously expressed EP2-GPRC5B: Since protein overexpression indeed constitutes a rather artificial system, we also performed co-IP experiments in primary mouse macrophages and – following a suggestion from Reviewer 2 - have now included GPRC5B-KO macrophages as an additional specificity control (Fig. 2E). These data show that even in primary cells, in which of endogenous GPCR expression is rather low, GPRC5B can co-precipitate with EP2.

Figure 2: A-D, Effect of GPRC5B mutations (A,B) or decoy peptides (C,D) on the co-immunoprecipitation of FLAG-tagged GPRC5B with HA-tagged EP2: **A,C**, Representative immunoblots shown in the manuscript; **B,D**, Densitometric analysis of signal strength for co-precipitated GPRC5B-FLAG (FLAG(coIP)) relative to immunoprecipitated HA-EP2 (HA(IP)) from 3-4 independent experiments. **E**, Western blot detection of EP2 and GPRC5B (G5B) signals in lysates of RPM before (“input”) and after immunoprecipitation of EP2 using anti-EP2 antibodies (“Pulldown EP2”) (IgG and GPRC5B-KO as negative controls).

Data are means \pm SEM, comparisons between different groups were performed using one-way ANOVA with Dunnett’s correction for multiple testing. *, $P < 0.05$; **, $P < 0.01$.

Taken together, these data show in our view clearly that the interaction between GPRC5B and EP2 does not reflect simple “stickiness” of either EP2 or GPRC5B. We have added the new co-IP quantifications to the revised manuscript in Supplemental Figures 10E and 11A and mention them on pages 10, line 35, and page 11, line 20. Data showing lack of interaction with chemokine receptors are shown in Supplemental Figure 6A and are described on page 8, line 2. The improved endogenous co-IP analysis in macrophages is replacing the old Figure 4A.

1b. Fig. 3K showed the strong colocalization of GPRC5B and EP2. As both are plasma membrane proteins or partially localized at the plasma membrane, this data does not support their dimerization or interaction.

Indeed, colocalization at the membrane does not prove dimerization or interaction, but it is a prerequisite for it and should therefore – in our view - be shown. The hypothesis of physical interaction relies on our coimmunoprecipitation data under basal conditions and in the presence of mutations or decoy peptides (see our response above), and the colocalization data are only of supportive nature. We do hope that our wording in the main text makes this sufficiently clear (“In line with a physical interaction between GPRC5B and EP2, we found that in HEK cells co-transfected with GPRC5B-FLAG/Myc and HA-EP2, HA and FLAG signals showed overlapping intracellular distribution (Fig. 3K).”)

1c. GPRC5B-mut set3 interacted with EP2, but did not affect EP2 signaling (Fig. 6F and 6G). One could argue that the interaction between the two receptors is not absolutely required for GPRC5B to affect EP2. (Same question from reviewer #2)

This is a valid point, the data indeed suggest that GPRC5B-mut set3 has lost the ability to facilitate EP2-dependent cAMP production even though co-IP and EP2 membrane localization are normal. This suggests that the various residues identified in our *in silico* analysis have different relevance:

- Set1 residues are essential for the physical interaction with EP2, their mutation results in loss of interaction, reduced EP2 membrane localization and reduced EP2 signalling.
- Mutation of set2 residues does neither affect co-IP nor EP2 localisation and signalling.
- Mutation of set3 residues is not sufficient to completely break the interaction between GPRC5B and EP2 and does not affect intracellular EP2 trafficking, but still impairs the ability of GPRC5B to facilitate EP2-dependent cAMP production. This indicates that GPRC5B can affect EP2 function beyond regulation of intracellular trafficking, but the exact nature of this facilitation is currently unclear. Studies in other GPCR dimers suggested that the interaction between the protomers may modulate ligand binding or effector coupling,^{1,2} and it seems possible that set3 residues contribute to such effects.

Taken together, we believe that our results for GPRC5B-mut set1 clearly show that complete loss of GPRC5B/EP interaction is associated with altered trafficking and reduced EP2 signalling, but we agree that GPRC5B might modulate EP2 responses by additional, yet unknown mechanisms. We have added a corresponding statement to the discussion (page 13, line 23ff): “Interestingly, mutation of amino acids Q116, D118, L128 (set 3) reduced GPRC5B-mediated facilitation of EP2 signaling without breaking the physical interaction and without altering EP2

membrane localization, suggesting that GPRC5B might influence EP2 signaling through additional, yet unknown mechanisms.”

1d. *The disruption of the peptide on the GPRC5B-EP2 interaction in co-IP was not impressive (Fig. 7B).*

It is true that the decoy peptide did not result in a full abrogation of co-IP, but the reduction is clear, reproducible and significant. This is, in our view, quite an accomplishment given the challenges associated with *in silico* modelling and docking of a receptor such as GPRC5B, for which the crystal structure has not been resolved and which has no strong homology to GPCRs with known structure. In view of this, we were pleased to see that GPRC5B mutants and decoy peptides were indeed able to modulate the interaction, and to make this point more clear we have quantified the effect of target and control peptides on co-immunoprecipitation of GPRC5B-FLAG/Myc with HA-EP2 in HEK cells in three independent experiments (Fig. 3):

Figure 3: Densitometric analysis of signal strength for co-precipitated GPRC5B-FLAG (FLAG(coIP)) relative to immunoprecipitated HA-EP2 (HA(IP)) from 3-4 independent experiments as shown in A and C. Quantification of the effect of target and control peptides on co-immunoprecipitation of GPRC5B-FLAG with HA-EP2 in HEK cells (n=3).

Data are means \pm SEM, comparisons between different groups were performed using one-way ANOVA with Dunnett’s correction for multiple testing. *, $P < 0.05$.

The quantification of control and target peptides on co-IP are now shown in Suppl. Fig. 11A of the revised manuscript.

2. *Effect of GPRC5B on EP2 localization:*

2a. *Throughout the manuscript, the effects of GPRC5B on the surface expression or subcellular distribution of EP2 as quantified using WGA or CD11 as markers were relatively weak (albeit statistically significant).*

It is true that knockdown or overexpression of GPRC5B does not result in excessive changes of EP2 membrane availability, but they are robust, significant, and associated with corresponding changes on the signalling level. In HEK cells, for example, overexpression of GPRC5B almost triples the amount of EP2 that co-localizes with WGA (see manuscript Fig. 5E) and butaprost-

induced cAMP production – depending on the concentration – is increased by a factor of 2-4 (see manuscript Fig. 3F). In RPM, inactivation of GPRC5B results in reduction of surface EP2 availability by 20-50% (Figs. 5F, 5H-K), and butaprost-mediated effects are strongly reduced (see manuscript Figs. 4B-F). However, we can at this point not exclude that other factors than regulation of membrane localization contribute to GPRC5B-mediated facilitation of EP2 signalling (see also our response to question 1c), and to acknowledge this we have added the following statement to the discussion (page 13, line 23ff):

“Interestingly, mutation of amino acids Q116, D118, L128 (set 3) reduced GPRC5B-mediated facilitation of EP2 signaling without breaking the physical interaction and without altering EP2 membrane localization, suggesting that GPRC5B might influence EP2 signaling through additional, yet unknown mechanisms.”

2b. From images presented in manuscript, it is difficult to see clear effects of GPRC5B on EP2 subcellular localization. For example, in Fig. 5H, it seems more EP2 staining (red) in GPRC5B knockout cells as compared with control cells.

As in all figures, the images depicted in Fig. 5H were chosen because they are close to the average of their respective group and we feel that – especially at high magnification – the reduced EP2 immunoreactivity within the CD11b-positive membrane area of KO RPM can be appreciated. However, we followed the reviewer’s suggestion and investigated whether there is more EP2 staining in GPRC5B KO RPM, but found this not to be the case (Fig. 4).

Figure 4: Statistical evaluation of EP2 signal strength in individual RPM from control mice and M-G5b-KOs (n=20-22). Data are means \pm SEM, comparisons between groups were performed using unpaired t test.

2c. Where is EP2 normally expressed in cell? Fig. 5D, 5J and 6H showed apparent different intracellular distribution of EP2.

This is an important question, and we observed certain differences between the localization of endogenously expressed EP2 receptor in primary macrophages and overexpressed HA-tagged EP2 in HEK cells.

In permeabilized macrophages, endogenous EP2 was detected both at the (CD11b-positive) membrane and in intracellular organelles (Fig. 5A), and these organelles were in

further co-stainings identified as Golgi apparatus (marker GM130, Fig. 5A) and ER (marker calreticulin).

Figure 5: Five examples showing the intracellular localization of endogenous EP2 (red) relative to the CD11b-stained plasma membrane or Golgi marker GM130 in wildtype RPM.

In HEK cells, overexpressed EP2 was also found in the WGA-stained membrane, but intracellular staining was here more dominant (Fig. 6A). Furthermore, the pattern of cytosolic staining was more variable: in most cases we observed a broad cytoplasmic distribution close to the membrane (examples 1-3 in Fig. 6A), but in some cases also punctate or perinuclear patterns were observed (examples 4+5 in Fig. 6A). We believe that the stronger and more diverse intracellular localization in HEK cells is due to the forced overexpression of EP2, which results – depending of the strength of overexpression in individual cells – in different degrees of organelle overload and consecutive protein mislocalization. However, our studies in overexpressing HEK cells were designed to study the effect of GPRC5B on EP2 membrane localization, and these analyses were not affected by differences in intracellular distribution. We nevertheless agree with the reviewer that it would be better to show in the figures only the most representative EP2 distribution type, which is in this case the “broad” cytoplasmic distribution shown in examples 1-3. We therefore replaced the “punctate” example in Fig. 5D by a more representative picture for condition “HA-EP2 alone” (Fig. 6B and Fig. 5D of revised manuscript).

Figure 6: A, Five examples showing the intracellular localization of overexpressed, HA-tagged EP2 (red) relative to the WGA-stained plasma membrane in wildtype RPM. **B**, New version of manuscript Fig. 5D, now displaying cells with more representative EP2 staining pattern.

2d. The image shown in the top right of Fig. 6C is exactly the same as one shown in the top panel 2nd left of Fig. 6H.

It is true that the same cell is shown in Fig. 6C (where the distribution of the GPRC5B-Myc signalling relative to WGA is the focus) and in Fig. 6H (where the distribution of EP2-HA relative to WGA is the focus). However, to avoid unnecessary confusion we have replaced the cell exemplifying G5B-wt in Fig. 6C by another cell (Fig. 7 and Fig. 6C of the revised manuscript).

Figure 7: New version of Fig. 6C, in which the cell representing G5B-wt has been replaced.

2e. The effects of the peptide on the surface expression and subcellular location of EP2 were not studied.

Following the reviewer's suggestion we analysed and quantified the subcellular localization of EP2 in RPM under basal conditions and after incubation with DMSO alone or different peptides. We found that while control peptide did not cause significant changes, target peptide reduced colocalization of EP2 with membrane marker CD11b and increased colocalization with Golgi marker GM130 (Fig. 8A,C). Colocalization with ER marker calreticulin was increased in target peptide cells compared to basal cells, but not in compared to control peptide treated cells (Fig. 8B,C). We have added these data to Suppl. Fig. 11B-D of the revised manuscript and describe them on page 11, line 23ff

Figure 8: A-B, Effect of target and control peptides on EP2 localization in RPM: wildtype RPM were treated with solvent (basal), control peptide, or target peptide (1 μ M each) for 1 hour, then cells were stained with anti-CD11b antibodies (for plasma membrane), permeabilized, and stained either for EP2(ec) and Golgi marker GM130 (A) or EP2(ec) and ER marker calreticulin (B). **C,** Statistical evaluation of the colocalization of EP2(ec) with CD11b, GM130 or calreticulin (n=11-18 cells per condition).

Data are means \pm SEM, comparisons between groups were done by two-way ANOVA with Sidak's post hoc test (C). *, $P < 0.05$; ***, $P < 0.001$; ****, $P < 0.0001$.

3. Effect of GPRC5B on EP2 signaling:

3a. It is interesting to note that GPRC5B knockdown inhibited the signaling via EP2, but not EP1 and EP4. However, this inhibitory effect was no more than 20% (Fig. 3D)

It is true that the reduction is here only around 20% (Fig. 9A, same as manuscript Fig. 3D), which might be due to the fact that these experiments were done in EP2-overexpressing cells, i.e., we are studying the impact of a relatively low number of endogenously expressed GPRC5B molecules on a significantly larger number of overexpressed EP2 receptors. It is unfortunately not possible to study GPRC5B knockdown without EP2 overexpression, since endogenous EP2 responses are very low in HEK cells. We therefore corroborated our findings in macrophages, which express higher levels of endogenous EP2 and GPRC5B, and found that GPRC5B-deficiency had very strong effects on butaprost-induced cAMP production both in RPM and BMDM (Fig. 9B,C, taken from manuscript Figs. 4B and Suppl. Fig. 7A).

Figure 9: Consequence of GPRC5B knockdown or knockout for butaprost-induced cAMP production in EP2-overexpressing HEK cells (A) or primary RPM and BMDM (B,C). Data are means \pm SEM; comparisons between groups were done using two-way ANOVA and Sidak's post-hoc test. ***, $P < 0.001$; ****, $P < 0.0001$.

3b. The effect of GPRC5B overexpression on basal cAMP production was quite dramatic (Fig. 3F). How to explain?

We indeed detected in some experiments a strong increase in basal cAMP production upon GPRC5B expression. Interestingly, this was mainly observed when using the cAMP-Glo assay (as in Fig. 3F), while the effect was relatively weak when using ELISA (as in Fig. 3I). Fig. 10 shows a paired analysis of all existing data (each pair one experiment) for the two methods (Fig. 10A,B). Though significant in both methods, the increase is clearly stronger when using the Glo assay, and this difference may be related to the longer IBMX pretreatment (2 h in Glo assay versus 30 min in ELISA), resulting in the amplification of an initially mild effect.

The mechanisms underlying the basal cAMP increase as such are currently unknown, but one hypothesis would be endogenous production of PGE₂ by HEK cells. We used existing RNA sequencing data to check whether HEK cells express genes related to PGE₂ synthesis and indeed found expression of different phospholipase A2 isoforms (*PLA2G*), cyclooxygenases

COX1 and COX2 (*PTGS1*, *PTGS2*), and prostaglandin E synthases (*PTGES*, *PTGES2*, *PTGES3*) (Fig. 10C). However, at this point we also cannot exclude that GPRC5B facilitates agonist-independent EP2 activation.

Figure 10: **A**, Basal cAMP production in HEK cells transfected with a cAMP GloSensor plasmid, EP2 receptor, and either empty vector (EV) or GPRC5B (G5B) (n=14). **B**, Basal cAMP production in HEK cells transfected with EP2 receptor and either empty vector (EV) or GPRC5B (G5B) was determined by ELISA (n=8). **C**, Expression of genes related to PGE₂ synthesis was determined in EV-transfected HEK cells (n=3).

Data are means \pm SEM; Differences between groups were determined using paired t test (A,B). *, $P < 0.05$;

3c. The data shown in Fig. 4J-L are confusing. Although RA stimulation produced opposite effects on the expression of GPRC5B and EP2, it would also effect the expression of many other proteins.

It is true that RA affects numerous processes, of which upregulation of GPRC5B (Fig. 4J) and downregulation of EP2 (Fig. 4K) are only two. However, in Fig. 4L we specifically investigate butaprost-induced cAMP production, a process that will almost exclusively be affected by the membrane availability of EP2 and the presence of $G\alpha_s$ and adenylyl cyclases, respectively (phosphodiesterases are in the assay inhibited by IBMX). According to Balmer and Blomhof,³ neither *Gnas* nor adenylyl cyclases are regulated by RA, suggesting that improved butaprost effects 24 h after RA treatment might indeed be related to increased GPRC5B-mediated facilitation of remaining EP2 receptors. We agree, however, that showing changes on the transcriptional is not sufficient to make this point and have therefore added western blotting data for RA-induced regulation of GPRC5B and EP2 (Fig. 11).

Figure 11: A-D, Effect of RA (1 μ M, 24h) on expression of GPRC5B (A,B) or EP2 (C,D) was determined in BMDM by immunoblotting: immunoblots (A,C) and densitometric analysis of signal intensity (B,D), GAPDH as loading control (n=3).

Data are means \pm SEM; Differences between groups were determined using unpaired t test. ***, $P < 0.001$.

These new data are shown in shown in Suppl. Figs. 8A-D of the revised manuscript.

4. Minor points:

4a. What genotypes revealed in Suppl. Fig. 1H-J?

The colour coding for the genotypes shown in Suppl. Fig. 1 was indicated at the top of the figure, but to avoid confusion we have duplicated the legend twice and now show it also in the middle and at the bottom of the supplemental figure.

4b. Why are different inflammation markers differentially regulated (Fig. 1E, 1F, Suppl. 1G)?

This is indeed an interesting point. In LPS-stimulated RPM, knockout of GPRC5B resulted mainly in upregulation of *Nos2*, whereas inflammatory genes such as *Tnf*, *Il6*, or *Il1b* were not significantly increased (Fig. 12A-D). It is in this context interesting to remember that in GPRC5B-deficient BMDM, in contrast, a broad facilitation of LPS-induced inflammatory gene expression was observed (Fig. 12E-H).

Figure 12: Expression of inflammatory genes in resting and LPS (1 μ g/ml, 6h)-stimulated RPM (A-D) and BMDM (E-H) was determined by qRT-PCR.

Data are means \pm SEM; comparisons between genotypes were performed using two-way ANOVA with Sidak's multiple comparison test; *, $P < 0.05$; ***, $P < 0.001$.

This raises two questions:

- 1) Why are inflammatory genes such as *Tnf/Il6/Il1b* increased in BMDM, but not in RPM?
- 2) Why is - despite otherwise normal inflammatory gene expression - *Nos2* expression increased in RPM?

We believe that these questions are interlinked and therefore would like to address them together. RPM and BMDM are developmentally and functionally very dissimilar, and literature suggests that they differ with respect to basal activation and responsiveness to inflammatory stimulation.⁴⁻⁷ This can also be seen in our qRT-PCR data above, in which BMDM show much stronger LPS responses than RPM (Fig. 12). To better visualize this point, we have plotted Δ Ct values as a measure of gene expression in basal and LPS stimulated BMDM and RPM (Fig. 13). These data do not only show that LPS-induced upregulation of *Il6*, *Il1b*, and *Tnf* is weaker in RPM (green) than in BMDM (red), but also that RPM are already in the basal state on a significantly higher gene expression level, especially for *Il6* and *Il1b*.

Figure 13: Expression of inflammation markers in basal and LPS-stimulated BMDM (red) and RPM (green) was analyzed by qRT-PCR (data displayed as Δ ct values (ct value *target gene* – ct value *Gapdh*)). Data are means \pm SEM; comparisons between genotypes were performed using 2-way ANOVA with Tukey's posthoc test; ***, $P < 0.001$; ****, $P < 0.0001$.

As to the mechanism how GPRC5B regulates inflammatory gene expression in these two cell populations, we hypothesized an interference with NF- κ B activation: inflammation markers such as *Nos2*, *Tnf*, *Il6*, and *Il1b* are mainly regulated by NF- κ B,⁸ and elevated cAMP levels have been shown to interfere with p65-NF- κ B activation.^{9, 10} To test whether NF- κ B activation is indeed enhanced in the absence of GPRC5B, we determined I κ B α degradation in BMDM and found it to be significantly lower in LPS (1h)-stimulated KO cells compared to control (Fig. 14A,B), indicative of increased NF- κ B activation. Interestingly, RPM showed hardly any I κ B α degradation after LPS exposure (Fig. 14C,D), suggesting that these cells – maybe due to their higher basal activation – are less responsive to LPS. This, in turn, might explain while loss of

GPRC5B-EP2-mediated suppression of NF- κ B activation is in these cells functionally less relevant and therefore no difference in *Il6*, *Tnf*, *Il1b* expression is observed.

Figure 14: A,B, LPS-induced degradation of NF- κ B inhibitor I κ B α was determined by western blotting in BMDM: representative immunoblots (A) and statistical evaluation (B), GAPDH as loading control. **C,D**, LPS-induced degradation of NF- κ B inhibitor I κ B α was determined by western blotting in RPM, GAPDH as loading control.

Data are means \pm SEM; comparisons between groups are performed using unpaired t test (B); **, $P < 0.01$.

However, even in the absence of strong NF- κ B activation some LPS-induced activation of inflammatory gene expression is observed in RPM, most prominently for *Nos2* (Fig. 12A above). The *NOS2* promoter is not only regulated by its NF- κ B binding site, but has several other regulatory components that are essential for complete expression of *NOS2*, for example binding sites for AP-1, signal transducer and activator of transcription-1 (STAT1), and IFN regulatory factor-1 (IRF-1).¹¹⁻¹³ In addition, epigenetic regulation of *NOS2* promoter accessibility by the NLRC4/caspase-1 axis has been suggested in macrophages.¹⁴ All these alternative regulators of *NOS2* expression are present in RPM, and though their role has not been specifically studied in RPM, it seems likely that they contribute to LPS-induced upregulation of *NOS2*. Interestingly, cAMP has been shown to interfere with the activation of STAT1 or AP-1 signalling,^{15, 16} and it is tempting to speculate that altered activation of these pathways might underlie increased *NOS2* expression in GPRC5B-deficient RPM. However, since we did not address these hypotheses experimentally, we feel that these consideration would be rather distracting for the reader and prefer not to address them in the revised version of the manuscript.

4c. In Fig. 5K, should WGA be CD11b?

That is true, the labelling was wrong and has been corrected in Fig. 5K as shown below (Fig. 15).

Figure 15: Corrected version of Figure 5K (CD11b instead of WGA).

4d. In Fig. 4H, Western blots should be shown.

Following the reviewer's suggestion we analysed GPRC5B expression after siRNA-mediated knockdown (G5B kd) or plasmid-mediated overexpression (G5B OE) in THP1 cells on the protein level (Fig. 16).

Figure 16: THP1 cells were transfected with plasmids and siRNAs as indicated and the effect on GPRC5B expression was determined by immunoblotting (GAPDH as loading control). EV, empty vector; EP, expression plasmid.

We are showing these new data in Suppl. Fig. 7E of the revised manuscript.

4e. The data presented in Fig. 4H-I are not correct. siRNA and overexpression experiments should have different control groups.

Regarding this point, we beg to differ. The aim of the experimental design was indeed to compare cells with siRNA-mediated knockdown and plasmid-based overexpression to the same control group, and for this purpose all three groups received both plasmid transfection

(EV or G5B expression plasmid) and siRNA transfection (siControl or siG5b) (the same strategy is shown in Fig. 16 above). This experimental design allows side by side comparison with on control group, and the method had been described in detail in the methods section on page 21, line 4ff. Additionally, we have now included the following statement in the legend for Fig. 4H-I: “All three groups were transfected both with plasmid and siRNAs (Control: EV+siControl; G5B kd: EV + GPRC5B siRNA; G5B OE: GPRC5B plasmid + control siRNA)”.

4f. In Fig. 5D, based on the figure legend, cells were stained with WGA, fixed, permeabilized and then stained with antibodies. It is confusing how WGA stained Golgi-like intracellular compartments?

It is indeed puzzling that WGA staining is under these experimental conditions sometimes found in intracellular compartments, but also others have observed WGA staining of vesicles/organelles in the absence of permeabilization.¹⁷⁻¹⁹ Whether this intracellular staining is due low amounts of WGA permeating the intact cell membrane or whether WGA that has initially been membrane-bound is released during fixation/permeabilization to then secondarily enter the cell, is currently unclear. However, we did not observe this phenomenon regularly, so we have replaced this particular cell in Fig. 5D by a more representative example (shown also in Fig. 17 below).

Figure 17: New version of manuscript Fig. 5D, now displaying cells with more representative EP2 and WGA staining pattern (see also our response to question 2c).

4g. In Fig. 5J, the images shown in control group have different orientation.

That is true, and we have corrected the problem as shown in Fig. 18 below.

Figure 18: Corrected version of Fig. 5J (pictures for control-EP2, control-GM130, and control-EP2+GM130 were turned by 90°).

Reviewer #2 (Remarks to the Author)

The manuscript by Kwon et al titled “Orphan G protein-coupled receptor GPRC5B controls macrophage function by facilitating prostaglandin E receptor 2 signaling” examines the function of the orphan GPRC5B in controlling immune cell function via dimerization with the EP2 receptor. The study utilizes a comprehensive approach to study two different populations of primary macrophages derived from wildtype and GPRC5B KO mice and human cells using both in vitro and in vivo approaches. The authors conclude that GPRC5B dimerizes with EP2 receptor in macrophages resulting in an increase in EP2 expression at the cell surface, enhanced anti-inflammatory signaling and responses. The data are robust including 7 Figures with 10 -16+ panels and 9 Supplementary Figures with multiple panels, at times the data seem excessive, which contributes to a lack of focus and strong rationale for the studies. For example, are the in vivo MI and colitis studies essential to the main conclusions of the paper? Other concerns relate the Introduction, which lacks critical relevant background information and Results section that lack strong rationale for studies, detailed interpretation, and conclusions as well as some issues with data analysis and lack of control experiments that provide a mechanistic understanding of GPRC5B regulation of EP receptor function.

We thank the reviewer for the critical feedback and will address all issues in the context of the reviewer’s specific comments below.

Comments.

1. Figure 1 shows GPRC5b expression is greater in RPM than BMDM, generation of GPRC5b KO mouse using LysM CRE specific for macrophages, neutrophils, granulocytes and impact on inflammatory signaling, migration etc. Panel D, is an immunoblot of GPRC5b protein expression and shows multiple bands ranging from 35 to 55 kDa. Are all the bands presumed to be the receptor? This is important since data shown in other Figures show only a small portion of the blot with only one band GPRC5b that migrates at the lower molecular weight 35 to 40 kDa and sometimes higher? The authors need to determine if all of the GPRC5b low and high molecular weight species are expressed on the cell surface and determine which GPRC5b form is dimerizing with EP2.

This is an important point, especially when considering the well-known problems with the specificity of many GPCR antibodies.²⁰ In the course of this project we tested four different commercial antibodies, none of which was able to differentiate control and knockout samples. Fortunately, Fu *et al.* reported a self-made GPRC5B antibody with improved specificity,²¹ and this antibody was used to generate all GPRC5B immunoblots shown in this manuscript. In RPM, we indeed observed two specific bands for GPRC5B, a dominant band between 40 and 55 kDa and a weaker band below 40 kDa (Fig. 19A, same as in manuscript Fig. 1D). In addition, there was an unspecific band that did not disappear in the KO samples. In this particular blot it was not easy to correctly position the molecular weight labels because the protein ladder was on the other side of the blot and the gel had run slightly askew, and to remedy this we ran new samples with ladders on both sides (Fig. 19B, C). Based on these new blots we estimate that the upper GPRC5B band is between 42-45 kDa and the weaker band at 36-37 kDa (Figs. 19B, C). Fu *et al.*²¹ observed with the same antibody two bands of comparable size in mouse brain, and the authors showed that the upper band disappeared after deglycosylation (Fig. 19D). We therefore assume that also in our samples the upper band is glycosylated GPRC5B.

To address these issues in the text, the legend of Fig. 1D has been amended as follows: “Knockout efficiency in RPM was analyzed by immunoblotting (unspecific band around 38 kDa; the higher of the two specific bands probably represents glycosylated GPRC5B;²¹ GAPDH as loading control).” Furthermore, the protein ladder labeling in Fig. 1D has been slightly adjusted to better reflect the true molecular weight of the different bands.

[panel d is redacted]

Figure 19: A-C, Lysates of control and GPRC5B-deficient RPM were analyzed by immunoblotting using the antibody described by the group of C. Orlandi (Fu et al.²¹, GAPDH as loading control). **D**, Figure 2E from Fu et al.²¹: Brain tissue lysates in the native state (CTRL) and after deglycosylation (DEGLYC) were analysed by immunoblotting using GPRC5B antibodies. Glycosylated forms located above 37 kDa collapse to the unmodified forms after the treatment.

As to the question which of the two GPRC5B forms co-precipitates with EP2, we revisited the existing data and noticed an unfortunate mislabelling in Fig. 4A of the original manuscript: the bands of the molecular weight ladder seen in the GPRC5B blots are not 40 and 35 kDa, but 55 and 40 kDa (Fig. 20A). These data indicate that the upper band co-precipitates with EP2, but whether this is also true for the lower band cannot be judged based on these blots. However, this reviewer suggested in question 4a to perform endogenous co-IP in macrophages from control and M-G5b-KO mice, and we used this opportunity to re-evaluate the role of the lower GPRC5B band in co-IP. As seen previously, input lysates showed a dominant band at 42-45 kDa and a weaker band at 36-37 kDa (in addition the unspecific band around 38 kDa). After pulldown of EP2, both the upper and the lower band were detected in WT cells, whereas in KO cells no bands were found, even though EP2 pulldown was efficient (Fig. 20B). These data indicate that both GPRC5B bands co-precipitate with EP2 and we show the improved endogenous co-IP data in Fig. 4A of the revised manuscript. The issue of the different molecular weight species is addressed in the legend for Figure 1D.

Figure 20: A, Original, wrongly labelled (left) and corrected (right) version of the endogenous pulldown data shown in Fig. 4A of the previous version of the manuscript. B, Western blot detection of EP2 and GPRC5B (G5B) signals in lysates of RPM before (“input”) and after immunoprecipitation of EP2 using anti-EP2 antibodies (“Pulldown EP2”) (IgG and GPRC5B-KO as negative control).

Panels J and L are missing statistical analysis.

Unpaired t tests were performed for panels J and L, but the difference was not significant and is therefore not indicated. A description of the statistical tests employed can be found in the end of each figure legend.

Overall, the description of the data this figure needs to be improved, it lacks clear interpretation of the findings and conclusions.

We added the following statement summarizing the findings of figure 1 on page 5, line 32ff: “Taken together, GPRC5B-deficient RPM are characterized by increased migration and phagocytosis, resulting in improved bacterial clearance in a peritonitis model.”

2. Figure 2.

2a. The rationale for studying BMDM in a similar manner RPM is not clear.

BMDM and RPM are developmentally and functionally very different populations, and phenotypes observed in one cell type are not necessarily present in the other.^{5-7, 22, 23} Our data show that hyperactivity of GPRC5B-deficient RPM is beneficial in the context of peritonitis, but the peritoneal FACS analyses indicated that also BMDM recruitment is enhanced. It therefore seemed both interesting and relevant to also characterize the role of GPRC5B in BMDM, and our *in vitro* studies confirmed the hyperreactivity phenotype in these

cells. The *in vivo* consequences of such BMDM hyperreactivity are very hard to predict and might range from beneficial effects such as improved pathogen clearance to detrimental consequences such as prolonged or misguided inflammatory responses or aggravation of autoimmune disorders.²⁴ To shed a little light on the consequences of macrophage hyperreactivity in the absence of GPRC5B, we therefore employed two *in vivo* models that involve damage-induced monocyte/macrophage recruitment and activation, myocardial infarction and dextran sulfate sodium (DSS)-induced colitis. We tried to summarize these considerations in the following two explanatory statements in the main text (page 6):

“The increased recruitment of monocyte-derived macrophages during bacterial peritonitis suggests that also BMDM function is altered in M-G5b-KOs, which led us to study consequences of GPRC5B deficiency in these cells.”

“Altered responsiveness of BMDM might affect immune responses beyond acute defense against peritoneal pathogen invasion, which led us to study M-G5b-KOs in two disease models that involve damage-induced monocyte/macrophage recruitment and activation, myocardial infarction and dextran sulfate sodium (DSS)-induced colitis.
31,32”

2b. Panel F, multiple GPCR agonists enhance GPRC5b KO macrophage migration and raises the question of whether GPRC5b regulates expression of the GPCRs via dimerization.

We agree that it would be interesting to investigate whether also non-prostanoid receptors, for example chemokine receptors, dimerize with GPRC5B. To test this at least for some of the chemokines used in the migration assay in Fig. 2F, we obtained HA-tagged expression plasmids for CCR5 (receptor for CCL5) and FPR1 (receptor for f-MLP) and investigated whether they co-immunoprecipitated after pulldown of FLAG-tagged GPRC5B. While HA-tagged EP2 co-precipitated as expected, CCR5 clearly failed to do so (Fig. 21), suggesting that impaired CCL5-induced migration is not due to direct modulation of CCR5 function by GPRC5B. Receptor FPR1 unfortunately was not expressed in HEK cells and its interaction with GPRC5B can therefore not be determined.

Figure 21: Western blot detection of HA and FLAG signals in lysates of HEK cells expressing GPRC5B-FLAG in combination with different HA-tagged receptors before (“input”) and after immunoprecipitation of FLAG-GPRC5B (“Pull-down FLAG”). Receptor FPR1 failed to be expressed.

We have included these data as Suppl. Fig. 6A of the revised manuscript and describe them on page 8, line 2.

2c. Panels I – J, show MI and colitis model study results. It is unclear why these studies are critical to the main conclusions of study.

Our analyses in the peritonitis model show that hyperactivity of GPRC5B-deficient macrophages is under these circumstances beneficial, but this is not necessarily the case in other conditions. Macrophage hyperactivity might result in prolonged or misguided inflammatory responses or aggravation of autoimmune disorders,²⁴ and we believe that is important to determine the *in vivo* consequences of macrophage hyperactivation in a variety of models. However, we agree that only key data should be presented in the main figures and therefore a considerable part of the *in vivo* data is presented in the supplement.

3. Figure 3.

3a. Panel A, HEK293 cells and co-IP are used to demonstrate GPRC5b interaction with EP1, EP2 and DP1 receptor. The immunoblots of GPRC5b show a band around 40 kDa. Is this appropriate size for GPRC5b? Is the GPRC5b 40 kDa form expressed at the cell surface and colocalized with EP2 at the plasma membrane

The predicted size of human GPRC5B-FLAG is 44.8 kDa, but we indeed observed the FLAG signal in GPRC5B-FLAG-transfected HEK cells closer to 40 kDa, though the gels did not run long enough to allow detailed evaluation of the molecular weight. A similar difference between predicted and observed molecular weight was seen with plasmids encoding murine GPRC5B, and we have in both cases confirmed cDNAs by DNA sequencing. The reason for the lower apparent weight is not fully clear, but it is tempting to speculate that a cleavable signal peptide predicted for the N-terminus might play a role (<https://services.healthtech.dtu.dk/services/SignalP-5.0/>). Removal of this signal peptide would reduce the predicted molecular weight to 41.6 kDa, which seems reasonably close to the observed molecular weight.

As to the cellular localization of GPRC5B-FLAG, we found it to be expressed at the cell surface where it co-localizes with EP2 (Fig. 22 and Fig. 3K of the revised manuscript).

Figure 22: Spatial correlation between HA and FLAG signals in HEK cells co-transfected with HA-EP2 and GPRC5B-FLAG/Myc: exemplary photomicrographs and statistical analysis (Pearson's coefficient, n=29 cells).

3b. – in panel D, an apparent non-specific bands from RPM GPRC5b KO lysates migrates at 40 kDa.

We are not sure which figure the reviewer is referring to, Fig. 3D does not show immunoblotting data. Maybe the reviewer is referring to Fig. 1D? If so – yes, we indeed observed an apparently unspecific band around 38 kDa, which did not disappear in the KO. We have included a corresponding statement in the Figure legend for Fig. 1D: “Knockout efficiency in RPM was analyzed by immunoblotting (unspecific band around 38 kDa; the higher of the two specific bands probably represents glycosylated GPRC5B²¹; GAPDH as loading control).”

In panel H and other panels throughout the paper, the statistical analysis comparisons should be between control (si or WT) with GPRC5B si or KO with and without agonist. The conclusions for this set of results are missing.

Following the reviewer's suggestion we now analyse both genotype effect and treatment effect in Figures 3G, 3H and 3J (Fig. 23).

Figure 23: Data from main text Figures 3G (A), 3H (B), and 3J (C) have been reanalysed using two-way ANOVA and Sidak's post-hoc test to show not only differences between treatments, but also between genotypes. *, $P < 0.05$; **, $P < 0.01$; ***, $P < 0.001$; ****, $P < 0.0001$.

In figures that mainly focus on the comparison of genotypes, however, we would suggest to keep the existing analyses in order not to overload figures with statistical information.

We added a statement summarizing the findings of Fig. 3 on page 8, line 21 (“Taken together, GPRC5B interacts with EP2 and facilitates EP2-dependent cAMP production in HEK cells”).

4. Figure 4.

4a. Panel A the GPRC5b band appears as a doublet migrating at 40 kDa? The IgG pulldown lane for G5B looks concerning. The authors should include co-IP from WT and GPRC5b KO mice RPM.

As already indicated in our response to question 1 of this reviewer, the protein ladder in manuscript Fig. 4A was unfortunately mislabelled, and the corrected version can be seen below in Fig. 24A. This corrected version indicates that the co-precipitated form of GPRC5B runs between 40 and 55 kDa, and though the band is indeed a bit fuzzy, closer inspection did not truly reveal a double band (magnification in Fig. 24B). We also followed the reviewer's suggestion to perform endogenous co-IP in parallel in macrophages from control and M-G5b-KO mice and found that no GPRC5B signal was detected in the KO samples, even though EP2 pull-down was efficient (Fig. 24C). We show these improved data for the endogenous co-immunoprecipitation in Fig. 4A of the revised manuscript.

Figure 24: **A**, Corrected version of old Figure 4A, in which the protein ladder was partially mislabelled (corrections in red). **B**, Magnification of the G5B input blot shown in A. **C**, Western blot detection of EP2 and GPRC5B (G5B) signals in lysates of control and KO RPM after immunoprecipitation of EP2 using anti-EP2 antibodies (“Pull-down EP2”) (IgG as negative control).

4b. Similar studies should be performed in THP1 human macrophages with siRNA GPRC5 knockdown as a control.

Following the reviewer's advice we tried endogenous co-IP in THP1, but failed. We believe that this is due to the much weaker expression of GPRC5B in THP1 compared to RPM (see Fig. 25 for a comparison of GPRC5B signals in THP1 vs RPM). We also cannot exclude that the GPRC5B antibody used in this study recognizes mouse GPRC5B more efficiently than human GPRC5B.

Figure 25: Comparison of GPRC5B immunoreactivity in lysates of THP1 cells (A) and RPM (B). GAPDH as loading control.

4c. In panel B, as a control the authors should show that *GRPC5b* KO does not affect EP2 expression.

This is of course an important aspect, and we determined whether loss of GPRC5B affects EP2 expression on RNA or protein level. As shown in Figure 26, neither qRT-PCR nor immunoblotting revealed differences in *Ptger2*/EP2 expression in RPM or BMDM between the genotypes.

Figure 26: A,B, *Ptger2* expression in RPM (A, n=12)) and M0 BMDM (B, n=15) was determined by qRT-PCR (data normalized to *Gapdh*, control set to 1). C-E, EP2 expression was determined by western blotting in RPM (C) and M0 BMDM (D) lysates from control mice (Contr) and M-G5b-KOs (KO) (representative blots, GAPDH as loading control). E shows the densitometric analysis of band intensity after normalization to GAPDH (n=5-8).

Data are means \pm SEM; comparisons between genotypes were done unpaired t test. *, $P < 0.05$; **, $P < 0.01$; ***, $P < 0.001$; ****, $P < 0.0001$.

These data are shown in Figs. 5A-C and Suppl. Fig. 9A of the revised manuscript (the immunoblots are shown in a reduced version for the sake of space).

5. Figure 5.

5a. Panel C data of the EP2 receptor immunoblot need to be quantified from at least 3 independent experiments because the control EP receptor blots looks higher than EP2 expression in KO.

As requested by the reviewer, we quantified EP2 receptor expression in immunoblots both for RPM and BMDM and found that expression did not differ between the genotypes (Fig. 27).

Figure 27: EP2 expression in RPM and BMDM was quantified in 5-8 independent experiments relative to GAPDH as loading control.

Data are means \pm SEM; comparisons between genotypes were done unpaired t test. *, $P < 0.05$; **, $P < 0.01$; ***, $P < 0.001$; ****, $P < 0.0001$.

The quantification has been added as Suppl Fig. 9A. Furthermore, we have replaced the misleading examples in Fig. 5C by more representative blots.

5b. Panel D, why is WGA staining appear to be inside the cell.

It is indeed puzzling that WGA staining is under these experimental conditions sometimes found in intracellular compartments, but also others have observed WGA staining of vesicles/organelles in the absence of permeabilization.¹⁷⁻¹⁹ Whether this intracellular staining is due low amounts of WGA permeating the intact cell membrane or whether WGA that has initially been membrane-bound is released during fixation/permeabilization to then secondarily enter the cell, is currently unclear. However, we did not observe this phenomenon regularly, so we have replaced this particular cell in Fig. 5D by a more representative example (shown also in Fig. 28 below).

Figure 28: New version of manuscript Fig. 5D, now displaying cells with more representative EP2 and WGA staining pattern.

5c. Panel K, images of calreticulum staining are missing.

Representative stainings for calreticulin are presented in Supplemental Figure 9C and Fig. 29 below:

Figure 29: RPM from control and KO mice were stained with CD11b (for plasma membrane), then permeabilized and stained for EP2(ec) and ER marker calreticulin. Shown are exemplary photomicrographs, statistical evaluation is in Fig. 5I (n=10).

6. Figure 6.

6a. Panel A, how valid is the homology model of GPRC5b-EP2 receptor, can the authors provide the PLDDT (predicted local distance values).

IDDT (Local Distance Difference Test, <https://www.ncbi.nlm.nih.gov/pmc/articles/PMC3799472/>) is calculated to measure the accuracy of a model compared to the reference (ground truth) structure. Since the reference structure is usually not available, the quality of AlphaFold models is reported as pLDDT (Predicted Local Distance Difference Test), which is a score that predicts the actual IDDT.

As an alternative metric to AlphaFold's pLDDT, the quality of homology models of the monomers can be assessed using QMEANDisCo (Qualitative Model Energy ANalysis Distance Constraints, <https://doi.org/10.1093/bioinformatics/btz828>). QMEANDisCo scores range from 0 to 1, with higher scores indicating better agreement with known high-quality protein structures. The global QMEANDisCo scores for the EP2 and GPRC5B models are 0.67 and 0.62, respectively, while the values for the transmembrane domains are 0.81 and 0.73, respectively. As mentioned, the EP2 crystal structure is available, and the model was used solely to fill the gaps.

These facts have been added to the Methods section on page 33, line 8ff

6b. Panel D, immunoblot for G5B WT and Mut shows a different size compared to G5B bands shown in Figure 1.

Indeed, immunoblot analyses of endogenous GPRC5B showed in RPM a dominant band between 42-45 kDa and a weaker band at 36-37 kDa, and previous studies with the same

GPRC5B antibody suggested that the upper band represents glycosylated GPRC5B.²¹ In HEK cells, we detected overexpressed, FLAG-tagged GPRC5B close to 40 kDa, although the gels did not run long enough to allow detailed evaluation of the molecular weight. The exact reasons for these differences between species and cell types are not clear, but it is possible that differences in cellular glycosylation play a role. Other obvious reasons such as faulty cDNA sequences in our expression plasmids or unspecific binding of the primary GPRC5B antibodies have been excluded (please see also our responses to questions 1a and 3a of this reviewer).

6c. Panel F, Immunoblot of EP2-G5B co-IP's need to be quantitated from at least three independent experiments.

As suggested by the reviewer, we quantified the strength of G5B-FLAG co-immunoprecipitation (co-IP) relative to the strength of HA-EP2 immunoprecipitation (IP) for wildtype GPRC5B and the different GPRC5B mutants shown in Fig. 6F (Fig. 30A). In addition, we also provide now a quantification for the co-IP experiments performed with GPRC5B-wt vs GPRC5B-mut7 in manuscript Fig. 6D (human sequence) and Suppl. Fig. 10B (mouse sequence) (Figs. 30B,C):

Figure 30, A-C, Densitometric analysis of signal strength for co-immunoprecipitated GPRC5B-FLAG (FLAG(coIP)) relative to immunoprecipitated HA-EP2 (HA(IP)) in HEK cells expressing wildtype GPRC5B (G5B-wt) or the indicated GPRC5B mutants: A, quantification for manuscript Fig. 6F; B, quantification for manuscript Fig. 6D; C, quantification for Suppl. Fig. 10B (n=3-6 each).

Data are means \pm SEM; comparisons between genotypes were done using one-way ANOVA with Dunnett's multiple comparison test (A) or unpaired t test (B,C). *, $P < 0.05$; **, $P < 0.01$.

These new data are shown in Supplemental Figures 10A, C, E, and referred to on page 10, lines 28, 31, and 35.

6d. Panel F and G, show loss of EP2-G5B mut 7 interaction and the function, however G5B mut set3 retains interaction with EP2 but appears to lose function, which is counter intuitive, and need to address.

This is a valid point, the data indeed suggest that GPRC5B-mut set3 has lost the ability to facilitate EP2-dependent cAMP production even though co-IP and EP2 membrane localization are normal. This suggests that the various residues identified in our *in silico* analysis have different relevance:

- Set1 residues are essential for the physical interaction with EP2, their mutation results in loss of interaction, reduced EP2 membrane localization and reduced EP2 signalling.
- Mutation of set2 residues does neither affect co-IP nor EP2 localisation and signalling.
- Mutation of set3 residues is not sufficient to completely break the interaction between GPRC5B and EP2 and does not affect intracellular EP2 trafficking, but still impairs the ability of GPRC5B to facilitate EP2-dependent cAMP production. This indicates that GRC5B can affect EP2 function beyond regulation of intracellular trafficking, but the exact nature of this facilitation is currently unclear. Studies in other GPCR dimers suggested that the interaction between the protomers may modulate ligand binding or effector coupling,^{1, 2} and it seems possible that set3 residues contribute to such effects.

Taken together, we believe that our results for GPRC5B-mut set1 clearly show that complete loss of GPRC5B/EP interaction is associated with altered trafficking and reduced EP2 signalling, but we agree that GPRC5B might modulate EP2 responses by additional, yet unknown mechanisms. We have added a corresponding statement to the discussion (page 13, line 23): “Interestingly, mutation of amino acids Q116, D118, L128 (set 3) reduced GPRC5B-mediated facilitation of EP2 signaling without breaking the physical interaction and without altering EP2 membrane localization, suggesting that GPRC5B might influence EP2 signaling through additional, yet unknown mechanisms.”

6e. Panel H, are the G5B wildtype and mut receptors expressed at the cell surface.

This is an important issue, and we have added the images showing localization of Myc-tagged GPRC5B (wildtype and mutant) to Figure 6H of the revised manuscript as shown in Fig. 31:

Figure 31: Revised version of manuscript Fig. 6H, now including images showing the GPRC5B(wt/mut)-Myc signal.

7. Figure 7 shows HIV-TAT -peptide expression to block interaction with a modest effect on cAMP formation.

7a. Panel B, impact of target peptide on EP2-G5B interaction immune-complex need to be quantified from at least three independent experiments

As suggested by the reviewer, we have quantified the effect of target and control peptides on co-immunoprecipitation of GPRC5B-FLAG/Myc with HA-EP2 in HEK cells in three independent experiments (Fig. 32).

Figure 32: Quantification of the effect of target and control peptides on co-immunoprecipitation of GPRC5B-FLAG with HA-EP2 in HEK cells (n=3): Densitometric analysis of signal strength for co-precipitated human GPRC5B-FLAG (FLAG(coIP)) relative to immunoprecipitated human HA-EP2 (HA(IP))

Data are means \pm SEM; comparisons between genotypes were done using one-way ANOVA with Dunnett's multiple comparison test. *, $P < 0.05$.

The quantification of the effect of control and target peptide on co-IP are shown in the new Suppl. Fig. 11A and mentioned on page 11, line 20 of the revised manuscript.

7b. In addition, the authors should determine if the target peptide enhanced EP2 retention in the golgi

Following the reviewer's suggestion we investigated the effect of control and target peptide on EP2 localization in membrane versus Golgi apparatus by immunostaining in RPM. We found that while control peptide did not cause significant changes, target peptide reduced colocalization of EP2 with membrane marker CD11b and increased colocalization with Golgi marker GM130 (Fig. 33). We have added these new data to Suppl. Fig. 11) of the revised manuscript and describe them on page 11, line 23ff.

Figure 33: A, Effect of target and control peptides on EP2 localization in RPM: wildtype RPM were treated with solvent (basal), control peptide, or target peptide (1 μ M each peptide) for 1 hour, then cells were stained with anti-CD11b antibodies (for plasma membrane), permeabilized, and stained for EP2(ec) and Golgi marker GM130. **B**, Statistical evaluation of the colocalization of EP2(ec) with CD11b and GM130 (n=15-18 cells per condition).

Data are means \pm SEM, comparisons between groups were done by two-way ANOVA with Sidak's post hoc test (B). *, $P < 0.05$; ***, $P < 0.001$; ****, $P < 0.0001$.

8. The authors should attempt to determine if dimer G5B-EP2 complex functions differently compared to the EP2 protomer for example by examining G protein coupling, desensitization or internalization.

Following the reviewer's suggestion we analysed EP2 internalization in the presence and absence of GPRC5B. In contrast to other prostanoid receptors, EP2 was previously shown not to undergo agonist-induced internalization,^{25, 26} and when we determined surface EP2 availability at different time points after butaprost exposure in BMDM, we indeed did not observe significant changes at 10, 30 and 60 minutes (Fig. 34A). The difference in membrane

availability between the genotypes, however, was stable over time, though it did not reach significance at 10 minutes (Fig. 34A).

To investigate whether GPRC5B influences EP2 function beyond regulation of membrane availability, we used an EP2 conformation sensor analogous to a previously described EP4 conformation sensor.²⁷ In this EP2 mutant, fluorophores are attached to intracellular loop 3 and C-terminus in close proximity, resulting in Förster resonance energy transfer (FRET). Agonist-induced activation of this EP2 mutant causes a decrease in FRET efficiency reflected as a decrease in the emission ratio (acceptor emission divided by donor emission). The EP2 activation sensor has retained the ability to co-precipitate with GPRC5B (Fig. 34B) and shows facilitation of cAMP production in the presence of GPRC5B (Fig. 34C). Both butaprost and PGE₂ induced concentration-dependent FRET changes in sensor-expressing HEK cells, but these responses were not altered by overexpression of GPRC5B (Fig. 34D). This suggests that GPRC5B does not interfere with the affinity of the receptor or the amplitude of the conformational change.

Figure 34: A, ELISA-based detection of EP2(ec) in the plasma membrane of M0 BMDM after different times of stimulation with 1 μM butaprost (n=8). **B-D, Characterization and application of the EP2 conformation sensor (EP2-GFP): B**, Western blot detection of FLAG, GFP, and HA signals in lysates of HEK cells expressing FLAG-tagged GPRC5B in combination with the GFP-tagged EP2 conformation sensor (EP2-GFP) or the wild-type HA-tagged EP2 before (“input”) and after immunoprecipitation of GPRC5B-FLAG (“Pulldown FLAG”). **C**, Butaprost-induced cAMP production in HEK cells transfected with a cAMP GloSensor plasmid, EP2-GFP, and empty vector (EV) or GPRC5B (G5B),

respectively (n=8). **D**, HEK cells transfected with EP2 receptor sensor alone (red) or together with GPRC5B (blue) were subjected to single cell FRET recordings. In the course of the measurement, different concentrations PGE₂ (solid line) or butaprost (dashed line) were applied. pEC₅₀ values for PGE₂ were calculated as 6.7 (95% CI: 6.6 to 6.8) for EP2+GPRC5B vs 6.8 (95% CI: 6.7 to 6.9) for EP2 only. pEC₅₀ values for butaprost were calculated as 6.1 (95% CI: 6.0 to 6.3) for EP2+GPRC5B vs 6.0 (95% CI: 5.8 to 6.1) for EP2 only.

Data are means ± SEM; comparisons were done using 2-way ANOVA with Sidak's post hoc test (A) or one-way ANOVA with Tukey's multiple comparison test (C); **, $P < 0.01$; ***, $P < 0.001$; ****, $P < 0.0001$.

We have added the internalization data to supplemental Fig. 9B and describe them on page 10, line 6ff. We also considered adding the sensor data to the supplemental figures, but since the description of the assay and its various control experiments is rather lengthy, we feel that it would be distracting for the reader. We therefore have currently only added the following statement to the discussion (page 13, line 26ff)

“We used a FRET-based EP2 conformation sensor constructed in analogy to a recently described EP4 sensor²⁷ to test whether GPRC5B interferes with EP2 affinity or the amplitude of the agonist-induced conformational change, but did not find altered FRET responses in the presence of overexpressed GPRC5B (data not shown)”.

Reviewer #3 (Remarks to the Author):

In this manuscript, Kwon et al. investigate the contribution of the orphan receptor GPCRC5B towards macrophage function, and its mechanism of action. Utilizing a mouse model of myeloid-lineage-specific KO of GCRC5b, they demonstrate that both peritoneal macrophages and BMDM display augmented chemotaxis, phagocytosis, and cytokine production. In a model of peritonitis, myeloid GPCRC5b deficiency appeared beneficial to improve clearance of pathogens. In models of chronic inflammation (myocardial infarction, DSS colitis), myeloid GPCRC5b deficiency was detrimental which they attribute to exacerbated macrophage-mediated inflammation and tissue damage. Detailed experiments in HEK293 cells (with some replicated in primary human macrophages), delineate a negative regulatory impact of GPCRC5b on EP2 signaling through physical interaction and control of surface expression.

Overall, this is a very well conducted study, with detailed and comprehensive experimental design, and thorough investigation of the function and mechanism of the orphan receptor GPCRC5B on macrophages. The manuscript is well written, and the figures are clear and data well presented. I have only a few questions to ensure that their overall conclusions are soundly supported by the data presented:

We thank the reviewer for the positive feedback and will address all issues in the following paragraphs.

Major:

1. In macrophages of their M-G5b-KO mice, is EPA surface expression increased, and EPA-mediated signalling augmented as their overall hypothesis would suggest? And does augmented EPA-mediated signaling contribute to the altered in vivo macrophage functions observed in these mice, as their hypothesis would suggest?

The role of eicosapentaenoic acid (EPA) in the regulation of inflammatory processes is indeed a fascinating topic, and EPA is known - among many other effects - to modulate macrophage activation. For example, EPA was shown to reduce expression of pro-inflammatory cytokines in LPS-stimulated THP1 macrophages,^{28, 29} and also in RAW 264.7 or J774 macrophages, EPA modulated inflammatory activity.^{30, 31} However, the observed effects showed a certain variability depending on the activation state of the respective cell line and the inflammatory parameter evaluated.^{30, 31} We agree that it is interesting to investigate whether EPA levels or EPA-mediated effects are altered in GPRC5B-deficient macrophages, though according to our literature research so far no direct links between EP2-mediated anti-inflammatory signaling and EPA expression/signaling have been suggested.

To address the question of EPA expression, we first determined EPA levels in BMDM from control and M-G5b-KO mice by mass spectrometry, but did not find significant changes (Fig. 35A).

To investigate whether EPA-mediated effects are altered in GPRC5B-deficient macrophages, we preincubated BMDM from control and KO mice with EPA for 48 h as described previously^{28, 29} and studied the consequences for LPS-induced inflammatory gene expression. As observed by others, pre-treatment with EPA 0.1 mM resulted in a reduction of inflammatory gene expression in BMDM from control mice (Fig. 35B,C), though not all genes were equally affected (Fig. 35D). Interestingly, a similar degree of reduction was observed in

BMDM of M-G5b-KOs, indicating that loss of GPRC5B does not have strong effects on EPA-induced responses.

Figure 35: **A**, Levels of eicosapentaenoic acid (EPA) were determined by LC-MS/MS in 1×10^7 M0 BMDM from control mice and M-G5b-KOs ($n=6$). **B-D**, Inflammatory gene expression was determined in M0 BMDM precubated with EPA 0.1 mM or vehicle for 48 h, then stimulated with LPS ($1 \mu\text{g/ml}$) for 6 h ($n=3$, data normalized to *Gapdh* and control set to 1).

Data are means \pm SEM; comparisons between genotypes were performed using unpaired t test (A) or two-way ANOVA with Sidak' correction for multiple testing (B-D). *, $P < 0.05$.

2. The structural work, determination of interacting residues, and development of a peptide blocking strategy is very nice. However, I was surprised that there were no experiments to confirm the functional implications of uncoupling GPCRC5b and EP2 interactions. That is, is macrophage function similarly affected by blocking physical interaction of GPCRC5b with EP2 as it is when GPCRC5a is deleted/knocked down? This is particularly important given the authors proposal that peptide-based blockade of GPCRC5B-AP2 interactions could be used therapeutically.

This is indeed a very important point, and we investigated whether inhibition of the GPCRC5B-EP2 interaction by decoy peptides was able to partially reproduce the enhanced phagocytosis seen in GPCRC5B-KO BMDM. We found that target peptide, but not control peptide indeed enhanced E.coli uptake in control BMDM (Fig. 36A,C). In BMDM from M-G5b-KOs, in contrast, neither control nor target peptide had significant effects (Fig. 36B,C).

Figure 36: A-C, Peptide effect (1 μ M each) on phagocytosis of pHrodo E. coli bioparticles in M0 BMDM: Exemplary traces from control (A) and KO (B) mice; C: statistical evaluation of AUC (n=9). Preincubation with peptides was 1 h. Please note that the basal difference between control and KO is less pronounced in the presence of vehicle DMSO

Data are means \pm SEM; differences between peptide-treated and untreated groups were analyzed using Kruskal-Wallis test with Dunn's multiple comparisons test (C). n, number of independent experiments); ***, $P < 0.001$.

We show these data in Figures 7E-G of the revised manuscript and describe them on page 11, line 25.

3. Minor:

3a. Fig 1A-B – *GPRC5b* is expressed in both RPM and BMDM (albeit at much lower levels), but appears negligibly expressed in T cells and B cells. Is *GPRC5b* expressed at similarly high levels in other tissue resident macrophages from other organs in mice, or is this unique to peritoneal macrophages?

This is an interesting question, and we reanalyzed *Gprc5b* expression in a previously published single-cell RNA sequencing data set.³² These data show that GPRC5B is weakly expressed in macrophages from mouse heart, lung, and brain (Fig. 37). However, the percentage of GPRC5B-expressing cells is very low (also for other GPCRs expected to be expressed in macrophages), which is most likely due to the well-known problem of poor sensitivity for low abundance transcripts in single-cell transcriptomics.

Figure 37: The expression of *Gprc5b* and various prostanoid receptors known to be expressed in macrophages was analyzed in single-cell RNA sequencing data set GSE188647.³²

We therefore also checked *Gprc5b* expression in resident macrophages by qRT-PCR. To do so, we isolated macrophages by fluorescent-activated cell sorting using the following antibody combinations and protocols:

- Peritoneum: CD45⁺, F4/80⁺, Tim4⁺ (Roberts et al. 2017³³)
- Liver: CD45⁺, F4/80⁺, Tim4⁺ (Kulle et al. 2022³⁴)
- Heart: CD45⁺, F4/80⁺, Tim4⁺ (Sansone et al. 2020³⁵)
- Lung: CD45⁺, CD11b^{low}, CD170⁺ (Alexander et al. 2013³⁶)

After isolation, *Gprc5b* expression was analysed by qRT-PCR. We found robust *Gprc5b* expression in peritoneal and alveolar macrophages, while hepatic Kupffer cells and cardiac macrophages showed lower expression (Fig. 38). We conclude that other resident macrophage populations do express *Gprc5b*, though transcript levels might vary between tissues.

Figure 38: *Gprc5b* expression was analyzed in resident macrophages from different organs by qRT-PCR (C, n=4, data normalized to *Gapdh* and peritoneal cells set to 1). Data are means ± SEM.

We mention these findings in the discussion on page 14, line 34ff: “GPC5B is also expressed in resident macrophages of liver, heart and lung (data not shown), but its role in these cells remains to be determined.”

*3b. In addition to BMDM, is GPRC5b expressed in other myeloid cells? Data on neutrophils is presented in Supp Fig 3, but I wonder about **circulating monocytes? Platelets?** I appreciate that this manuscript focuses on macrophages, but given the importance of prostaglandins towards the functions of these other cells types it would be of interest to know whether this receptor is expressed in other hematopoietic cell populations.*

This is another interesting question and we checked *Gprc5b* expression in platelets and monocytes isolated from murine blood. We found that freshly isolated Ly6c-positive murine monocytes expressed *Gprc5b* at slightly lower levels than BMDM and RPM, whereas expression in platelets was with ct values in the range of 41 to 42 (corresponding to dct values between 25 and 26) barely detectable (Fig. 39A). We also reanalyzed *Gprc5b* expression in previously published expression data for murine megakaryocytes and found that even though also here signals for *Gprc5b* were detected, they were lower than mRNAs for GPCRs with known function in platelets such as the P2Y12 receptor, the thromboxane A2 receptors *Tbxa2r*, or thrombin receptors *F2r* and *F2r12* (Fig. 39B, data reanalyzed from GSE222512³⁷). We conclude that while monocytes probably express functionally relevant amounts of *Gprc5b*, expression in platelets is marginal.

Figure 39: A, *Gprc5b* expression was determined by qRT-PCR in platelets and Ly6c-positive monocytes isolated from mouse blood (n=3-4 per condition; data displayed as Δ ct values (ct value *Gprc5b* – ct value *Gapdh*)). Data for RPM and BMDM are shown for the sake of comparison. **B**, Expression of select GPCRs as well as reference genes *Pf4* (platelet factor 4) or *Gapdh* was re-analyzed in data set GSE222512³⁷). Data are means \pm SEM.

We included a statement regarding *Gprc5b* expression in platelets and monocytes on page 7, line 8-10: “In other populations of the myeloid lineage, such as eosinophils or platelets, *Gprc5b* expression is negligible, while blood monocyte express *Gprc5b* at levels slightly lower than BMDM (data not shown).”

Reviewer #4 (Remarks to the Author):

In this study, the authors investigated the role of the orphan G protein-coupled receptor GPRC5B and the prostaglandin receptor EB2 on the regulation of macrophage function. For this purpose, the authors employed various in vivo models and an impressive array of in vitro and in silico exploratory approaches. The topic of finetuning the pro- and anti-inflammatory regulatory pathways during inflammatory processes is of great scientific interest and certainly holds great translational potential for the development of future therapeutic strategies. Generally, the manuscript is well written, and the results are presented in a structured manner. However, the following points need to be addressed:

We thank the reviewer for the supportive feedback and will address all issues in the following paragraphs.

1. The authors show the expression of GPRC5B in RPMs, BMDMs and lymphocytes, but not in other leukocytes. Is there anything known about granulocytes, e.g. neutrophils? LysM-Cre is also very prominently expressed in neutrophils, so in how far does the conditional knockout refer to the macrophage function and renders possible neutrophil-mediated effects irrelevant?.

It is indeed possible that the phenotypes observed in M-G5b-KOs are not solely due to loss of GPRC5B in macrophages, but also due to other cell types known to express *LysM-Cre*, most prominent among them neutrophils. GPRC5B is weakly expressed in neutrophils, and qRT-PCR showed that this expression is further reduced in M-G5b-KOs (Fig. 40A). We therefore studied the consequences of GPRC5B-deficiency in neutrophils and found that while phagocytosis was not altered (Fig. 40B), *in vitro* migration was increased (Fig. 40C). *In vivo*, however, we did not find evidence for enhanced neutrophil recruitment in bacterial or thioglycolate-induced peritonitis (Fig. 40D,E), suggesting that the promigratory phenotype observed *in vitro* is not relevant under *in vivo* conditions. In line with this, we did not observe significant difference in neutrophilic infiltration and consecutive skin inflammation in the model of bullous pemphigoid-like epidermolysis bullosa acquisita (EBA), a blistering skin disease that strongly depends on neutrophilic infiltration³⁸ (Fig. 40F,G). Taken together, though GPRC5B-deficient neutrophils showed enhanced migration *in vitro*, we did not find clear evidence for altered neutrophil recruitment *in vivo*. We show these data in Supplemental Figure 3 of the revised manuscript and describe them on page 6, line 32ff of the revised manuscript.

Figure 40: Phenotypes in GPRC5B-deficient neutrophils **A**, Knockout efficiency was determined by qRT-PCR in Ly6G-positive cells harvested from bone marrow (n=8, data normalized to *Gapdh* and controls set to 1). **B**, Phagocytic activity of Ly6G-positive neutrophils isolated from murine bone marrow was determined by uptake of pHrodo *E. coli* bioparticles (n=8). **C**, Basal and chemoattractant-induced transwell migration was studied in Ly6G-positive neutrophils isolated from murine bone marrow (C5a: 20 ng/ml; fMLP: 10 nM, LTB4: 1 uM, CXCL1: 1 uM, CXCL3: 1 uM; n=6). **D,E**, The number of Ly6G-positive neutrophils was determined in control and M-G5b-KOs in the basal state and 24 h after induction of bacterial peritonitis (D) or 12 h after i.p. administration of 1 ml Brewer's thioglycolate (TG) solution (E)(n=4). **F,G**, EBA in control and M-G5b-KO mice: Affected body surface area (ABSA, F) and immune cell infiltration into skin biopsies obtained on d16 (G) (n=5).

Data are means \pm SEM; comparisons between genotypes were performed using Mann-Whitney test (A), two-way ANOVA (B,C,D,E,G) or two-way RM-ANOVA (F) with Sidak' correction for multiple testing. ****, $P < 0.0001$; ***, $P < 0.001$; **, $P < 0.01$; n, number of individual mice.

To investigate *Gprc5b* expression in eosinophils, we analyzed our own single-cell expression data and publicly available RNA sequencing data, but did not find evidence for *Gprc5b* expression in individual *Siglec1*-positive eosinophils from murine blood or bone marrow (Fig. 41A) or in a bulk RNA sequencing of *Siglec1*-positive eosinophils from murine thymus, spleen, or lung (GSE266084 from Ota *et al.*, 2024³⁹)(Fig. 41B). These data suggest that *Gprc5b* is not expressed in murine eosinophils, and we included a corresponding statement on page 7, line 8 of the revised manuscript.

Figure 41: A, *Gprc5b* expression in *Siglec1*-positive eosinophilic granulocytes from murine blood or bone marrow was analyzed in an unpublished single-cell RNA sequencing data set available in the lab. **B**, Expression of *Siglec1* (as marker for eosinophils) and *Gprc5b* was analyzed in publicly available RNA sequencing data from murine thymus, spleen, or lung (GSE266084 from³⁹). Data are means \pm SEM.

2. Figure 1G-H: While the text refers to LPS treatment of RPMs, the figure refers to C5a, CCL5 and fMLP stimulation. Please clarify.

Figures 1G and H investigate basal and chemokine-induced migration in RPM. These cells show in their naïve state rather modest responses to chemoattractants, which can be enhanced by pretreatment with LPS (Fig. 42).

Figure 42: A, Comparison of basal and C5a (20 ng/ml)-induced transwell migration between naïve RPM and RPM that had been pretreated with LPS 1 µg/ml for 3h to facilitate migration (n=1).

We therefore used in all our migration experiments RPM that had been activated with LPS prior to the analysis. To clarify this point we have slightly rephrased the legend for Fig. 1G/H:

“**G,** Basal and C5a (20 ng/ml)-induced transwell migration in RPM (all cells pretreated with LPS 1 µg/ml for 3h to facilitate migration) (n=3). **H,** The distance travelled by individual RPM in response to different chemotactic factors (C5a, 20 nM; CCL5, 10 ng/ml; fMLP, 10 nM) was determined by live cell imaging (n=1000 cells from 2 mice per group; cells pretreated with LPS 1 µg/ml for 3h to facilitate migration).”

3. Figure 1M-N: Was the activation and recruitment of other immune cells, e.g. neutrophils, significantly altered in M-G5b-KO mice?

Following the reviewer’s suggestion we analyzed the number of B cells, T cells, and neutrophils in peritoneal lavage fluid before and 24 h after induction of bacterial peritonitis. B cells, which are abundant in the peritoneal cavity under basal conditions, showed the same disappearance reaction as resident macrophages, whereas T cells and neutrophils were recruited from the blood (Fig. 43). However, neither B cell disappearance nor T cell or neutrophil recruitment differed between the genotypes, suggesting that *LysM*-Cre-dependent inactivation of GPRC5B does not significantly affect recruitment of these cells.

Figure 43: A-C, Numbers of CD19⁺ B cells (A), TCRβ⁺ T cells (B), or Ly6G⁺ neutrophilic granulocytes (C) before and 24 h after i.p. injection of fecal bacteria (n=4-5).

Data are means ± SEM; comparisons between genotypes were performed using two-way ANOVA with Sidak’ correction for multiple testing.

We show these additional data in Supplemental Figures 1K,L of the revised manuscript and describe them on page 5, line 32.

4. Figure 1N-P: From the figure and the figure legend it remains unclear which bar refers to which genotype. Please clarify. Furthermore, in how far do the observed effects after 24-54 days relate to the findings during the first 24 hours?

The color coding in Figure 1 is in all panels white for control and gray for M-G5b-KOs, and we have now added this information to the figure legend. In addition, we added the color legend once more at the bottom of the revised Figure 1.

Regarding the time course shown in Figs. 1O and 1P we have to apologize: The x axis label should read “Hours”, not “Days”. We have corrected the mistake.

5. Figure 2A: should *GPRC5b* expression actually not be fully abrogated in conditional KO mice? Why is there still partial expression both in RPMs and BMDMs?

This is a valid question, and two factors might contribute to incomplete inactivation of *Gprc5b* in RPM and BMDM:

- 1) Contamination with non-myeloid cells that do not express *LysM-Cre* and therefore are not recombined. For isolation of RPM, we used the “mouse macrophage isolation Kit (peritoneum)” (Miltenyi, 130-110-434), which depletes non-target cells (T cells, B cells, NK cells, dendritic cells, erythroid cells, granulocytes) using a magnetic cell sorting (MACS) approach. Flow cytometric analysis showed that MACS purification very efficiently reduced B cells and other F4/80-negative cells (Fig. 44A), suggesting that contamination with non-macrophage cell types probably does not play a major role in this preparation. In BMDM, macrophage enrichment was obtained by culture in the presence of M-CSF, which resulted on day 7 in a highly F4/80-positive population that showed only minor contamination with CD19-positive B cells, TCR β -positive T cells or Lyg6G-positive neutrophils (Fig. 44B-D).

Figure 44: A, Flow cytometric analysis of peritoneal cells before and after MACS. **B-D,** Flow cytometric analysis of BMDM harvest after 7 days of M-CSF differentiation.

- 2) Presence of macrophages that do not express *LysM-Cre*. According to the original publication from Clausen et al.,⁴⁰ the recombination efficiency of *LysM-Cre* in RPM was calculated to be 83.2% for FACS-purified F4/80-positive macrophages. In BMDM, recombination efficiency ranged (depending on the differentiation state of the macrophages) between 38% (F4/80⁺/MHCII⁺ cells) and 98% (F4/80⁺/MHCII⁻). The proportion of F4/80⁺/MHCII⁺ cells was in our preparation around 7% (Fig. 45), and these cells will most likely contribute to remaining *Gprc5b* transcripts in BMDM.

Figure 45: A, Flow cytometric analysis of the proportion of MHCII⁺;F4/80⁺ cells on day 7 of BMDM differentiation.

6. *Figure 2B-C: this finding appears contradictory compared to the data shown in Figure 1F and Supplemental Figure 1G. Please clarify.*

This is indeed an interesting point. In LPS-stimulated RPM, knockout of GPRC5B resulted mainly in upregulation of *Nos2*, whereas inflammatory genes such as *Tnf*, *Il6*, or *Il1b* were not increased (Fig. 46A-D). In GPRC5B-deficient BMDM, in contrast, a broad facilitation of LPS-induced inflammatory gene expression was observed (Fig. 46E-H).

Figure 46: Expression of inflammatory genes in resting and LPS (1 $\mu\text{g/ml}$, 6h)-stimulated RPM (A-D) and BMDM (E-H) was determined by qRT-PCR.

Data are means \pm SEM; comparisons between genotypes were performed using two-way ANOVA with Sidak's multiple comparison test; *, $P < 0.05$; ***, $P < 0.001$.

The reasons for this discrepancy between BMDM and RPM are not fully understood, but it seems likely that differences in the cellular response to LPS play a role. RPM and BMDM are developmentally and functionally very dissimilar, and literature suggests that they differ with respect to basal activation and responsiveness to inflammatory stimulation.⁴⁻⁷ This can also be seen in our qRT-PCR data above, in which BMDM show much stronger LPS responses than RPM (Fig. 46). To better visualize this point, we have plotted ΔCt values as a measure of gene expression in basal and LPS stimulated BMDM and RPM (Fig. 47). These data do not only show that LPS-induced upregulation of *Il6*, *Il1b*, and *Tnf* is weaker in RPM (green) than in BMDM (red), but also that RPM are already in the basal state on a significantly higher gene expression level, especially for *Il6* and *Il1b*.

Figure 47: Expression of inflammation markers in basal and LPS-stimulated BMDM (red) and RPM (green) was analyzed by qRT-PCR (data displayed as Δ ct values (ct value *target gene* – ct value *Gapdh*)). Data are means \pm SEM; comparisons between genotypes were performed using 2-way ANOVA with Tukey's posthoc test; ***, $P < 0.001$; ****, $P < 0.0001$.

As to the mechanism how GPRC5B regulates inflammatory gene expression in these two cell populations, we hypothesized an interference with NF- κ B activation: inflammation markers such as *Nos2*, *Tnf*, *Il6*, and *Il1b* are mainly regulated by NF- κ B,⁸ and elevated cAMP levels have been shown to interfere with p65-NF- κ B activation.^{9, 10} To test whether NF- κ B activation is indeed enhanced in the absence of GPRC5B, we determined I κ B α degradation and found it to be significantly lower in LPS (1h)-stimulated KO cells compared to control (Fig. 48A,B), indicative of increased NF- κ B activation. Interestingly, RPM showed hardly any I κ B α degradation after LPS exposure (Fig. 48C,D), suggesting that these cells – maybe due to their higher basal activation – are less responsive to LPS. This, in turn, might explain while loss of GPRC5B-EP2-mediated suppression of NF- κ B activation is in these cells functionally less relevant and therefore no difference in *Il6*, *Tnf*, *Il1b* expression is observed.

Figure 48: A,B, LPS-induced degradation of NF-κB inhibitor IκBα was determined by western blotting in BMDM: representative immunoblots (A) and statistical evaluation (B), GAPDH as loading control. **C,D**, LPS-induced degradation of NF-κB inhibitor IκBα was determined by western blotting in RPM, GAPDH as loading control. Data are means ± SEM; **, $P < 0.01$.

Data are means ± SEM; comparisons between groups are performed using unpaired t test (B); **, $P < 0.01$.

However, even in the absence of strong NF-κB activation some LPS-induced activation of inflammatory gene expression is observed in RPM, most prominently for *Nos2* (Fig. 46A above). The *NOS2* promoter is not only regulated by its NF-κB binding site, but has several other regulatory components that are essential for complete expression of *NOS2*, for example binding sites for AP-1, signal transducer and activator of transcription-1 (STAT1), and IFN regulatory factor-1 (IRF-1).¹¹⁻¹³ In addition, epigenetic regulation of *NOS2* promoter accessibility by the NLR4/caspase-1 axis has been suggested in macrophages.¹⁴ All these alternative regulators of *NOS2* expression are present in RPM, and though their role has not been specifically studied in RPM, it seems likely that they contribute to LPS-induced upregulation of *NOS2*. Interestingly, cAMP has been shown to interfere with the activation of STAT1 or AP-1 signalling,^{15, 16} and it is tempting to speculate that altered activation of these pathways might underlie increased *NOS2* expression in GPRC5B-deficient RPM. However, since we did not address these hypotheses experimentally, we feel that these consideration would be rather distracting for the reader and prefer not to address them in the revised version of the manuscript.

7. Figure 2I-K: Representative images of cardiac histology and exemplary cardiac ultrasound measurements should be included in the figure file.

Following the reviewer's suggestion we included representative histological sections and snapshots from ultrasound measurements in Suppl. Figs. 2H and 2K (data also shown in Fig. 49 below).

Figure 49: A, Representative picosirius red-stained sections underlying the scar size analysis shown in main Fig. 2K. **B,** Representative images of the cardiac long axis in B-mode, hatched lines indicate the volume of the left ventricle.

8. Given the effects of GPRC5B knockout in neutrophils shown in Supplemental Figure 3, the negative *in vivo* results come rather as a surprise. Furthermore, the EBA model is not among the most commonly used models to study the pathophysiological relevance of gene knockouts in neutrophils and the authors should consider additional models. This is very important, as *LysM-Cre* driven conditional knockout models also affect gene expression in neutrophils.

Indeed, based on the enhanced neutrophil migration observed *in vitro* (Suppl. Fig. 3B) one might have expected enhanced responses also *in vivo* – on the other hand, phagocytosis (Suppl. Fig. 3C) was not altered, suggesting a weaker phenotype in GPRC5B-deficient neutrophils compared to macrophages. We also agree that EBA – though being clearly a highly neutrophil-dependent disease³⁸ – is not the most obvious model. We therefore added an *in vivo* model that investigates neutrophil migration / inflammatory recruitment in a more general way, thioglycolate (TG)-induced peritonitis.⁴¹ In this model, same as in the bacterial peritonitis model, we did not observe differences in neutrophil recruitment between the genotypes (Fig. 50), indicating that enhanced neutrophil migration observed *in vitro* is not relevant under *in vivo* conditions

Figure 50: The number of Ly6G-positive neutrophils was determined in control and M-G5b-KOs in the basal state and 12 h after i.p. administration of 1 ml Brewer's thioglycolate solution (n=4).

Data are means \pm SEM; comparisons between genotypes were performed using two-way ANOVA with Sidak' correction for multiple testing.

We added these new data as Suppl. Fig. 3F to the revised manuscript and describe them on page 7, line 7.

9. Supplemental Figure 4: what is the relevance of this data for the study?

A previously published study suggested that GPRC5B might regulate the phosphorylation of sphingomyelin synthase 2, thereby influencing cellular ceramide and sphingoid base levels.⁴² Changes in ceramide or sphingoid base levels may affect macrophage function not only by altering membrane properties, but also because of their (direct or indirect) role in cellular signaling.^{43, 44} We therefore tested whether the phenotype of GPRC5B-deficient macrophages might be related to altered ceramide / sphingoid base levels, but found no clear differences between the genotypes. Even though these are negative data, we believe they might be interesting for researches in the lipid field and would therefore suggest to keep them in the supplement.

10. Supplemental Figure 6B: the knockdown efficiency reaches about 75% on average, with low efficiency samples in the range of roughly 50%. This knockdown efficiency appears rather low and raises the question if it is enough to detect possible effects.

True, though the current knockdown efficiency of about 75% was good enough to reduce EP2-dependent effects, we cannot fully exclude that a better efficiency would also have effects on other GPRC5B-interacting receptors such as EP1. We therefore tested various alternative transfection protocols to improve knockdown efficiency without impairing viability and found that doubling of the siRNA concentration increased knockdown efficiency to 84% (Fig. 51A). We then re-evaluated the effect of these improved knockdown conditions on EP1-mediated calcium mobilization and found that it was still unchanged (Fig. 51B), indicating that loss of GPRC5B indeed does not have clear effects on EP1-mediated signaling in HEK cells.

Figure 51: A, Efficiency of *Gprc5b* knockdown after transfection with doubled siRNA concentration was determined by qRT-PCR (n=12). **B,** Calcium mobilization induced by EP1 agonist 17-pt-PGE₂ was determined in HEK cells transfected with calcium sensor aequorin, EP1 receptor, and control siRNA or siRNA directed against GPRC5B using the improved transfection protocol used in A (n=4).

Data are means ± SEM; comparisons between genotypes were performed using unpaired t test (A) or two-way ANOVA and Sidak's post-hoc test (B). ***, $P < 0.001$.

11. Figure 4H: a knockdown efficiency of 50% does not appear suitable for functional assays.

We agree that a knockdown efficiency of 50% is suboptimal, but already this degree of knockdown was difficult to achieve. Macrophages are notoriously difficult to transfect and almost all transfection approaches drastically reduce cell viability or interfere with cell function. We compared two different protocols, RNAiMax and Amaxa nucleofection, which both resulted on the mRNA level in relatively poor transfection efficiencies of approximately 50% (Fig. 52A,B). This kd efficiency is in line with data shown in the "Amaxa™ 4D-Nucleofector™ Protocol for THP-1 [ATCC®]", where a reduction of target gene expression by 44% was achieved. Since cell viability seemed slightly better for the nucleofection protocol, we proceeded with this method and tested GPRC5B expression on the protein level, which was despite moderate efficiency on the mRNA level relatively good (Fig. 52C). In line with this, we observed the predicted reduction of cAMP levels in GPRC5B-deficient THP1 cells (Fig. 52C). We have added the western blot data shown in Fig. 52C to Suppl. Fig. 7E and mention them on page 9, line 3 of the revised manuscript.

Figure 52: A-C, Efficiency of GPRC5B knockdown in THP1 cells on the RNA (A,B) and protein (C) level using RNAiMAX (A) or Amaxa Nucleofection (B,C). **D,** Functional consequence of GPRC5B knockdown in THP1 (from manuscript Fig. 4J).

Data are means ± SEM; comparisons between groups were performed using 2-way ANOVA with Tukey's posthoc test; *, $P < 0.05$; ***, $P < 0.001$; ****, $P < 0.0001$.

12. Figure 4J and general comment: gene expression should not only be assessed on the transcriptional level, but also by protein expression.

This is a valid point, and we have now prepared immunoblots showing the knockout efficiency in BMDM (Fig. 53A), examples for GPRC5B knockdown and overexpression in THP1 cells (Fig. 53B), and the effect of RA on GPRC5B expression in BMDM (Fig. 53C,D). Furthermore, we have included western blotting data for the regulation of EP2 expression, for example for RA-induced downregulation of EP2 in BMDM (Fig. 53E,F) or LPS-induced downregulation of EP2 in BMDM (Fig. 53G,H).

Figure 53: A-D, Detection of GPRC5B by immunoblotting: A, BMDM lysates obtained from control and M-G5b-KO (M-KO) BMDM; **B,** The effect of GPRC5B knockdown (G5B kd) or overexpression (G5B OE) was determined in THP1 by immunoblotting (Control: EV + control siRNA; G5B kd: EV + GPRC5B siRNA; G5B OE: GPRC5B expression plasmid + control siRNA). **C,D,** Lysates of resting and RA-treated BMDM. **E-H, Detection of EP2 by immunoblotting** in lysates of BMDM that were untreated or stimulated with RA or LPS for 24 h as indicated. EV, empty vector, siContr, scrambled control siRNA.

Data are means \pm SEM; comparisons between groups were performed using unpaired t test; *, $P < 0.05$; ***, $P < 0.001$.

These new data are shown in shown in Suppl. Fig. 7E (knockdown and overexpression in THP1), in Suppl. Fig. 8A,B (effect of RA on GPRC5B expression in BMDM), in Suppl. Fig. 8C,D (effect of RA on EP2 expression in BMDM), and in Suppl. Fig. 8F,G (effect of LPS on EP2 expression in BMDM).

References

1. Gurevich VV and Gurevich EV. How and why do GPCRs dimerize? *Trends Pharmacol Sci.* 2008;29:234-40.
2. Milligan G. G protein-coupled receptor hetero-dimerization: contribution to pharmacology and function. *Br J Pharmacol.* 2009;158:5-14.
3. Balmer JE and Blomhoff R. Gene expression regulation by retinoic acid. *J Lipid Res.* 2002;43:1773-808.
4. Zajd CM, Ziemba AM, Miralles GM, Nguyen T, Feustel PJ, Dunn SM, Gilbert RJ and Lennartz MR. Bone Marrow-Derived and Elicited Peritoneal Macrophages Are Not Created Equal: The Questions Asked Dictate the Cell Type Used. *Frontiers in immunology.* 2020;11:269.
5. Locati M, Curtale G and Mantovani A. Diversity, Mechanisms, and Significance of Macrophage Plasticity. *Annu Rev Pathol.* 2020;15:123-147.
6. Lazarov T, Juarez-Carreno S, Cox N and Geissmann F. Physiology and diseases of tissue-resident macrophages. *Nature.* 2023;618:698-707.
7. Davies LC and Taylor PR. Tissue-resident macrophages: then and now. *Immunology.* 2015;144:541-8.
8. Pahl HL. Activators and target genes of Rel/NF-kappaB transcription factors. *Oncogene.* 1999;18:6853-66.
9. Parry GC and Mackman N. Role of cyclic AMP response element-binding protein in cyclic AMP inhibition of NF-kappaB-mediated transcription. *J Immunol.* 1997;159:5450-6.
10. Takahashi N, Tetsuka T, Uranishi H and Okamoto T. Inhibition of the NF-kappaB transcriptional activity by protein kinase A. *Eur J Biochem.* 2002;269:4559-65.
11. Chu SC, Marks-Konczalik J, Wu HP, Banks TC and Moss J. Analysis of the cytokine-stimulated human inducible nitric oxide synthase (iNOS) gene: characterization of differences between human and mouse iNOS promoters. *Biochem Biophys Res Commun.* 1998;248:871-8.
12. Marks-Konczalik J, Chu SC and Moss J. Cytokine-mediated transcriptional induction of the human inducible nitric oxide synthase gene requires both activator protein 1 and nuclear factor kappaB-binding sites. *J Biol Chem.* 1998;273:22201-8.
13. Saura M, Zaragoza C, Bao C, McMillan A and Lowenstein CJ. Interaction of interferon regulatory factor-1 and nuclear factor kappaB during activation of inducible nitric oxide synthase transcription. *J Mol Biol.* 1999;289:459-71.
14. Buzzo CL, Medina T, Branco LM, Lage SL, Ferreira LC, Amarante-Mendes GP, Hottiger MO, De Carvalho DD and Bortoluci KR. Epigenetic regulation of nitric oxide synthase 2, inducible (Nos2) by NLRC4 inflammasomes involves PARP1 cleavage. *Sci Rep.* 2017;7:41686.
15. Kawada N, Uoya M, Seki S, Kuroki T and Kobayashi K. Regulation by cAMP of STAT1 activation in hepatic stellate cells. *Biochem Biophys Res Commun.* 1997;233:464-9.
16. Lamph WW, Dwarki VJ, Ofir R, Montminy M and Verma IM. Negative and positive regulation by transcription factor cAMP response element-binding protein is modulated by phosphorylation. *Proc Natl Acad Sci U S A.* 1990;87:4320-4.
17. Piplani N, Roy T, Saxena N and Sen S. Bulky glycolyx shields cancer cells from invasion-associated stresses. *Transl Oncol.* 2024;39:101822.
18. Montgomery MR and Hull EE. Alterations in the glycome after HDAC inhibition impact oncogenic potential in epigenetically plastic SW13 cells. *BMC Cancer.* 2019;19:79.
19. Kilgore JA, Dolman NJ and Davidson MW. A review of reagents for fluorescence microscopy of cellular compartments and structures, Part III: reagents for actin, tubulin, cellular membranes, and whole cell and cytoplasm. *Curr Protoc Cytom.* 2014;67:12 32 1-12 32 17.
20. Michel MC, Wieland T and Tsujimoto G. How reliable are G-protein-coupled receptor antibodies? *Naunyn Schmiedebergs Arch Pharmacol.* 2009;379:385-8.
21. Fu W, Franchini L and Orlandi C. Comprehensive Spatial Profile of the Orphan G Protein Coupled Receptor GPRC5B Expression in Mouse Brain. *Front Neurosci.* 2022;16:891544.

22. Varol C, Mildner A and Jung S. Macrophages: development and tissue specialization. *Annu Rev Immunol.* 2015;33:643-75.
23. Park MD, Silvin A, Ginhoux F and Merad M. Macrophages in health and disease. *Cell.* 2022;185:4259-4279.
24. Yang S, Zhao M and Jia S. Macrophage: Key player in the pathogenesis of autoimmune diseases. *Frontiers in immunology.* 2023;14:1080310.
25. Desai S, April H, Nwaneshiudu C and Ashby B. Comparison of agonist-induced internalization of the human EP2 and EP4 prostaglandin receptors: role of the carboxyl terminus in EP4 receptor sequestration. *Mol Pharmacol.* 2000;58:1279-86.
26. Nishigaki N, Negishi M and Ichikawa A. Two Gs-coupled prostaglandin E receptor subtypes, EP2 and EP4, differ in desensitization and sensitivity to the metabolic inactivation of the agonist. *Mol Pharmacol.* 1996;50:1031-7.
27. Kurz M, Ulrich M, Bittner A, Scharf MM, Shao J, Wallenstein I, Lemoine H, Wettschureck N, Kolb P and Bunemann M. EP4 Receptor Conformation Sensor Suited for Ligand Screening and Imaging of Extracellular Prostaglandins. *Mol Pharmacol.* 2023;104:80-91.
28. Weldon SM, Mullen AC, Loscher CE, Hurley LA and Roche HM. Docosahexaenoic acid induces an anti-inflammatory profile in lipopolysaccharide-stimulated human THP-1 macrophages more effectively than eicosapentaenoic acid. *The Journal of nutritional biochemistry.* 2007;18:250-8.
29. Mullen A, Loscher CE and Roche HM. Anti-inflammatory effects of EPA and DHA are dependent upon time and dose-response elements associated with LPS stimulation in THP-1-derived macrophages. *The Journal of nutritional biochemistry.* 2010;21:444-50.
30. Honda KL, Lamon-Fava S, Matthan NR, Wu D and Lichtenstein AH. EPA and DHA exposure alters the inflammatory response but not the surface expression of Toll-like receptor 4 in macrophages. *Lipids.* 2015;50:121-9.
31. Martins de Lima-Salgado T, Coccuzzo Sampaio S, Cury-Boaventura MF and Curi R. Modulatory effect of fatty acids on fungicidal activity, respiratory burst and TNF-alpha and IL-6 production in J774 murine macrophages. *Br J Nutr.* 2011;105:1173-9.
32. Dick SA, Wong A, Hamidzada H, Nejat S, Nechanitzky R, Vohra S, Mueller B, Zaman R, Kantores C, Aronoff L, Momen A, Nechanitzky D, Li WY, Ramachandran P, Crome SQ, Becher B, Cybulsky MI, Billia F, Keshavjee S, Mital S, Robbins CS, Mak TW and Epelman S. Three tissue resident macrophage subsets coexist across organs with conserved origins and life cycles. *Sci Immunol.* 2022;7:eabf7777.
33. Roberts AW, Lee BL, Deguine J, John S, Shlomchik MJ and Barton GM. Tissue-Resident Macrophages Are Locally Programmed for Silent Clearance of Apoptotic Cells. *Immunity.* 2017;47:913-927 e6.
34. Kulle A, Thanabalasuriar A, Cohen TS and Szydlowska M. Resident macrophages of the lung and liver: The guardians of our tissues. *Frontiers in immunology.* 2022;13:1029085.
35. Sansonetti M, Waleczek FJG, Jung M, Thum T and Perbellini F. Resident cardiac macrophages: crucial modulators of cardiac (patho)physiology. *Basic Res Cardiol.* 2020;115:77.
36. Misharin AV, Morales-Nebreda L, Mutlu GM, Budinger GR and Perlman H. Flow cytometric analysis of macrophages and dendritic cell subsets in the mouse lung. *Am J Respir Cell Mol Biol.* 2013;49:503-10.
37. Du CH, Wu YD, Yang K, Liao WN, Ran L, Liu CN, Zhang SZ, Yu K, Chen J, Quan Y, Chen M, Shen MQ, Tang H, Chen SL, Wang S, Zhao JH, Cheng TM and Wang JP. Apoptosis-resistant megakaryocytes produce large and hyperreactive platelets in response to radiation injury. *Mil Med Res.* 2023;10:66.
38. Kasperkiewicz M, Sadik CD, Bieber K, Ibrahim SM, Manz RA, Schmidt E, Zillikens D and Ludwig RJ. Epidermolysis Bullosa Acquisita: From Pathophysiology to Novel Therapeutic Options. *J Invest Dermatol.* 2016;136:24-33.
39. Ota A, Iguchi T, Nitta S, Muro R, Mino N, Tsukasaki M, Penninger JM, Nitta T and Takayanagi H. Synchronized development of thymic eosinophils and thymocytes. *Int Immunol.* 2024.
40. Clausen BE, Burkhardt C, Reith W, Renkawitz R and Forster I. Conditional gene targeting in macrophages and granulocytes using LysMcre mice. *Transgenic Res.* 1999;8:265-77.

41. Bosse R and Vestweber D. Only simultaneous blocking of the L- and P-selectin completely inhibits neutrophil migration into mouse peritoneum. *Eur J Immunol.* 1994;24:3019-24.
42. Hirabayashi Y and Kim YJ. Roles of GPRC5 family proteins: focusing on GPRC5B and lipid-mediated signalling. *J Biochem.* 2020;167:541-547.
43. Olona A, Hateley C, Muralidharan S, Wenk MR, Torta F and Behmoaras J. Sphingolipid metabolism during Toll-like receptor 4 (TLR4)-mediated macrophage activation. *Br J Pharmacol.* 2021;178:4575-4587.
44. Maceyka M and Spiegel S. Sphingolipid metabolites in inflammatory disease. *Nature.* 2014;510:58-67.

1. Related to previous review 1a: specific interaction or heterodimerization between GPRC5B and EP2 is one of important conclusions presented in the manuscript and this conclusion is solely based on the data generated from co-IP. As pointed out in previous review, these data are not strong enough to support the conclusion and would like to suggest the authors to prove the two GPCRs can physically interact or are in close proximity in other protein-protein interaction assays, such as live cell BRET or FRET, using other GPCRs as controls, and also determine the effects of decoy peptides.

To further support the notion that the GPRC5B/EP2 interaction is specific, we followed the reviewer's suggestion and performed FRET analyses. To do so, HEK293 (HEK) cells were transfected with EP2-mTurquoise2 (mTq2) and one of the following mCitrine (mCit)-labelled receptors: GPRC5B, the β 2 adrenergic receptor (β 2AR), the glucagon like peptide 1 receptor (GLP1R), or the purinergic receptor P2RY12. We found that initial FRET between EP2 and GPRC5B was significantly stronger than between any of the other pairs, indicating that EP2 indeed specifically interacts with GPRC5B (Fig. 1).

Figure 1: HEK cells were transfected with EP2-mTurquoise2 (EP2-mTq2) and different mCitrine (mCit)-labeled receptors and initial FRET was determined (n=9-12 individual cells in 2-3 independent experiments). Data are means \pm SEM; comparisons between groups were done using one-way ANOVA with Tukey's multiple comparison test. ****, $P < 0.0001$.

We also used NanoBRET to further characterize the interaction between EP2 and GPRC5B, in this case using two other class C GPCRs as controls: GPRC5C, the receptor with the highest homology to GPRC5B, and GPR156, a class C orphan with lower homology to GPRC5B. To do so, HEK cells were transfected with EP2-Venus as acceptor and one of the following NLuc-labelled receptors as donor: GPRC5B-NLuc, GPRC5C-NLuc, GPR156-NLuc or NLuc alone. These experiments showed that GPRC5C showed comparable energy transfer to EP2 as GPRC5B, whereas energy transfer from GPR156 was significantly lower (Fig. 2A). This was in line with co-IP data showing co-precipitation of both GPRC5B-FLAG and GPRC5C-FLAG with HA-EP2, whereas no co-IP was observed for GPR156-FLAG (Fig. 2B).

Figure 2: A, Bioluminescence resonance energy transfer (BRET) between GPRC5B-NLuc, GPRC5C-NLuc, GPR156-NLuc or NLuc alone (donors) and EP2-Venus (acceptor) in HEK cells transfected with different ratios of donor and acceptor plasmids; data are expressed as ratio of acceptor emission (535 nm) to donor emission (460 nm) (n=4 per condition). **B,** Western blot detection of HA and FLAG signals in lysates of HEK cells expressing HA-tagged EP2 in combination with empty vector (EV), FLAG-tagged human GPRC5B, GPRC5C, and GPR156: “input” shows lysates before, “pulldown HA” after immunoprecipitation with anti-HA beads. Data are means \pm SEM; comparisons between groups were done using two-way ANOVA with Sidak’s multiple comparison test. ****, $P < 0.0001$.

Finally, we investigated whether the NanoBRET interaction between GPRC5B and EP2 is affected by decoy peptides. HEK cells were transfected with GPRC5B-NLuc and EP2-Venus and BRET was determined under basal conditions as well as in the presence of 10 μ M control or target peptide. In line with disturbed physical interaction between GPRC5B and EP2, we observed reduced energy transfer between the partners in target peptide treated cells, but not in control peptide treated cells (Fig. 3).

Figure 3: Bioluminescence resonance energy transfer (BRET) between GPRC5B-NLuc (donor) and EP2-Venus (acceptor) in HEK cells transfected with different ratios of donor and acceptor plasmids and then cultured for 1 h in the presence or absence of 10 μ M control or target peptide. Data are expressed as ratio of acceptor emission (535 nm) to donor emission (460 nm) ($n=4$ per condition). Data are means \pm SEM; comparisons between groups were done using two-way ANOVA with Sidak's multiple comparison test. ****, $P < 0.0001$.

Taken together, both FRET and NanoBRET data support the specificity of the GPRC5B/EP2 interaction observed in co-IP experiments. We have added FRET data to Supplemental Figure 6 and describe them on page 8, lines 5-6. The NanoBRET data including compound effects on NanoBRET are shown in Suppl. Fig. 11C and are described on page 11, lines 22-23.

2. Related to previous review 1b: as the authors think that colocalization of the two receptors supports their interaction or dimerization, can their colocalization be disrupted by decoy peptides?

Following the reviewer's suggestion we determined the Pearson's coefficient for colocalization of HA-EP2 and G5B-Myc in HEK cells treated with control peptide, target peptide or no peptide. We observed a reduction of colocalization in target peptide treated samples, but not control peptide samples (Fig. 4).

Figure 4: A,B, Effect of target and control peptides on the co-localization of GPRC5B-FLAG/Myc (G5B-Myc) with HA-EP2 in HEK cells. 48 h after co-transfection of HA-EP2 and G5B-Myc plasmids, HEK cells were cultured for 1 h with or without 10 μ M control or target peptide. Cells were then fixed, permeabilized and stained with anti-Myc and anti-HA antibodies and the colocalization of both signals was determined (A, exemplary photomicrographs; B, statistical evaluation of the Pearson's coefficient, n=10-11 cells). Data are means \pm SEM; comparisons between groups were done using one-way ANOVA with Tukey's multiple comparison test. ****, $P < 0.0001$.

We have added these data as Supplemental Figures 11A,B to the revised manuscript and describe them in the results section on page 11, line 21-22.